# Suppression of kernel vibrations by layer-by-layer ligand engineering boosts photoluminescence efficiency of gold nanoclusters

Yuan Zhong[1,7], Jiangwei Zhang [2,7], Tingting Li[3], Wenwu Xu [4], Qiaofeng Yao[5], Min Lu[1], Xue Bai[1], Zhennan Wu[1] ✉, Jianping Xie [6] ✉ & Yu Zhang[1] ✉

The restriction of structural vibration has assumed great importance in attaining bright emission of luminescent metal nanoclusters (NCs), where tremendous efforts are devoted to manipulating the surface landscape yet remain challenges for modulation of the structural vibration of the metal kernel. Here, we report efficient suppression of kernel vibration achieving enhancement in emission intensity, by rigidifying the surface of metal NCs and propagating as-developed strains into the metal core. Specifically, a layer-by-layer triple-ligands surface engineering is deployed to allow the solution-phase Au NCs with strong metal core-dictated fluorescence, up to the high absolute quantum yields of $90.3 \pm 3.5\%$. The as-rigidified surface imposed by synergistic supramolecular interactions greatly influences the low-frequency acoustic vibration of the metal kernel, resulting in a subtle change in vibration frequency but a reduction in amplitude of oscillation. This scenario therewith impedes the non-radiative relaxation of electron dynamics, rendering the Au NCs with strong emission. The presented study exemplifies the linkage between surface chemistry and core-state emission of metal NCs, and proposes a strategy for brighter emitting metal NCs by regulating their interior metal core-involved motion.

Properties of metal nanoparticles are to a great extent ruled by their structural motion strength in frequency, anisotropy, polarity, etc., at the mercy of vibration and rotation behaviors of intrinsic total structure and their anti-jamming ability to external environment[1–5]. This correlation becomes more prominent when the length scale downs to the regime of few- to hundred-atom metal nanoclusters (NCs), which are protected by an organic monolayer with the entire diameter within a few nanometers (typically <3 nm). The structure of ultrasmall metal NCs can be described by a well-defined metal(0)@metal(I)-ligand core-shell scheme, in which the metal(I)-ligand "staple motifs" wrap over the metal(0) kernel following a "divide-and-protect" model[6,7]. Such ultrasmall size and structural uniqueness enable the structural motion of metal NCs active and to be regarded as the "metallic molecules", which adds to the impacts of structural vibration on the kinetics of matter and energy transformation, and hence the properties[8–12]. Moreover, ultrasmall metal NCs featuring atomic precision in size, composition, and atomic packing mode, exhibit the single metal-atom or single ligand-molecule dependent properties[13–20]. Therefore, studies on engineering

[1]State Key Laboratory of Integrated Optoelectronics, College of Electronic Science and Engineering, Jilin University, Changchun 130012, P. R. China. [2]Innovation Center of Energy Material and Chemistry; College of Chemistry and Chemical Engineering, Inner Mongolia University, Hohhot 010021, P. R. China. [3]College of Materials Science and Engineering, Jilin Jianzhu University, Changchun 130012, P. R. China. [4]Department of Physics, School of Physical Science and Technology, Ningbo University, Ningbo 315211, P. R. China. [5]Joint School of National University of Singapore and Tianjin University, International Campus of Tianjin University, Binhai New City, Fuzhou 350207, P. R. China. [6]Department of Chemical and Biomolecular Engineering, National University of Singapore, Singapore 117585, Singapore. [7]These authors contributed equally: Yuan Zhong, Jiangwei Zhang. ✉e-mail: wuzn@jlu.edu.cn; chexiej@nus.edu.sg; yuzhang@jlu.edu.cn

the structural vibration of metal NCs deserve particular attention to permit maximization and even customization of properties.

Given the photoluminescence (PL) of metal NCs, the lack of a detailed understanding of the underlying photophysical mechanisms has greatly hampered the rational design of metal NCs with improved and tailored optical properties. To date, restriction of intra- and inter-molecular motion related to structural vibration and rotation of the metal(I)-ligand motifs has been witnessed as a well-recognized and effective design principle for achieving strong emission[21,22]. In this vein, versatile strategies were deployed by committing to rigidity improvement of the surface shell. At a single-cluster level, constructing long and/or interlocked surface motifs was identified as one of the most effective ways[23,24]; Beyond the single-cluster level, that is to treat metal NCs as building blocks to direct their aggregation, strategies of directed self-assembly, crystallization, and scaffolds confinement, etc. were widely developed[25–34]. Nevertheless, the current advance in the enhancement of surface-state emission is far from ideal, in particular when taking the oxygen-quenching and solution-processability into consideration (it generally confers solid-/crystal-state metal NCs aggregates with a bright phosphorescence but sharply attenuated in the solution state), which has not yet reached an unambiguous agreement regarding metal NCs as competitive fluorophores to serve practical applications[35]. In parallel, ultrasmall size imparts metal NCs discrete energy levels thus the molecular-like metal core-dictated emission. In view of a shielding effect of the exterior organic mono-layer, the metal core-state emission holds great potential in solving the challenging issues above, studies on which have recently attracted immense interest[36]. In comparison to the blossom of manipulation chemistry of surface-state emission, the progress on modulating metal core-state emission markedly lags behind. At present, alloying metal kernel or structural isomerization are dominant methods to affect structural rigidity and subsequent relaxation of electron dynamics[37–41], giving rise to enhanced core-state emission but still at a low level so far (commonly below 20% photoluminescence quantum yields, PLQY). A nontrivial undertaking remains as mapping out how to modulate metal kernel to considerably restrict its structural vibration, meeting the increasing demand of concurrently improving emission intensity and colloidal stability of metal NCs.

Herein, we report modulation of the metal kernel of metal NCs to suppress their low-frequency acoustic vibration (i.e., the coherent oscillation) in an indirect way, that is to deliver the confinement effect from the shell- to core-domain of metal NCs driven by a triple-ligands engineering induced surface rigidity enhancement method. Most importantly, the as-prepared colloidal metal NCs feature an ultrahigh absolute PLQY, up to $90.3 \pm 3.5\%$ at room temperature. Specifically, we employed the 6-Aza-2-thiothymine (ATT), L-arginine (ARG), and tetraoctylammonium (TOA) in sequence as a serial triple-ligands model to perform the layer-by-layer cocooning of $Au_{10}$ superatom by multiple supramolecular interactions. As a result, the surface rigidity of Au NCs is greatly improved while there is a negligible impact on the electronic properties and local structures of their metal kernel. To our surprise, the as-rigidified surface can regulate the low-frequency acoustic vibration of the metal kernel, leading to a reduction in the amplitude of coherent oscillation. This suppression of coherent oscillation brings a significant influence on the dynamics of excited-state electrons. Global fittings of their femtosecond-transient absorption (TA) map data indicate that the metal core-directed structural relaxation of electron dynamics is restrained. As a result, the triple-ligands capped Au NCs present an intense metal core-dictated fluorescence. Moreover, their molecular-state nature is verified and the possible core-shell electronic relaxation is ruled out through a series of pump-power and pump-energy dependent TA measurements. Notably, the as-demonstrated surface engineering strategy not only achieves intense core-state emission but also provides an opportunity to endow metal NCs with diverse surface chemistry[42]. In an exemplified case, utilizing the water-soluble ligand of trimethylphenylammonium (TMPA) as the outmost ligand instead of the toluene-soluble TOA, the water-soluble Au NCs with a high PLQY of $86.7 \pm 3.6\%$ can be obtained. Our findings suggest a versatile triple-ligands engineering strategy in the design of colloidal metal NCs with intense luminescence, adding to their acceptance in diverse sectors of practical applications.

## Results

### Photoluminescent properties and structural anatomy

The ATT, ATT-ARG, and ATT-ARG-TOA protected Au NCs (abbreviated as Au-1, Au-2, and Au-3 hereafter) were prepared through a layer-by-layer triple-ligands self-assembly method (Fig. 1a). In a typical synthesis, the ligand of ATT was employed as a reductant to Au(III) ions and the first-layer protectant. The as-resulted Au-1 NCs show two prime absorption peaks located at 403 and 473 nm, and exhibit a weak emission centered at 532 nm with an absolute PLQY < 0.3% (Fig. 1b and Supplementary Fig. 1). After incorporating the second-layer ligand of ARG to Au-1 through the hydrogen bonding between the guanidine group of ARG and neighboring pyrimidine ring of ATT, the absorption curve features three sharper peaks centered at 403, 458, and 506 nm, and the absolute PLQY of as-obtained Au-2 NCs is improved to $59.6 \pm 2.8\%$ with constant emission at 532 nm (Fig. 1b and Supplementary Fig. 1). Most interesting, further dramatic enhancement in absolute PLQY, up to the ultrahigh value of $90.3 \pm 3.5\%$ at 538 nm (Fig. 1b and Supplementary Fig. 1), is achieved when the third-layer ligand of TOA (20 mg mL$^{-1}$, Supplementary Fig. 2) anchored on the surface of Au-2 NCs, by the electrostatic interactions between the deprotonated carboxyl of ARG and quaternary ammonium cations (QACs) of TOA. The differences in the absorption characters between Au-1 and Au-2 NCs indicate that the core structure is slightly distorted from Au-1 to Au-2 NCs, but almost maintained from Au-2 to Au-3 NCs. Meanwhile, Au-3 NCs manifest a lengthened fluorescent lifetime of 61.0 ns, compared to that of 43.8 ns in Au-2 NCs and 3.5 ns in Au-1 NCs, respectively (Fig. 1c). Combining the measurements of the absolute PLQYs and fluorescent lifetimes of serial Au NCs excited by near-band gap energies, we found the radiative decay rate ($k_r$) is slightly promoted (~3.6-fold) in the sequence of Au-1, Au-2, and Au-3 NCs, which should be related to the increased refractive index ($n_{eff}$) from 1.3334 in Au-1 NCs, 1.3339 in Au-2 NCs, to 1.4943 in Au-3 NCs (Supplementary Table 1)[43]. Instead, the non-radiative decay rate ($k_{nr}$) is sharply declined (~57.4-fold), indicating that the as-adopted triple-ligands surface engineering can greatly suppress the non-radiative relaxation channels. The inherent features of narrow FWHM of ~30 nm, small Stokes shift, as well as the ns-level decay time, suggest the nature of metal core-dictated fluorescence in serial Au NCs[44,45], which can be maintained in the powder and thin film state (Supplementary Figs. 3–6 and Table 1). The two-dimensional (2D) photoluminescent maps and time-resolved emission spectra (TRES) studies suggest the fluorescence of serial Au NCs is excitation- and time-independent, evidenced by their single emissive center (Supplementary Figs. 7, 8).

To unravel the underlying mechanism of fluorescence boosting in the serial Au NCs, the structural and compositional evolutions during the layer-by-layer triple-ligands engineering need to identify. We first carried out the transmission electron microscopy (TEM), $^1$H-nuclear magnetic resonance ($^1$H-NMR), and matrix-assisted laser desorption ionization time-of-flight (MALDI-TOF) mass spectra measurements (Supplementary Figs. 9–11). As a result, the chemical formula of serial Au NCs can be well-defined as $Au_{10}(ATT)_6$, $Au_{10}(ATT)_6(ARG)_3$, and $Au_{10}(ATT)_6(ARG)_3(TOA)_3$, corresponding to Au-1, Au-2, and Au-3, respectively. As for the crystal structure of serial Au NCs, we have tried to grow their high-quality single crystals but all our attempts failed to acquire X-ray-quality ones. This should be attributed to (i) the supramolecular interactions of metal NCs are more diverse and complex (e.g., the synergistic and competitive effect between hydrogen bonding and electrostatic interactions), and thereby challenging to balance

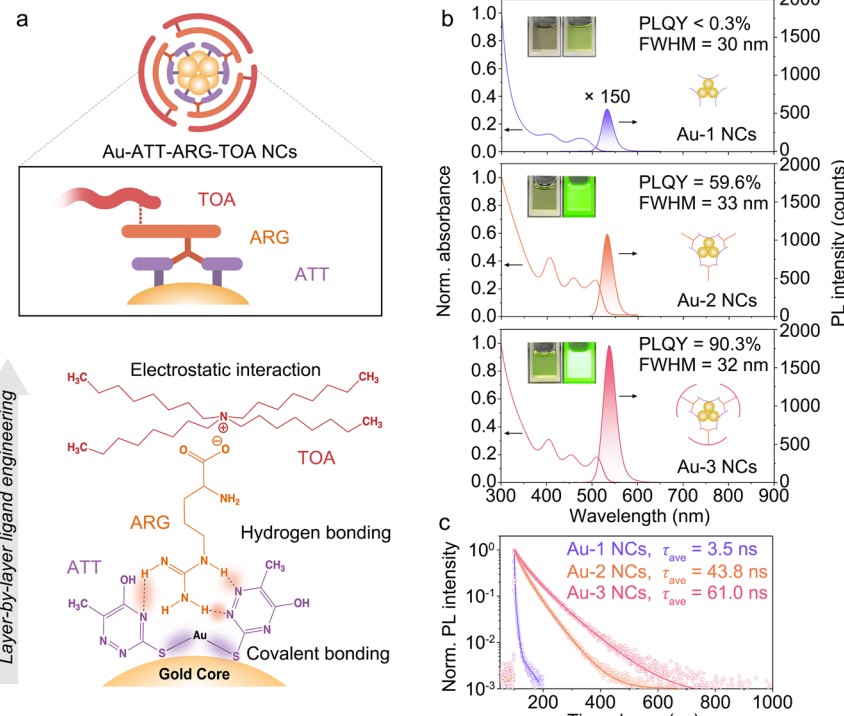

**Fig. 1 | Structural evolution and optical properties of Au-1, Au-2, and Au-3 NCs.**
**a** Schematic illustration of the triple-ligands layer-by-layer self-assembly evolution from the initial Au-1, intermediate Au-2, to terminal Au-3 NCs. The schematic items colored in purple, orange, and pink denote to gold NCs coated with ATT, ARG, and TOA ligands, respectively. **b** UV-vis absorbance and luminescence spectra of Au-1,
Au-2, and Au-3 NCs; the insets are the digital optical photographs of Au-1, Au-2, and Au-3 NCs under ambient atmosphere (left) and 365 nm light irradiation (right). The corresponding prototypes of each NC are depicted in the photographs.
**c** Luminescence time-resolved spectra of Au-1, Au-2, and Au-3 NCs upon 405 nm excitation. Source data are provided as a Source Data file.

multiple driving forces for crystallization; (ii) the triple-ligands layer-by-layer self-assembly in our cases enables relatively flexible and variant surface environments of metal NCs, which further impedes the formation of long-range-ordered packing to generate high-quality single crystals. Moreover, to push our step in addressing their atomic-level crystal structure, we proposed a reasonable structure in the supplementary information file. It should be mentioned that an empirical rule to estimate the crystal structure of water-soluble metal NCs is comparing their absorption spectra with that of known metal NCs[46]. The target metal NCs of Au-1 in this work is $Au_{10}(SR)_6$, whose crystal structure has already been predicted[47]. We have simulated the UV-vis absorption spectrum of as-reported $Au_{10}(SR)_6$ NCs and found it matched well with our experimentally recorded absorption spectrum (Supplementary Fig. 12), indicating similar Au-S frameworks between Au-1 NCs and the reported $Au_{10}(SR)_6$ NCs[48]. That is, the $Au_7$ kernel is fused by two $Au_4$ tetrahedrons and three monomeric [S-Au-S] staple motifs are anchored on the surface of the $Au_7$ kernel by covalent Au-S bonding (Supplementary Fig. 13). While the second-layer ligands of ATT and the third-layer ligands of TOA can be further anchored through supramolecular hydrogen bonding and electrostatic interactions to form Au-2 and Au-3 NCs, respectively.

## Table 1 | Optical parameters of Au-1, Au-2, Au-3, Au-2-TMPA, and Au-2-TMBA NCs

| Gold NCs | Absolute PLQY | FWHM (nm) | Stokes shift (cm⁻¹) |
|----------|---------------|-----------|---------------------|
| Au-1 | <0.3% | 30 | 1479 |
| Au-2 | 59.6 ± 2.8% | 33 | 826 |
| Au-3 | 90.3 ± 3.5% | 32 | 982 |
| Au-2-TMPA | 86.7 ± 3.6% | 30 | 845 |
| Au-2-TMBA | 79.5 ± 3.2% | 31 | 831 |

With respect to such structural implications and to address a more precious identification, the serial Au NCs are further characterized by the X-ray techniques (Fig. 2). In the synchrotron-based X-ray absorption near edge structure (XANES) spectra (Fig. 2a), all the absorption signals of Au-1, Au-2, and Au-3 NCs superposed each other in the gap between the Au foils and AuS references. The white line corresponding to the absorption edge in the XANES region refers to the electronic transition from core level to unoccupied 5d valence states of Au atoms. The variation in the intensity of the white-line peak can promote the shift of absorption edge toward different directions in energy, accounting for different oxidation states[49,50]. Based on this principle, we confirm that the metal kernel in serial Au NCs keeps a coincident oxidation state. This point is in accordance with their X-ray photoelectron spectroscopy (XPS) profiles (Fig. 2b). The Au 4f XPS peak positions of serial Au NCs maintain consistency and locate in the transition region between those of Au(0) nanoparticles and Au(I)-(p-MBA) complexes. Besides, it also presents a constant ratio of Au(I) to Au(0) of 3:2 (Supplementary Fig. 14), which agrees well with the as-inferred molecular formulas above. Next, the Fourier-transformation (FT) at the R space of extended XAFS spectra (EXAFS) is executed to quantitatively acquire local atomic structural information of serial Au NCs (Fig. 2c). In detail, the scattering path in the first and second coordination shell, corresponding to the Au-S and $Au_{center}$-$Au_{apex}$ bonding, where $Au_{center}$ and $Au_{apex}$ stand for Au atoms at the center and apex of the $Au_7$ kernel, is consistently centered at 2.42 and 2.76 Å for the serial Au NCs. This uniformity can be further visualized in the Au $L_3$-edge 3D wavelet-transformed (WT)-EXAFS spectra (Fig. 2d–f and Supplementary Fig. 15). As shown in Fig. 2d–f, the Au-S, and $Au_{center}$-$Au_{apex}$ bonding related scattering path signals are probed at [χ(k), χ(R)] of [4.0 2.42] and [8.4 2.76] for Au-1 NCs, [4.2 2.42] and [8.4 2.76] for Au-2 NCs, and [4.0 2.42] and [8.8 2.76] for Au-3 NCs, respectively. The subsequent fine fitting results of least-squares χ(R) and k²χ(k)

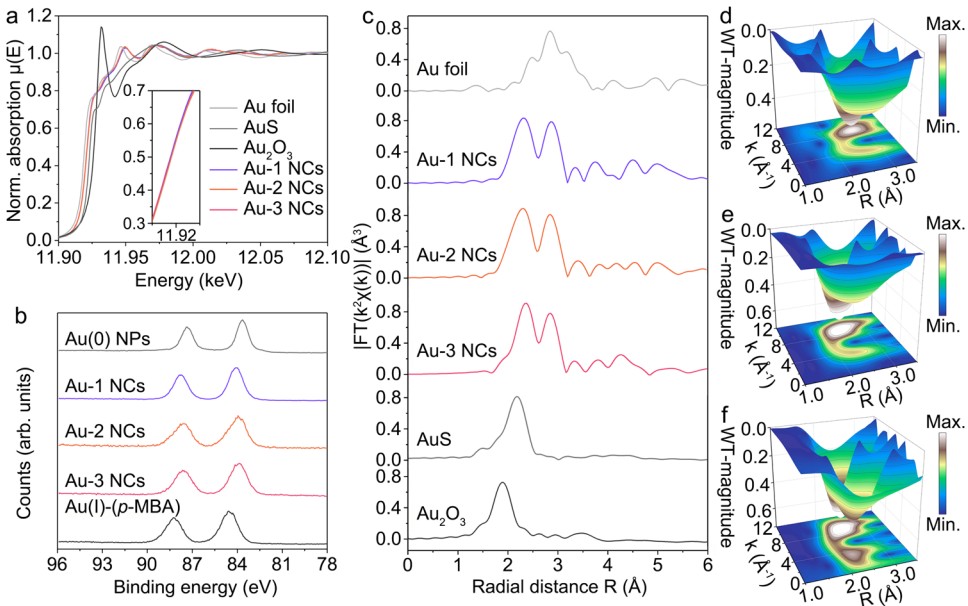

**Fig. 2 | Electronic properties and local structures of Au-1, Au-2, and Au-3 NCs.**
**a** Au $L_3$-edge XANES spectra of Au foil, AuS, $Au_2O_3$, Au-1, Au-2, and Au-3 NCs; the inset shows the magnified XANES spectra in the energy range of 11.919-11.925 KeV. **b** Au 4$f$ XPS spectra of Au(0) NPs (gold NPs), Au(I)-(*p*-MBA) complexes, Au-1, Au-2, and Au-3 NCs. **c** Au $L_3$-edge FT-EXAFS spectra of Au foil, AuS, $Au_2O_3$, Au-1, Au-2, and Au-3 NCs. 3D WT for the $k^2$-weighted EXAFS signals of (**d**) Au-1, (**e**) Au-2, and (**f**) Au-3 NCs. Source data are provided as a Source Data file.

space spectra demonstrate that our proposed structures of serial Au NCs are reliable (Supplementary Fig. 16). The quantitative coordination number (CN) was also extracted for three Au NCs (Supplementary Tables 2–4), which exhibits a similar value of $CN_{Au-S} = \sim1.1$ and $CN_{Au-Au} = \sim3.3$ in average. This local structure information manifests again the fact that the different surface landscape makes a small influence on the metal kernel and the Au(I)-S interface of serial Au NCs.

On the basis of the above results, we conclude that the core-state fluorescence boosting involved structural evolution exclusively stems from the enhancement of surface rigidity in the Au NCs series. In this regard, it suggests that tailoring in the surface chemistry of metal NCs has an unexpected influence on the core-state emission of metal NCs.

## Ultrafast electron dynamics

To reveal the underlying mechanism of fluorescence boosting, we performed the femtosecond-TA measurements for serial Au NCs. As shown in Fig. 3a–c, these TA maps were recorded upon fixed excitation of 400 nm and pump power of 300 μJ cm$^{-2}$. The TA map of Au-1 NCs shows distinct ground-state bleaching (GSB) at 492 nm. After anchored with ARG and TOA ligands, their GSB peaks get sharper and the corresponding peak position shift to 511 and 515 nm for Au-2 and Au-3 NCs, respectively. A broad excited-state absorption (ESA) band following behind their GSB is observed, which indicates the dense excited states of serial Au NCs[51]. As to GBS and ESA bands, their decay time gradually slows down from Au-1, Au-2 to Au-3 NCs (Supplementary Fig. 17). And, the ESA bands in Au-2 and Au-3 NCs experience a subtle redshift in the initial time scale of 5 ps, illustrating that the prominent re-excitation channels are altered from a higher excited state to a relatively lower one. This tendency becomes more obvious when we normalize the ESA and GSB kinetic traces of serial Au NCs (Fig. 3d). That is, Au-1 NCs exhibit the fastest decays and more overlap trend, while Au-3 NCs are decaying slowest and less overlapped. It means the non-radiative portion in Au-3 NCs has been greatly suppressed hence their longest lifetime of excited-state electrons appeared. Thereafter, we carried out the singular value decomposition (SVD) and global fitting of TA maps to extract the time constants of their decay processes. The global fittings generally give four decay components: 320 fs, 2.2 ps, 80.6 ps, and >1 ns for Au-1 NCs; 240 fs, 2.9 ps, 80.7 ps, and >1 ns

for Au-2 NCs; 118 fs, 4.1 ps, 80.5 ps, and > 1 ns for Au-3 NCs, respectively (Fig. 3e–g and Supplementary Fig. 18). Based on the distinct discrepancy in the time scale, we can rationally assign different time constants to different relaxation processes of excited electrons for decoding the photodynamics in serial Au NCs. First, the long-lived (>1 ns) relaxation component is assigned to the transition from the lowest excited singlet state ($S_1$) to the ground state ($S_0$), and this relaxation is not possible to acquire an accurate lifetime due to the limited decay time window (~8 ns) in our TA measurements. Nevertheless, it can be indirectly reflected by the profiles of fluorescent lifetimes during the PL process, as shown in Fig. 1c. Second, we take the initial several hundreds of femtoseconds (320, 240, and 118 fs) into consideration. The applied excitation energy of 3.10 eV (400 nm) is obviously larger than their HOMO-LUMO gap energies (Supplementary Fig. 19), and thus can give excess excited state energy to pump electrons higher than the $S_1$ state (e.g., hot $S_n$ state). Therefore, this ultrafast decay component should be better assigned to the internal conversion (IC) of hot electrons from $S_n$ to the $S_1$ state ($S_n \rightarrow S_1$). Third, the subsequent few-picosecond (2.2, 2.9, and 4.1 ps) and dozens of picoseconds (80.6, 80.7, and 80.5 ps) decays in all Au NCs should be both contributed to the structural relaxation in the sublevels of $S_1$. Especially, the few-picosecond component is mainly caused by above mentioned core-directed structural vibration, while the dozens of picoseconds component originates from core-ligand-directed interface vibration[52]. Both processes result in the dissipation of the energy of excited electrons through the redistribution of the electron density to cool down them to the lowest $S_1$ state. The first femtosecond decaying component shortens from 320, 240, to 118 fs from Au-1, Au-2, to Au-3 NCs, indicating the promoted IC relaxation whose occurrence is generally related to the varied energy gap between $S_n$ and $S_1$, and their electronic state[53]. Intriguingly, we also find the few-picosecond decaying component associated to core-directed structural vibration experiences a increase from 2.2, 2.9, to 4.1 ps. By contrast, the dozens of picoseconds component related to core-ligand-directed interface vibration is relatively more stable (i.e., 80.6, 80.7, and 80.5 ps). These results suggest that the core-directed structural vibration is the dominant non-radiative pathway for luminescence quenching, which has been inhibited through the triple-ligands surface engineering

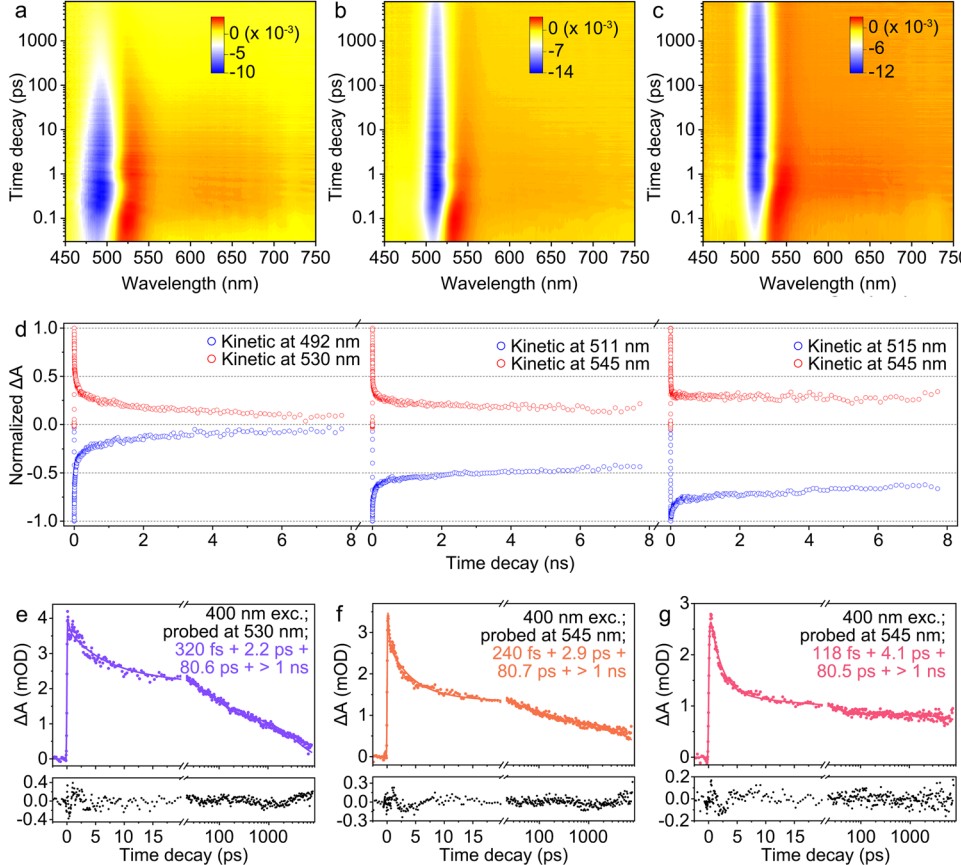

**Fig. 3 | Comparison of femtosecond-TA spectral features and their electron dynamics of Au-1, Au-2, and Au-3 NCs. a–c** Femtosecond-TA maps of Au-1, Au-2, and Au-3 NCs upon 400 nm excitation. The scaling of the color scales is the absorption intensity, and the units are milli-optical density. **d** Normalized ESA and GSB kinetic traces probed at 492 and 530 nm for Au-1 NCs, 511 and 545 nm for Au-2 NCs, and 515 and 545 nm for Au-3 NCs, respectively. **e–g** Selected kinetic decays around 530 and 545 nm and their corresponding fitting lines and residuals extracted from the global fitting of the TA maps of Au-1, Au-2, and Au-3 NCs. Symbol: ΔA is the change in absorption intensity in units of optical density. Source data are provided as a Source Data file.

strategy. As a result, more population of excited electrons can slower relax to the lowest sublevel of the $S_1$ state, and subsequently, participate in the radiative process from $S_1$ to $S_0$ state.

Of particular note, the first few-picosecond component in serial NCs is comparable to the typical electron-phonon coupling time (~1–5 ps) in metallic nanoparticles (NPs). To identify the possible metallic-state essence of the serial Au NCs, we performed pump-power-dependent TA measurements, ranging from 80 to 300 μJ cm$^{-2}$ under 400 nm excitation. As shown in Supplementary Figs. 20–25, the whole characters of the TA map stay the same except for the intensity of the signal. And the negative-linear relationship of GSB intensity as a function of pump power reveals the molecular-like single-electron transition in excited-state dynamics (Supplementary Fig. 26). Furthermore, the decay kinetics at the corresponding ESA positions are collected, normalized, and plotted (Supplementary Fig. 27), reflecting power-independent electron relaxation thus a clarification of molecular-state (i.e., non-metallic-state) nature of serial Au NCs. We also perform the corresponding global fitting treatments. As summarized in Supplementary Table 5, the stable time constants of IC and structural relaxation are clearly different from those of power-dependent electron-phonon or photon-photon coupling generally recorded in the metallic counterparts[54,55].

To unscramble the relaxation pathway of excited electrons, the pump-energy-dependent TA measurements are further carried out for serial NCs. With declining pump energy from 3.10 (400 nm) to 2.43 eV (510 nm), the recorded TA signals drop-down progressively (Supplementary Figs. 28–33). Global fitting of TA maps gives the gradually

reduced time constants for the IC relaxation but relatively stable values for the second and third picosecond decays for serial Au NCs when they were pumped by high energies. However, upon near-HOMO − LUMO gap pumping, only two picosecond-level time constants and > 1 ns component are fitted out (Supplementary Figs. 34–36 and Supplementary Table 6). It should be noted that the near-HOMO − LUMO gap pumping can merely excite the metal core state for metal NCs. The remaining two picosecond-level decay components again evidence that they belong to metal core-directed structural relaxation. In addition, the ultrafast IC decay is completely disappeared and no additional ultrafast component is detected under near-bandgap excitation for all Au NCs. Keeping this in mind, we are now safe to rule out the contribution of possible core-shell relaxation to the inherent PL of serial Au NCs because this process is typically observed in an especially short time scale (e.g., 1 ps in $Au_{25}(SR)_{18}$[56] and $Au_{38}(SR)_{24}$[57]). Therefore, the excited electrons in serial Au NCs are limited in the domain of the metal kernel, irrelevant to any surface-related non-radiative channels.

## Coherent oscillation of metal kernel

Besides the electron dynamics, femtosecond-TA can also detect phonon dynamics of metal NCs that account for the excited-state energy deactivation, which is helpful to figure out how the rigidified surface affects the metal core-related photophysics. For molecular-like metal NCs, a short impulsive laser can excite the coherent acoustic vibration of the metal core for a few to tens of picoseconds. As a result, a wavepacket motion could be produced and superimposed on the kinetic traces of electrons, through the impulsive stimulated Raman

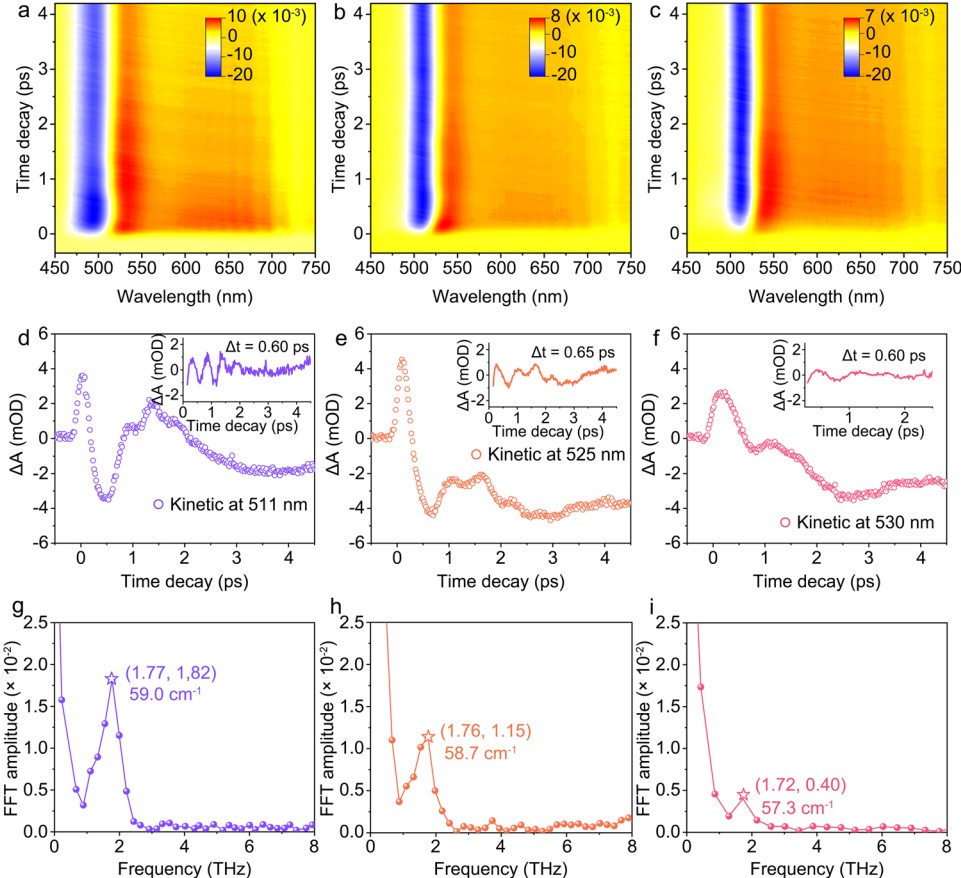

**Fig. 4 | The detection of coherent oscillations of metal kernel in Au-1, Au-2, and Au-3 NCs. a–c** Short-range femtosecond-TA data maps of Au-1, Au-2, and Au-3 NCs between 0.3 ps and 4.5 ps upon the excitation of 400 nm. The scaling of the color scales is the absorption intensity, and the units are milli-optical density. **d–f** Kinetic traces at zero positions at 511, 525, and 530 nm in the TA maps of Au-1, Au-2, and Au-3 NCs; the zero positions are the cross lines that the electron dynamic is inhibited to close to zero while phonon dynamic is visualized furthest. Insets in (**d–f**) are the pure acoustic oscillations and their periodic times are determined to be 0.60, 0.65, and 0.60 ps, respectively. **g–i** The corresponding FFT results of kinetic traces in (**d–f**) by plotting the amplitude as a function of frequency. The pentagram patterns denote the peak of FFT amplitude, and the coordinates in (**g–i**) are (1.77, 1.82), (1.76, 1.15), and (1.72, 0.40), respectively. Source data are provided as a Source Data file.

**Table 2 | The experimentally reported frequency of coherent oscillations in molecular-like metal NCs**

| Gold NCs | Au$_{246}$ | Au$_{144}$ | Au$_{38}$ | Au$_{30}$ | Rod Au$_{25}$ | Spherical Au$_{25}$ | Au$_{10}$–1/2/3 |
|---|---|---|---|---|---|---|---|
| Frequency (cm$^{-1}$) | 16.7 | 52.0 | 25.0 | 16.7 | 26.0 | 40.0 | 59.0/58.7/57.3 |
| Reference | 66 | 65 | 57, 67 | 68 | 64 | 56 | This work |

process. The coherent oscillation, which is the reflection of the excited state electronic-phonon coupling from the lowest frequency of the vibrational structure of clusters[58–60], thus can be distinguished from the TA decays at an early time at most probed wavelengths. As shown in Fig. 4a–c, we recorded the TA maps of serial Au NCs within a very short delay time range of 0.35 to 4.5 ps with 20 fs intervals. To elaborate oscillation effect, the decay kinetics at 511, 525, and 530 nm for three gold NCs are analyzed especially. As shown in Fig. 4d–f, the ultrafast signal rising is assigned to the photoexcitation process in electron dynamic owing to the high excitation energy of 400 nm. Then, it is followed by a decay process associated with obvious coherent oscillation with 2–3 periods. The pure acoustic oscillations (the insets in Fig. 4d–f) were extracted by subtracting the electron dynamics using exponential decay[61–63]. The periodic time is determined by calculating the time discrepancy between neighboring peak valleys in acoustic oscillation and is identified to be 0.60, 0.65, and 0.60 ps for Au-1, Au-2, and Au-3 NCs, respectively. Furthermore, the Fast Fourier Transform (FFT) was conducted for periodic signal processing. As shown in Fig. 4g–i, serial Au NCs exhibit similar vibration

frequency of 59.0 (1.77 THz), 58.7 (1.76 THz), and 57.3 cm$^{-1}$ (1.72 THz) but a reduction in the variation amplitude. The relatively low-frequency vibration in serial NCs indicates the oscillations are categorized into acoustic vibration and are better ascribed to the quadrupolar-like vibration mode which is induced by the periodical expansions and contractions of atoms along one direction associated with out-of-phase oscillations in the perpendicular plane[64,65]. Plenty of pioneering works have confirmed that the oscillation frequency of Au NCs still depends on their overall size, but has a more irregular dependence with respect to larger spherical gold NPs due to the discrete structure of Au NCs and their ligands (Table 2)[56,57,64–68]. The similar variation frequency in serial NCs demonstrates their similar local structure of metal kernel. The reduced amplitude in oscillation qualitatively illustrates that the confinement effect provided by the enhanced surface rigidity can be delivered into metal kernel and impressively suppress their core-directed structural vibration. This scenario subsequently affects the dynamics of the excited electron associated with their contribution to the core-state fluorescence boosting of metal NCs.

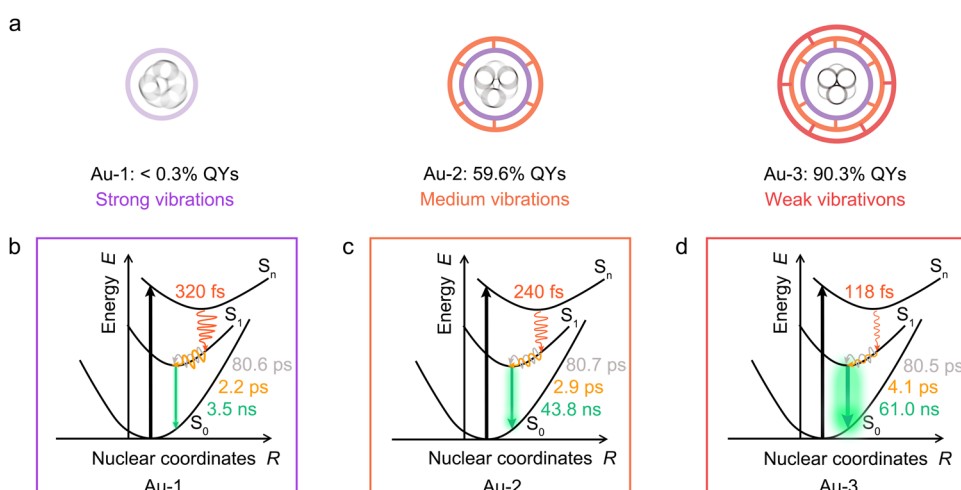

**Fig. 5 | Schematic illustration of the excited-state electron dynamics and coherent vibrations of Au-1, Au-2, and Au-3 NCs. a** Schematic illustration of coherent vibrations and corresponding QYs of Au-1, Au-2, and Au-3 NCs. The purple, orange, and red circles stand for layer-by-layer coating of ATT, ARG, and TOA ligands, and the black circles refer to the central gold atoms in three Au NCs. The energy plots show the excited-state electron dynamics of (**b**) Au-1, (**c**) Au-2, and (**d**) Au-3 NCs. The red, orange, and gray solid curves in the energy diagrams stand for internal conversion, core-directed, and core-ligand-related structural vibration, respectively. The green solid lines with outer glow denote the fluorescent relaxation. The symbols of $S_n$, $S_1$, and $S_0$ are the higher singlet state, the lowest singlet state, and the ground state, respectively.

Based on the above results obtained from ultrafast excited-state dynamics and optical coherent oscillations, now we are ready to construct a schematic illustration (Fig. 5) to correlate the coherent vibration and PL properties in three Au NCs. In a general model, the external excitation can pump electrons into a high-energy singlet state of $S_n$. Then, the excited electrons experience ultrafast IC relaxation within several hundreds of femtoseconds to cool into the $S_1$ state. The singlet populations continue to non-radiatively relax to the $S_1$ state with the lowest energy through the multiple core-directed and core-ligand-directed structural vibration in few-picosecond and dozens of picosecond time scales, respectively. Nevertheless, the core-directed structural vibration gives dominating non-radiative channels for the contribution of hot electrons in the $S_1$ state with respect to the core-ligand-directed structural vibration. Subsequently, the excited electrons localized in the gold core return to the $S_0$ ground state by radiatively emitting the *ns*-level core-sate green fluorescence. Note that there is no additional ultrafast core-shell electronic relaxation in the excited-state dynamics, which is understandable because of the lack of *μs*-level surface-state phosphorescence in three Au NCs. The triple-ligands layer-by-layer decorations of ARG and TOA on the surface of Au-1 NCs enable the indirect delivery of the confinement effect from the surface into the gold core, leading to suppressed coherent oscillation in Au-2 and especially Au-3 NCs. As a result, the suppressed non-radiative relaxation channels in turn boost the core-state green fluorescence[69], Thus, the absolute QYs are enhanced from <0.3% to 59.6% from Au-1 to Au-2 NCs, and a high value of 90.3% is achieved in the terminal Au-3 NCs.

## The versatility of vibration suppression boosted emission

It is clear that the core-state photoluminescence of serial Au NCs is tightly related to the rigidity of the surface structure. To verify the versatility of such the design principle and considering the intense demand of the water-soluble highly luminescent metal NCs, we deployed a series of water-soluble ligands to modify the Au-2 NCs. Wherein, as to the trimethylphenylammonium (TMPA) or benzyltrimethylammonium (TMBA) modified Au-2 NCs (Au-2-TMPA, Au-2-TMBA for short hereafter), under an optimized feeding concentration of 1.0 mg mL$^{-1}$ or 2.5 mg mL$^{-1}$ respectively, they can present an excellent absolute PLQY of 86.7 ± 3.6% and 76.5 ± 3.2% in the water solution at room temperature, respectively (Supplementary Figs. 37–39). And

their corresponding fluorescent lifetimes were also lengthened from 20.6 ns to 38.7 and 31.4 ns (Supplementary Figs. 37c, 38c), respectively. With respect to the amphiphilicity of QACs and the water solubility of as-synthesized Au-2-TMPA and Au-2-TMBA NCs, the absorbed TMPA or TMBA cations are most likely to form rigidified ligand-double-layers on the surface of Au-2 NCs (Supplementary Fig. 40) to invoke the strong π···π interactions between two adjacent phenyls rings[70].

To check whether the underlying mechanism of improved emission performance in Au-2-TMPA and Au-2-TMBA NCs is identical to that of Au-3 NCs. We first perform the $^1$H-NMR test, which demonstrates that the QACs of TMPA and TMBA were anchored on the surface landscape of Au-2-TMPA and Au-2-TMBA NCs (Supplementary Figs. 41, 42). Further XPS, XANES, and EXAFS data manifest their similar electronic properties and local structures to that of pristine Au-2 NCs (Supplementary Figs. 43–45 and Supplementary Tables 7, 8). In regard to the electron dynamics, global fitting of the femtosecond-TA maps of Au-2-TMPA and Au-2-TMBA NCs also gives reduced IC time constants of 130 and 145 fs and time constants for core-directed structural vibration of 3.7 and 3.4 ps (Supplementary Fig. 46), respectively. While their time constants for core-ligand-directed interface vibration are relatively stable, to be 80.6 and 80.5 ps, respectively. These results unambiguously confirm that the non-radiative pathways induced by kernel vibration have been efficiently suppressed. Further careful checks on the phonon dynamics at the zero position of 530 nm in the corresponding short-range femtosecond-TA maps of Au-2-TMPA and Au-2-TMBA NCs corroborate such inference again. The periodic time of 0.65 ps in Au-2-TMPA and Au-2-TMBA NCs is the same as that in Au-2 NCs (Supplementary Figs. 47, 48a, 48b), and their corresponding variation frequencies of 1.71 and 1.73 THz are also comparable to that of 1.76 THz calculated in Au-2 NCs, evidencing that Au-2, Au-2-TMPA, and Au-2-TMBA NCs share a similar structure of metal core. More importantly, the progressively weakened oscillation amplitude from Au-2 NCs to the Au-2-TMBA NCs and Au-2-TMPA NCs (Supplementary Fig. 48c, 48d) suggests that the coherent oscillation of metal kernel has been greatly suppressed. However, these amplitudes of coherent oscillation are still larger than that detected in Au-3 NCs, which is mostly owing to the more rigid nature of the TOA ligand and the influence of vast tractive forces in the water phase. Next, additional attempts by using other water-soluble QACs with different lengths of alkyl chain failed to further enhance the optical properties of Au-2 NCs (Supplementary Figs. 49, 50). This can be ascribed to that they

are hard to form the double-layer shell structure or the less rigid essence of these ligands.

## Discussion

Notably, the structures of the serial Au NCs are not absolutely clear and there is reason to suspect structural differences as indicated by observed differences in the optical properties; However, it does not affect the conclusion and our emphasis in this work. Namely, the electronic structure and corresponding photoluminescence mechanism of the serial Au NCs are consistent. Only the different kernel vibrations induced non-radiative relaxation of electron dynamics are involved, resulting in the boosted absolute PLQY from Au-1, Au-2, to Au-3 NCs. Besides, the quantitative relationship between the amplitude of the oscillation in femtosecond time-resolved signal and the amplitude of the low-frequency coherent oscillation has not been clarified in all details and a fundamental relationship could not have been established from the presented data, requiring further studies.

In summary, we have synthesized core-state-emissive Au NCs with a high absolute PLQY of $90.3 \pm 3.5\%$ through a layer-by-layer triple-ligands self-assembly strategy. The improved surface rigidity of ATT-ARG-TOA co-protected Au NCs can deliver the confinement effect into the metal kernel, leading to a reduction in the amplitude of low-frequency coherent oscillation of metal kernel. This effect further works to weaken the non-radiative-related pathway of metal core-directed structural relaxation of electron dynamics and devote more energy to core-state fluorescence boosting. The presented layer-by-layer triple-ligands self-assembly strategy also can be popularized to enrich the surface chemistry of metal NCs in diverse colloid environments for broadening their scope of practical applications. This work exemplifies an effective design principle to modulate the metal kernel and further regulate their kernel vibration (i.e., coherent oscillation) by a layer-by-layer ligand engineering.

## Methods

### Materials and reagents

Hydrogen tetrachloroaurate(III) trihydrate ($HAuCl_4 \cdot 3H_2O$, ≥49.0 Au basis) was purchased from Sigma Aldrich; 6-Aza-2-thiothymine (98%) was purchased from Alfa Aesar; L-arginine (>99.0%), tetra-octylammonium bromide (98%), tetrapentylammonium (TPA) bromide (99%), tetramethylammonium (TMA) chloride (99%), dodecyl-trimethylammonium (DTA) chloride (99%), stearyl trimethyl ammonium (STA) bromide (99%), hexadecyl trimethyl ammonium (HTA) bromide (99%), Trimethylphenylammonium chloride (98%), Benzyl-trimethylammonium (TMBA) chloride (98%), sodium hydroxide (NaOH, 99.99%) and Poly(vinyl alcohol) (PVA) 1795 (92–94% hydrolyzed) were purchased from Aladdin Reagent Co. (Shanghai, China); didodecyldi-methylammonium (DDA) bromide (99%) was purchased from MACKLIN Reagent Co. (Shanghai, China); hydrochloric acid (HCl, analytical reagent) and toluene (analytical reagent) were purchased from Xilong Scientific Co., Ltd. (Shanghai, China). All chemicals were used as received without additional purification. Ultrapure Millipore water (18.2 MΩ) was used throughout the experiments to dissolve regents and as reaction media.

### Synthesis of Au-1 NCs

In a typical synthesis of Au-1 NCs, 5 mL ATT (80 mM, dissolved in 0.2 M NaOH solution) was added to 5 mL of $HAuCl_4$ solution (10 mg mL$^{-1}$) under 600 rpm stirring at room temperature. The body color of the solution transformed from reddish-brown to light yellow, indicating the reduction of Au(III) ions by thiolate. The pH of the mixture system was carefully tuned to 9.50 by adding a certain amount of diluted hydro-chloric acid (0.12 M). This processing could slightly deepen the yellow color of the solution. The scintillation vials were then packaged with a layer of tinfoil and continued to stir for 1 h. The as-obtained products were subjected to purification by using an ultrafiltration tube (Millipore,

50 kDa). The remaining concentrated solution was taken out and redissolved in 10 mL ultrapure water. Several drops of freshly prepared NaOH solution (0.2 M) were added to adjust the pH back to 11.20. The as-synthesized Au-1 NC was stored at 4 °C for subsequent preparation.

### Synthesis of Au-2 NCs

Prior to this synthesis, ARG aqueous solution (40 mM) was first pre-pared. Then, 1.11 mL of as-prepared ARG stock solution was dropped into the above Au-1 solution (10 mL) under 600 rpm stirring, which could immediately turn the upper surface of the mixture into light green. The scintillation vials were then placed in a water bath and stirred at 40 °C for 24 h, and Au-2 NCs with the entire green body could be obtained later. This product was moved to a refrigerator and aging at 4 °C for 2 days to stabilize its physicochemical properties.

### Synthesis of Au-3 NCs

In a typical synthesis, 1 mL of as-prepared Au-2 NCs was dissolved in 9 mL ultrapure water. Then, 1 mL of TOA-Br solution (5 ~ 25 mg mL$^{-1}$, dissolved in toluene) was quickly added to the above aqueous solution and the as-used scintillation vials were placed on a vortex mixer to adequately shake two immiscible solutions for 5 min. Due to the strong electrostatic interaction between the carboxyl anions of ATT and the quaternary ammonium cations of TOA, the phase transformation of gold NCs from aqueous solution to toluene phase could be achieved after shaking. The mixture was placed until the water and toluene layers were completely separated. The toluene solution of bright-green-emitting Au-3 NCs was then moved into a new 20 mL scintilla-tion vial and stored at 4 °C for the following measurements.

### Preparation of the powder of gold NCs

The high-quality powders of Au-1, Au-2, and Au-3 NCs were prepared through freeze-drying. The gold NCs were first pre-solidified thor-oughly by using liquid nitrogen, and then freeze-dried at −65 °C and 5 Pa for 2 days. The as-obtained powders were timely transformed into gloveboxes to prevent the reactions with water and oxygen in the air. These powders were then used for optical measurements and $^1$H-nuclear magnetic resonance (NMR) spectra tests.

### Preparation of the film of gold NCs

All the thin films of Au-1, Au-2, and Au-3 NCs were prepared by spin coating. Especially, considering the difficulty of film formation for the aqueous solutions of Au-1 and Au-2 NCs, the NCs/polymers composites were prepared first. Generally, the PVA aqueous solution (10 wt.%) was synthesized by directly heating the solution at 80 °C in an oil bath and stirring at 1500 rpm. After the polymer was completely dissolved and naturally cooled down to room temperature, 1.5 mL of as-synthesized PVA aqueous solution was mixed with 0.5 mL of Au-2 NCs and stirred at 600 rpm for 30 min. The obtained NCs/polymers composites were subjected to centrifugation at 450 × g for 3 min. Then, the Au-1/PVA and Au-2/PVA composites were spin-coated onto the glass substrate (2 cm × 2 cm) at 3000 rpm for 1 min. The thin film of Au-3 NCs was prepared by directly spin coating its toluene solution with the same operating conditions.

### Synthesis of Au-2 NCs paired with selected water-soluble ligands

The water-soluble QACs including TPA, TMA, DDAB, DTAC, STAB, HTAB, TMPA, and TMBA were weighted and directly added into the aqueous solution of Au-2 NCs under 600 rpm stirring and held for 1 h. For the concentration-dependent synthesis of TMPA and TMBA paired gold NCs, the adding concentrations of both cations were fixed at 1.0, 1.5, 2.0, 2.5, and 3.0 mg mL$^{-1}$.

### Characterization

TEM images were recorded on FEI Tecnai G2 F20 microscope oper-ated at 200 kV. MALDI-TOF mass spectra were tested on Brucker

Autoflex speed TOF under a positive linear mode. Trans-2-[3-(4-tert-Butylphenyl)−2-methyl-2-propenyldidene] malononitrile (DCTB) was employed as the matrix for all samples. [1]H- NMR measurements were performed with AS 400 MHz (Q. One Instruments Ltd.) system. $D_2O$ and DMSO-d6 were used as solvents to dissolve ARG, TMPA, TMBA, ATT, TOA, and their corresponding gold NCs for [1]H-NMR measurements, respectively. XPS spectra were collected on Thermo Scientific ESCALAB250 spectrometer. For XPS measurements, the solution of gold NCs were dropwise added on silica substrate (2 mm × 2 mm). UV-vis absorption spectra were recorded on Shimadzu UV-1900i spectrometer. The refractive indexes of serial Au NCs were tested on Abbe WAY-2W refractometer at room temperature. PL and PL excitation spectra were measured on a Hitachi F-4700 spectrometer with excitation and emission slits of 2.5 nm. Before the absolute PLQY measurement of the dilute solution of three gold NCs, their optical density (OD) at 405 nm was fixed at 0.1 by diluting them with water for Au-1 and Au-2 NCs and toluene for Au-3 NCs. The PLQY of all gold NCs were measured and calculated on a FLS1000 spectrofluorometer (Edinburgh Instruments Ltd.) attached with an integrating sphere coating with a $BaSO_4$ layer. A Xe lamp with a fixed emission wavelength at 405 nm for three samples was employed as the excitation source. The intensity factor was set as 100, and the excitation and emission slit were set as 2.2 and 0.22 nm, respectively. The scanning range was fixed from 385 nm to 650 nm, and the data were collected 1 nm per step. For the detailed measurements of the PLQY of Au-1, Au-2, and Au-3 NCs, the pure solvents of water (for Au-1 and Au-2 NCs) and toluene (for Au-3 NCs) were measured first in the integrating sphere as a blank reference. Then, the pure solvent was replaced with pre-adjusted gold NCs solutions, and their emission spectra were directly measured without any changes in the experimental conditions. The calculation of absolute QYs values was conducted on the built-in "Fluoracle" software (version 2.13.2). TRES and fluorescent lifetime measurements were carried out through a time-correlated single-photon counting (TCSPC) method on an Edinburgh FLS1000 spectrofluorometer (upon 405 nm excitation) with a pulsed LED excitation source.

## Femtosecond-TA measurements

Femtosecond-TA spectroscopy was performed on a commercial Ti: Sapphire laser system (Spitfire Spectra-Physics). The laser pulse (-100 fs) in the ultraviolet and near-infrared wavelength was first generated in a 3.5 mJ regenerative amplifier system (Spitfire, Spectra-Physics) and optical parametric amplifier (OPA, TOPAS). A small portion of the laser fundamental was focused into a sapphire plate to produce supercontinuum in the visible range, which overlapped in time and space with the pump. An electronically delayed supercontinuum light source with a sub-nanosecond pulse duration (EOS, Ultrafast Systems) was used as the probe. Multiwavelength transient spectra were recorded using dual spectrometers (signal and reference) equipped with array detectors whose data rates exceed the repetition rate of the laser (1 kHz). Solution samples in 1 mm path length cuvettes were excited by the tunable output of the OPA (pump). For the coherent oscillation measurements of gold NCs, 400 nm excitation was employed while near-band-gap excitation is hard to capture the oscillation signals for present Au-1, Au-2, and Au-3 NCs. It should be noted that the oscillation amplitude in different metal NCs is tightly related to the pumping energy and power, and the composition and measuring concentration of metal NCs. However, only the low-frequency coherent oscillation attributed to the quadrupolar-like mode of metal core was considered in this study, and structural and compositional evidence has proved that the metal cores in serial NCs are nearly identical even though different organic ligands were decorated on the surface of gold NCs. On the bases of this prerequisite, we rationalized the comparison of the acoustic oscillation amplitudes in serial NCs by carefully controlling the experimental conditions

including the same pumping wavelength of 400 nm and pumping power of 650 μJ cm$^{-2}$, and identical concentration (-0.5 mM, Au basis) of three samples. To harvest a fine TA map for coherent oscillation analysis, a step scan with 20 fs per step was conducted in a delayed time range of −0.5–4.5 ps. For the data analysis of all pristine TA data, background subtraction was first implemented with 5 spectra. The wavelength range was cropped from 450 nm to 750 nm to wipe out interferences from spurious signals. Chirp correction was then carried out to calibrate the initial time. Before global fitting, singular value decomposition (SVD) has been performed to evaluate the number of time constants needed. Subsequently, the TA map was globally fitted with a parallel model to give decay-associated spectra (DAS); the change of concentration can be described as:

$$C_i^{para} = \exp^{-t/\tau i} \qquad (1)$$

For the fast Fourier Transform (FFT), the decay traces at the crossing line between GSB and ESA bands (zero position) were extracted because of the minimized electronic decay signals at this position, and thus the acoustic oscillation signals can be easier identified. FFT of these decays was carried out by applying Hanning window function to prevent spectral leakage.

## X-ray absorption measurements

The X-ray absorption finds structure spectra Au L-edge were collected at BL14W1 beamline of Shanghai Synchrotron Radiation Facility (SSRF). The data were collected in a fluorescence mode using a Lytle detector while transmission mode was applied for the corresponding references. The samples were ground and uniformly daubed on the special adhesive tape. The acquired Fourier transformation extended XAFS (EXAFS) data were processed according to the standard procedures using the ATHENA module of Demeter software packages. The EXAFS spectra were obtained by subtracting the post-edge background from the overall absorption and then normalizing with respect to the edge-jump step. Subsequently, the χ(k) data of were Fourier transformed to real (R) space using a Hanning window (dk = 1.0 Å$^{-1}$) to separate the EXAFS contributions from different coordination shells. To obtain the quantitative structural parameters around central atoms, least-squares curve parameter fitting was performed using the ARTEMIS module of Demeter software packages. The following EXAFS equation was used:

$$\chi(k) = \sum_j \frac{N_j S_0^2 F_j(k)}{KR_j^2} \cdot \exp[-2k^2\sigma_j^2] \cdot \exp\left[\frac{-2R_j}{\lambda(k)}\right] \cdot \sin[2kR_j + \phi_j(k)] \quad (2)$$

The theoretical scattering amplitudes, phase shifts, and the photoelectron mean free path for all paths were calculated. Where $S_0^2$ is the amplitude reduction factor, $F_j(k)$ is the effective curved-wave backscattering amplitude, $N_j$ is the number of neighbors in the j$^{th}$ atomic shell, $R_j$ is the distance between the X-ray absorbing central atom and the atoms in the j$^{th}$ atomic shell (back scatterer), λ is the mean free path in Å, $\phi_j(k)$ is the phase shift (including the phase shift for each shell and the total central atom phase shift), $\sigma_j$ is the Debye-Waller parameter of the j$^{th}$ atomic shell (variation of distances around the average $R_j$). The functions $F_j(k)$, λ and $\phi_j(k)$ were calculated with the ab initio code FEFF9. The additional details for EXAFS simulations are given below. All fits were performed in the R space with a k-weight of 2 while phase correction was also applied in the first coordination shell to make R-value close to the physical interatomic distance between the absorber and shell scatterer. The coordination numbers of model samples were fixed as the nominal values. While the $S_0^2$, internal atomic distances R, Debye-Waller factor $\sigma^2$, and the edge-energy shift Δ were allowed to run freely.

## DFT calculations

Density-functional theory (DFT) calculations using the Gaussian 09 program package are performed to obtain the electronic properties of these clusters. Specifically, the Perdew-Burke-Ernzerhof (PBE) functional and the all-electron basis set 6-31 G* for H and S, effective-core basis set LANL2DZ for Au was adopted. In addition, the Polarizable Continuum Model (PCM) using the integral equation formalism variant (IEFPCM) calculations with radii and non-electrostatic terms for Truhlar and coworkers' SMD solvation model were employed to calculate the energy in water solution. Au-3 structure was optimized using a DFT method implemented in Materials Studio Dmol3 program. The choice of different software packages to optimize Au-1, Au-2, and Au-3, and further simulate their corresponding electronic properties is based on the consideration of guaranteeing the accuracy of simulated models but reducing the calculation time at the same time. The generalized gradient approximation with the Perdew-Burke-Ernzerhof (PBE) functional and the double numeric polarized (DNP) basis set were adopted. The water solution using methods based on the conductor-like screening model (COSMO) was adopted.

## Decay rate calculation

All the fluorescent lifetime collected through TCSPC were subjected to fitting according to the biexponential decay model:

$$I_{(t)} = I_0 + A_1 \exp^{\left(\frac{-t}{\tau_1}\right)} + A_2 \exp^{\left(\frac{-t}{\tau_2}\right)} \tag{3}$$

where $I_{(t)}$ and $I_0$ refer to the PL intensity detected at time $t$ and $O$, $A_1$ and $A_2$ stand for two constants, $\tau_1$ and $\tau_2$ are the corresponding decay components, respectively. The average fluorescent lifetime ($\tau_{ave.}$) thus could be calculated through the following equation:

$$\tau_{ave.} = \frac{A_1 \tau_1^2 + A_2 \tau_2^2}{A_1 \tau_1 + A_2 \tau_2} \tag{4}$$

The radiative lifetime ($\tau_r$) and non-radiative lifetime ($\tau_{nr}$) were calculated through:

$$\tau_r = \frac{\tau_{ave.}}{QY} \tag{5}$$

$$\tau_{nr} = \frac{\tau_{ave.}}{1 - QY} \tag{6}$$

Note that, Eqs. (5, 6) are merely suitable for luminescent materials involved in simple double-energy level systems. In this regard, the absolute QY values of Au-1, Au-2, and Au-3 NCs were measured to be <0.3% (-0.2%), 51.2 ± 2.6%, and 86.0 ± 3.3%, respectively, upon near-band gap excitation (i.e., 490 nm for Au-1 NCs, 500 nm for Au-2, and Au-3 NCs, respectively). In addition, their corresponding fluorescent lifetimes were recorded to be 4.7, 47.8, and 52.7 ns, respectively, excited by a 475 nm (near-band gap excitation) pulsed LED excitation source. These parameters were employed to calculate their corresponding $k_r$ and $k_{nr}$ according to the following two formulas:

$$k_r = \frac{1}{\tau_r} \tag{7}$$

$$k_{nr} = \frac{1}{\tau_{nr}} \tag{8}$$

## Reporting summary

Further information on research design is available in the Nature Portfolio Reporting Summary linked to this article.

## Data availability

The authors declare that the data supporting the findings of this study are available within the paper and its supplementary information files. All relevant data are available from the corresponding author on request. The source data underlying Figs. 1c, 2a–c, 3a, 4g–i are provided as a Source Data file. Source data are provided with this paper.

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

## Acknowledgements

Y.Zho., and J.Z. contributed equally to this work. We thank Prof. M.Z. from the University of Science and Technology of China (USTC) for his significant discussion about the TA analysis. We acknowledge the financial support from the National Natural Science Foundation of China (NSFC) (12174151 to Z.W., 61935009 to Y.Zho., 11974142, and U21A2068 to X.B.).

## Author contributions

Y.Zho., Z.W. and J.X. conceptualized the idea and co-supervised this work. Y.Zha., T.L., M.L. and X.B. performed the synthesis of metal NCs, optical spectra measurements, and serial femtosecond-TA experiments. W.X. carried out the DFT simulation. J.Z. conducted XAS characterization. Y.Zho., J.Z., T.L., W.X., Q.Y., M.L., X.B., Z.W., J.X. and Y.Zha. discussed the experimental data and commented on the original draft. Y.Zho., and Z.W. wrote the manuscript.

## Competing interests

The authors declare no competing interests.
