## [Peer Review File · Nature Communications]

Reviewer comments, first round –

Reviewer #1 (Remarks to the Author):

Authors report the strong enhancement of the photoluminescence quantum yields of an atomically defined Au cluster (Au₁₀) induced by the rigidification of the Au₇ kernel obtained by coating with different organic structures (thiothymine, arginine, tetraoctylammonium). Reduction of the amplitude of kernel mechanical vibrations strongly reduces non-radiative energy decay, thus inducing a large photoluminescence with longer time decays.

Enhancement of photoluminescence yield is an active field of research with many potential applications. The strategy used here, consisting in achieving strong emission by tailoring non-radiative decay through reduction of molecular motions is a promising and established approach.

This work is long and accurate. Many experimental and theoretical techniques are employed for investigation and full characterization of the effects involved (optical techniques, both steady state and time-resolved, X-ray, electron microscopy, NMR, mass spectroscopy, ...). Most of the conclusions are supported by scientific evidence. Nonetheless, I have some remarks that I am listing here.

1. The authors insist on a 7-fold reduction in the amplitude of low-frequency coherent oscillation of metal kernel. This quantitative conclusion comes from the reduction of the amplitude of the oscillation in femtosecond time-resolved signal. Now, the visibility of coherent oscillations depends on many different physical parameters, affecting both excitation and detection. Oscillation amplitude can rarely be compared among different samples (each one having a different excitation spectrum, composition, concentration, ...). A smaller amplitude in optical oscillations in one sample does not necessarily mean a smaller atom vibration displacement. This point should be clarified. Also, a longer coherent oscillation seems to overlap the one at 0.6 ps (Figure 4d-f), where does this longer vibrational mode come from?

2. While characterization of the samples is rigorous, explanation of the main mechanism (page 12) is speculative. It is based on the qualitative consideration that a reduced kernel motion is at the origin of both reduced electron internal conversion processes (few ps time constant) and reduced electron decay to the lowest excited state through structural relaxation (tens of ps). This in turns is supposed to enhance the radiative time constant (tens of ns), thus the emission strength. Authors should better consolidate this interpretation, with longer discussions and additional references.

3. All data are given with no discussion about their accuracy and uncertainty. No error bars are discussed all through the manuscript. Even the accuracy of the "90.3%" value (also included in the paper title) is not discussed.

4. The main claim of the paper (enhancement increase from 0.2 % to 59.6% and 90.3%) is reported on a graph (Figure 1a) where the PL intensity is in arbitrary units. Authors should better describe the technique used for absolute measurement (are arbitrary normalizations needed?) of PL yield.

5. Although the paper is written with accuracy, some English sentences still need to be verified (wrong prepositions, missing punctuations, some misspelling, ...).

Reviewer #2 (Remarks to the Author):

The work deals with an important topic, the luminescence of metal clusters. A common strategy to increase the luminescence quantum yield is to increase the rigidity of the ligand shell. Here the authors claim that for the systems studied it is the suppression of kernel oscillations that leads to

high quantum yield. I am not at all convinced for the reasons outlined below and do not recommend publication at this point. In general, several claims are not or not sufficiently supported by the experimental data.

Specific comments:

- The fits to the TA traces cannot (to say the least) be correct (most of the ones in the SI, but also the ones in Fig. 3). It seems like the authors have fixed the long times to the mean-lifetime result of the TCSPC (if the TCSPC is biexponential with 2 nanosecond lifetimes, the TA should be so too). E.g. the long lifetime in Fig. S28 is always the same...this can hardly be true.
- "...can be assigned to the internal conversion (IC) process of hot electrons from S_n to S₁ state." I am unsure about that. The prominent ESA band between 510 and 550 at short times could also be the hot S₁ state and it's cooling. If IC would really take that long, one should also see SE from the higher excited states. I rather think it is ultrafast IC and then hot S₁ which is cooling.
- The calculations of k_r and k_{nr} in Table S2 seem incorrect. Note also, that not only does k_{nr} decrease (as they postulate), but also does k_r increase when going from Au-1 to Au-3. Why is k_r increasing by so much? Is this reflected in the absorption counterpart (i.e. ϵ)?
- The fluorescence lifetime is NOT the radiation lifetime (as the authors state multiple times in the MS).
- Where is the stimulated emission in the TA spectra? In principle the SE should be as prominent as the GSB (in the end it is the same transition).
- It also seems that the TA-spectra are not centered around $t=0$, but rather shifted to some positive time-delay (see e.g Fig.3).
- What do the authors mean by "the time-constant of electron-phonon coupling"?
- The authors should use a colormap, which allows for knowing where $DA = 0$ is. It seems like the scale is always changing.
- "...the LUMO → HUMO transition that is dominated by the metal kernel." I am not sure the authors can show that convincingly. What specifically is the evidence for this assignment? In particular, changing the ligand layers does change the absorption spectrum (quite significantly), so that would rather imply the HOMO-LUMO has some ligand contribution.
- Taking the GSB/ESA zero crossing line for the wave-packet identification is a bit strange (also, what about the SE, which should be exactly there?). Usually people opt for looking at the oscillations either explicitly in the GSB, SE or ESA, to be able to assign where the wavepacket is located (in the GS or ES).
- Like always - the entire photophysics remain mysterious: Why are the emission decays multiexponential? (interestingly this biexponentiality in TCSPC does not show up in the TA...). Also, the absorption spectrum of Au-1 is blueshifted with respect to Au-2 and -3, but the emission spectra are almost congruent. Why is this?
- Figure S12: Why is there a calculated band for Au-2 at almost 600nm? (in contrast to: "...can well reproduce the experimental peaks"). Fitting single Gaussians (for the individual electronic transitions) to the absorption spectrum vs. wavelength makes no sense whatsoever. The authors use this to propose the existence of a band at 457nm. This is not convincing.
- It is important to know how the QYs with the integrating sphere were measured. Very often such a measurement requires to have the reference (solvent only) being measured using ND-filters, so as to remain in the linear regime of the detector (not doing so could give unreasonably large QYs). Nothing is mentioned here. The authors should explain the exact procedure and show the raw data.
- Figure S18: Are these not species associated spectra (SAS) instead of DAS?
- The chemical formula provided by the authors (page 5) for the three samples is an approximation at best. I guess in solution the situation is quite dynamic. Maybe NMR DOSY experiments could give some more information.
- Also, the writing (language) needs to be improved substantially.

In conclusion, the work does not meet the high standards normally required for publication in Nature Communications.

Reviewer #3 (Remarks to the Author):

The manuscript by Zhong et. al. entitled "Suppression of Kernel Oscillation Boosts 90.3% Absolute Photoluminescence Quantum Yields of Metal Nanoclusters", studied the efficient suppression of kernel oscillation achieving remarkable enhancement in emission intensity, by rigidifying the surface of metal NCs and propagating as-developed strains into the metal core. They used a layer-by-layer triple-ligands surface engineering to make the solution-phase Au NCs have strong metal core-dictated fluorescence, and the absolute quantum yield reached a record of 90.3%. They used ultrafast transient absorption (TA) to map out the photophysics of three Au nanoclusters. However, there is no crystal data in this work, and the current data cannot prove the formula of the samples. Moreover, the explanations of the TA data are confused, and the conclusions cannot be backed up by the current data. The following comments should be taken into consideration before the manuscript can be accepted.

Comments:

1. How did the authors obtain the accurate structures of Au NCs (Figure 1c) without knowing the crystal data? If there is no solid evidence, the crystal structure of nanocluster cannot be drawn as shown in Figure 1.
2. The authors claimed that ATT and ARG are bound by hydrogen bond on page 4. How did the authors determine the bonding between ligand layers?
3. In Figure S11, the MALDI-TOF mass spectra of three nanoclusters cannot prove the purity and the formula of the sample. ESI should be performed to further prove the purity and to determine the formula of three nanoclusters in the manuscript.
4. The emission band observed in three NCs are sharp and the stokes shift is very small, which is unusual for gold clusters. Is there an explanation for this?
5. Suppressing the structural vibration will prevent nonradiative relaxation, but suppression of the coherent oscillations of metal kernel cannot suppress the non-radiative processes. How did the authors understand the differences between "coherent vibrations" and "structural vibrations"?
6. The PLQY of Au-2 and Au-3 NCs are 59.6% and 90.3%, respectively. Anyway, if there is any strong emission observed, strong SE should be observed in the fs-TA spectra. I check the fs-TA data in Figure 3, unfortunately, why is there no any stimulated emission (SE, 500-600 nm) signals observed on the TA map. Also, I check the fs-TA map in Figure 3, it seems that the authors should recheck the chirp correction (it seems that the chirp correction is not fully corrected or completed), that the chirp corrections for fs-TA spectra should be rechecked carefully. By the way, I also suggest that the authors should redraw all the fs-TA map with time-delay related spectra line-by-line, because map cannot judge the spectral quality. All the fitted data should give error bar or error region, and necessary residual and χ^2 should also be given to judge the fitting quality.
7. The authors mentioned that "the amplitude of ps-level components gradually decreased in corresponding DAS spectra (Figure S18), but it is strengthened for the last ns-level component. Namely, the non-radiative pathways are significantly inhibited and the saved energy in excited electrons is transferred and contributed to the radiative transition, leading to the pronouncedly lengthened radiation lifetime." The statement in these sentences makes no sense. First, the DAS spectra cannot stand for the spectra of transient species, thus the opinion of inhibiting nonradiative pathways cannot be obtained. Second, the "amplitude" in the above sentence are in the GSB region, which cannot be used to compare the "radiative transition".
8. In page 10, the authors state "Under near-HOMO-LUMO gap pump, only the two ps-level time components are fitted out." However, time constants extracted from the global fitting of corresponding TA maps of Au-1, Au-2, and Au-3 NCs excited at near-HOMO-LUMO gap pump are ps and ns scales, respectively. Please recheck this sentence.
9. Why the lack of ultrafast decay component under near-bandgap excitation means the absence of core-shell relaxation? From Figure S27-S29 and Table S7, it seems that the second process disappeared, which means the ultrafast decay component didn't disappear.
10. In Figure 4, the FFT should be carried out after subtracting the population dynamics, refer to J. Phys. Chem. Lett. 2018, 9, 7085–7089, ACS Nano, 2010, 4, 3406-2412. Because the high amplitude may be caused by excited-state population dynamics.

Reviewer #4 (Remarks to the Author):

Briefly, the work is interesting but preliminary. The importance of the high QY in a gold nanocluster and the ways in which they can be achieved is of interest for Nature Communications, but the current manuscript draft and structural hypothesis needs to be improved. The authors either need to provide better support for their proposed structures or completely remove the current proposed structures.

More detailed comments:

p. 4 "Stocks" shift should be "Stokes"

Do the authors have a crystal structure for the Au₁₀(SR)₆ structure? The top and bottom faces appear very unprotected. The authors state "we further reconstruct the total structure of serial Au NCs based on grand unified model for describing the growth of the gold cores and ring model for describing interfacial interactions between the protection motifs and gold cores in thiolate-protected gold nanoclusters.[45] As depicted in Figure 1c, the resultant refined total structure ..." Again, it is not clear how they get this unless they are hypothesizing the structure based on other structures for other NC stoichiometries. The authors are doing some theory, or so they say in the SI (but it is not discussed at all in the main text?). Have they considered assessing isomer energies? Furthermore, the scheme presented in Figure 1 seems rather simplistic. The TOA ligands are unlikely to be planar as shown. They should be quite tetrahedral around the N atom, and the ligand is likely to 'clump'.

Why are two different levels of theory used (for Au-1 and Au-2 vs. for Au-3)? Why not reoptimize Au-1 and Au-2 with the second level of theory if the first is too demanding for the larger system? Actually, it does not specify in the SI if the two structures are optimized or not (it states that Au-3 is optimized). Consistency is desirable for comparing the systems.

The authors might consider simulating EXAFS spectra for their proposed structure/DFT structure in order to see if it matches experiment. A larger theory component to this work could help.

For the NMR signals in Figure S10 parts b, d, f, please provide the integrations. Not very much ATT appears in Au-3 NCs according to part f, as far as it appears to me. Also, ARG broadens more than ATT seems to, which may indicate that it is closely associated with gold atoms (not far away as suggested by the Scheme).

Could the authors provide a reference for "For molecular-like metal NCs, a short impulsive laser can excite the coherent acoustic vibrations of metal kernel for a few to tens of ps. " ? Is it the same as the following sentence? Could the authors also provide their coherent oscillation in wavenumbers and compare with other known values for acoustic vibrations? The authors say that other systems are "similar" regardless of size (Refs. 58-60), but do not provide numbers for comparison. It could be good to include a table of these values here?

The structures in Figure 5 do not look similar to the structures in the first figure?

I think the authors are overestimating the time required for structural relaxation. 95.4 ps seems very high.

The figures shown in Figure 5 a,b,c are essentially the same except for the times listed. Perhaps this figure could be changed to emphasize the differences between the systems?

The authors could improve this section to better explain what they are doing, what they learned, etc. for a nonspecialist reader.

Some grammar/word choices could be improved. For example, the authors use the following in the abstract that need to be restated for grammar/clarity:

"restriction of structural dynamic"

"yet remains challenging in consideration of the metal kernel as its low accessibility "

"resultant of marginal change"

etc.

In general, the authors are trying to be too "fancy" with their word choices in many sentences, but it is making the paper very challenging to read. Simplifying the sentences could improve readability. For example,

"This scenario therewith impedes the nonradiative internal conversion and structural relaxation of electron dynamics [essentially, nonradiative internal conversion], rendering the Au NCs with strong emission. The presented study exemplifies the crucial linkage between surface chemistry and core-state emission of metal NCs and deepens the concept of the metallic molecule for metal NCs [restate] by regulating the interior metal kernel involved motion [aka core vibrations] toward a bright emission. "

"exterior interference immunity to surroundings" [?]

"allow metal NCs more dynamic" Note: "dynamic" is not used properly throughout the paper. I'm not sure in these places if "dynamics", "vibrational motion", or other word choices would be best. Moreover, I really don't understand what the authors mean by "accessing" the metal kernel. If they were talking about catalysis, I would understand them referring to the ability of a small molecule to bind to the metal kernel. Here, they say "we report accessing the metal kernel of metal NCs to suppress their low-frequency acoustic vibration".

"negative-liner"

There are quite a few fragment sentences as the paper goes on.

Replies to reviewers' comments and descriptions of revisions made

Comments by Reviewer #1:

Authors report the strong enhancement of the photoluminescence quantum yields of an atomically defined Au cluster (Au₁₀) induced by the rigidification of the Au₇ kernel obtained by coating with different organic structures (thiothymine, arginine, tetraoctylammonium). Reduction of the amplitude of kernel mechanical vibrations strongly reduces non-radiative energy decay, thus inducing a large photoluminescence with longer time decays.

Enhancement of photoluminescence yield is an active field of research with many potential applications. The strategy used here, consisting in achieving strong emission by tailoring non-radiative decay through reduction of molecular motions is a promising and established approach.

This work is long and accurate. Many experimental and theoretical techniques are employed for investigation and full characterization of the effects involved (optical techniques, both steady state and time-resolved, X-ray, electron microscopy, NMR, mass spectroscopy, ...). Most of the conclusions are supported by scientific evidence. Nonetheless, I have some remarks that I am listing here.

Reply: We sincerely appreciate the reviewer's positive acknowledgment of the novelty and significance of our study. Indeed, the presented strategy of suppressing coherent vibration through indirectly increasing the external surface rigidity of gold NCs to achieve record absolute quantum yields (QYs) is distinctly different from previously documented strategies, such as increasing electron-donating capability of shell, improving the rigidity of shell, conferring ligands with multiple interaction sites, and diverse approaches of solvent-/cation-/crystallization-/scaffold confinement-induced aggregation-induced emission (AIE). Therefore, this mechanism provides a new understanding to manipulate the core-state emission of metal NCs, and we believe that will attract fundamental interest from heterogeneous readers of the community of *Nature Communications*, stimulating more research activities in a diverse field of luminescent materials, cluster chemistry, colloid and interface chemistry, and optical physics. We would also like to thank the reviewer for his/her inspiring and constructive comments/suggestions, which have been taken into careful consideration in this revision. Please see below for a point-to-point response to the reviewer's specific comments/suggestions

Specific comments:

1. The authors insist on a 7-fold reduction in the amplitude of low-frequency coherent oscillation of metal kernel. This quantitative conclusion comes from the reduction of the amplitude of the oscillation in femtosecond time-resolved signal. Now, the visibility

of coherent oscillations depends on many different physical parameters, affecting both excitation and detection. Oscillation amplitude can rarely be compared among different samples (each one having a different excitation spectrum, composition, concentration, ...). A smaller amplitude in optical oscillations in one sample does not necessarily mean a smaller atom vibration displacement. This point should be clarified. Also, a longer coherent oscillation seems to overlap the one at 0.6 ps (Figure 4d-f), where does this longer vibrational mode come from?

Reply: Thank you for this insightful comment. We are in complete agreement with the reviewer that the optical oscillation amplitude greatly depends on the composition and concentration of samples, and the excitation energy and power of apparatus, and therefore a smaller amplitude in optical oscillations in one sample does not necessarily mean a smaller atom vibration displacement. As for this problem, we are careful to deal with it. **First**, it was experimentally and theoretically demonstrated that the acoustic oscillations with different frequencies in the TA spectra of metal NCs originate from different structural components (*Nano Lett.* **2018**, 18, 6842; *Angew. Chem. Int. Ed.* **2017**, 56, 16257). The high-frequency oscillations (periods typically below 150 fs) are usually related to the acoustic vibration of Au-S bonds at the interface of staple motifs or the vibration inside the ligand shell (*J. Phys. Chem. C* **2014**, 118, 9604; *J. Phys. Chem. C* **2016**, 120, 25378; *ACS Nano* **2017**, 11, 11872; *J. Am. Chem. Soc.* **2011**, 133, 3752), whereas the low-frequency oscillations mostly involve in the atom displacement of a metal core (*J. Phys. Chem. C* **2010**, 114, 19935; *J. Phys. Chem. A* **2013**, 117, 10294; *J. Phys. Chem. C* **2017**, 121, 10686; *J. Phys. Chem. C* **2015**, 119, 18790). In our study, we focus on investigating the core-state emission of gold NCs, whose structural origin stems from the metal core instead of the interface/surface, relating to the low-frequency acoustic oscillations. The elaborate structural and compositional characterizations of Au-1, Au-2, and Au-3 NCs, including transmission electron microscopy (TEM), matrix-assisted laser desorption ionization mass spectra (MALDI-MS), X-ray photoelectron spectroscopy (XPS), and X-ray absorption near edge structure (XANES) spectra, have demonstrated that their metal cores are nearly identical even though different organic ligands were coated on the NCs. Therefore, Au-1, Au-2, and Au-3 can be regarded as the same species of NCs (i.e., the same metal kernel) to some extent when their metal cores are exclusively considered. **Second**, we also cautiously controlled the experimental conditions to collect the optical signals of acoustic oscillation in the TA spectra of three gold NCs. The same pumping wavelength of 400 nm and pumping power of 650 $\mu\text{J}/\text{cm}^2$ were employed. Moreover, the identical concentration (~ 0.5 mM, Au basis) of three gold NCs was adjusted before conducting their corresponding TA-based coherent oscillation measurements. Therefore, the external factors that affect the optical oscillation amplitude in different gold NCs were furthest minimized. Under these prerequisites, we believe the resultant optical oscillation amplitudes in Au-1, Au-2, and Au-3 NCs can be qualitatively compared to reflect the discrepancy in their core-atom displacement.

In addition, as suggested by the reviewer, we also notice that the quantitatively

description of a ~ 7 -fold reduction of kernel variation based on the observed ~ 7 -fold reduction in the optical oscillation amplitude is not rigorous in our original manuscript. In the revised manuscript, we have clarified this point and replaced the related statement with a more qualitative description of “remarkable reduction”.

In our original manuscript, we choose the GSB/ESA zero crossing line for the wavepacket identification due to the minimized electron dynamic and maximized phonon dynamic at this position. However, the slight electron decay signals still disturb the profile of acoustic oscillations. Combining with the suggestion provided by other reviewers, we extracted pure acoustic oscillations by subtracting the electron dynamic signals in our revised manuscript. As shown in Figure R1, only one FFT peak with a similar vibrational frequency was found for three gold NCs, indicating there are no additional vibrational modes. Accordingly, we have provided more detailed discussions on the coherent oscillations of Au-1, Au-2, and Au-3 NCs based on our short-range TA data and corresponding calculations.

Figure R1. The corresponding FFT results of acoustic oscillations in (a) Au-1, (b) Au-2, and (c) Au-3 NCs by plotting the amplitude as a function of frequency; the pentagram patterns denote the peak of FFT amplitude.

Revisions:

Page 12, Figure 4:

The as-extracted acoustic oscillations were plotted as insets in Figure 4d-f. Figure 4g-i have been re-drawn based on the pure acoustic oscillations in Au-1, Au-2, and Au-3 NCs, and their corresponding FFT. The oscillation frequency in wavenumbers was added for comparison with other metal NCs.

Page 13, Line 9-10:

“The pure acoustic oscillations (the insets in Fig. 4d-f) were extracted by subtracting the electron dynamics using exponential decay⁵⁹⁻⁶¹.”

Page 13, Line 14-19:

“As shown in Fig. 4g-i, serial Au NCs exhibit similar vibration frequency of 59.0 (1.77 THz), 58.7 (1.76 THz), and 57.3 cm^{-1} (1.72 THz) but a remarkable reduction

in the variation amplitude. The relatively low-frequency vibration in serial NCs indicates the oscillations are categorized into acoustic vibration and are better ascribed to the quadrupolar-like vibration mode which is induced by the periodical expansions and contractions of atoms along one direction associated with out-of-phase oscillations in the perpendicular plane^{62,63}.”

Page 23, Line 26-27:

Additional reference has been included to support our statements:

63. Zhang, W. et al. Coherent vibrational dynamics of Au₁₄₄(SR)₆₀ nanoclusters. *Chem. Sci.* **13**, 8124–8130 (2022).

Supplementary Information (SI), Page 3, Line 9-18:

“It should be noted that the oscillation amplitude in different metal NCs is tightly related to the pumping energy and power, and the composition and measuring concentration of metal NCs. However, only the low-frequency coherent oscillation attributed to the quadrupolar-like mode of metal core was considered in this study, and structural and compositional evidence has proved that the metal cores in serial NCs are nearly identical even though different organic ligands were decorated on the surface of gold NCs. On the bases of this prerequisite, we rationalized the comparison of the acoustic oscillation amplitudes in serial NCs by carefully controlling the experimental conditions including the same pumping wavelength of 400 nm and pumping power of 650 $\mu\text{J}/\text{cm}^2$, and identical concentration (~ 0.5 mM, Au basis) of three samples.”

2. *While characterization of the samples is rigorous, explanation of the main mechanism (page 12) is speculative. It is based on the qualitative consideration that a reduced kernel motion is at the origin of both reduced electron internal conversion processes (few ps time constant) and reduced electron decay to the lowest excited state through structural relaxation (tens of ps). This in turns is supposed to enhance the radiative time constant (tens of ns), thus the emission strength. Authors should better consolidate this interpretation, with longer discussions and additional references.*

Reply: We are grateful to this reviewer for his/her professional efforts in reviewing our manuscript. As per your nice suggestion, we have consolidated relevant interpretations and provided more discussions with necessary references to support our statement in the revised manuscript accordingly.

We should also acknowledge this reviewer and other reviewers’ constructive comments/suggestions, which have encouraged us to refit the TA kinetics and replot Figure 5. As shown in Figure R2, the schematical illustrations of Au-1, Au-2, and Au-3 NCs with strong, medium, and weak coherent vibration of central gold atoms were added. Their corresponding QYs and energy diagrams were also provided to correlate the coherent vibration and PL properties in three Au NCs. Indeed, our statement of enhanced PL properties in Au-2 and Au-3 NCs is based on the consideration of the

declined non-radiative decaying components. The triple-ligands layer-by-layer decorations of ARG and TOA on the surface of Au-1 NCs enable the indirect delivery of the confinement effect from the surface into the gold core, leading to significantly suppressed coherent vibration in Au-2 and especially Au-3 NCs. As a result, the non-radiative core-directed vibrational component increased from 2.2, 2.9, to 4.1 ps in subsequence of Au-1, Au-2, and Au-3 NCs. The suppressed non-radiative relaxation channels in turn boost the core-state green fluorescence, which is clearly evidenced by the promoted fluorescent lifetimes from 3.5 ns to 43.8 ns to 61.0 ns in Au-1, Au-2, and Au-3 NCs, respectively. Reasonably, the absolute QYs was enhanced from < 0.3% (~0.2%) to 59.6% from Au-1 to Au-2 NCs, and a record high value of 90.3% was achieved in terminal Au-3 NCs. An additional reference reported by Jin et al. (*Chem. Sci.* **2017**, 8, 2581) was cited to support our statement in the revised manuscript. Therefore, we believe that the scientific content and readability of our accordingly proposed mechanism are improved, will be attractive, and can be easier identified for the heterogeneous readers of *Nature Communications*.

Figure R2. Schematic illustration of the excited-state dynamics and coherent vibration of Au-1, Au-2, and Au-3 NCs. The red, orange, and gray solid curves in the energy diagrams stand for internal conversion, core-directed, and core-ligand-related structural vibration, respectively. The green solid lines with outer glow denote the fluorescent relaxation. The symbols of S_n , S_1 , and S_0 are the higher singlet state, the lowest singlet state, and the ground state, respectively. The purple, orange, and red circles at the bottom plane stand for layer-by-layer coating of ATT, ARG, and TOA ligands, and the black circles refer to the central gold atoms in three Au NCs.

Revisions:

Page 14, Figure 5:

Figure R2 has been included as Figure 5 in the revised manuscript.

Page 14, Line 9 to Page 15, Line 3:

“Based on the above results obtained from ultrafast excited-state dynamics and optical coherent oscillations, now we are ready to construct a schematic illustration (Fig. 5) to correlate the coherent vibration and PL properties in three Au NCs. In a general model, the external excitation can pump electrons into a high-energy singlet state of S_n . Then, the excited electrons experience ultrafast IC relaxation within several hundreds of femtoseconds to cool into the S_1 state. The singlet populations continue to non-radiatively relax to the S_1 state with the lowest energy through the multiple core-directed and core-ligand-directed structural vibration in few-picosecond and dozens of picosecond time scales, respectively. Nevertheless, the core-directed structural vibration gives dominating non-radiative channels for the contribution of hot electrons in the S_1 state with respect to the core-ligand-directed structural vibration. Subsequently, the excited electrons localized in the gold core return to the S_0 ground state by radiatively emitting the ns -level core-state green fluorescence. Note that there is no additional ultrafast core-shell electronic relaxation in the excited-state dynamics, which is understandable because of the lack of μs -level surface-state phosphorescence in three Au NCs. The triple-ligands layer-by-layer decorations of ARG and TOA on the surface of Au-1 NCs enable the indirect delivery of the confinement effect from the surface into the gold core, leading to significantly suppressed coherent oscillation in Au-2 and especially Au-3 NCs. As a result, the suppressed non-radiative relaxation channels in turn boost the core-state green fluorescence⁶⁷, Thus, the absolute QYs are enhanced from < 0.3% to 59.6% from Au-1 to Au-2 NCs, and a record high value of 90.3% is achieved in the terminal Au-3 NCs.”

Page 24, Line 1-2:

Additional reference has been added to support our statement:

67. Kang, X. et al. The tetrahedral structure and luminescence properties of bi-metallic $Pt_1Ag_{28}(SR)_{18}(PPh_3)_4$ nanocluster. *Chem. Sci.* **8**, 2581–2587 (2017).

3. All data are given with no discussion about their accuracy and uncertainty. No error bars are discussed all through the manuscript. Even the accuracy of the “90.3%” value (also included in the paper title) is not discussed.

Reply: Thank you for this insightful comment. The error regions/bars have been calculated and added to improve the preciseness of our experimental data. For example, the statistics of recorded PLQY values of serial Au NCs (i.e., Au-1, Au-2, and Au-3, as shown in Figure R3) were given accordingly in the revised supplementary file.

Figure R3. The statistics of the recorded PLQY values of < 0.3% for Au-1 NCs, 59.6 ± 2.8% for Au-2 NCs, and 90.3 ± 3.5% for Au-3 NCs, respectively.

Revisions:

Page 4, Line 14, 24, 28, 30, Page 15, Line 11, 12, SI, Page 43, 44, 58, Supplementary Figure 37, 38, Supplementary Table 2:

The error regions for the recorded PLQY values of all gold NCs have been calculated and added accordingly in the revised manuscript and supplementary file.

SI, Page 57, 62, 63, Supplementary Table 1, 6, 7:

The error regions for the fitted lifetimes of all gold NCs have been calculated and given accordingly in the revised supplementary file.

SI, Page 7, Supplementary Figure 1:

Figure R3 has been included as Supplementary Figure 1 in the revised supplementary file.

SI, Page 45, Supplementary Figure 39:

The statistics of the recorded PLQY values of Au-2-TMPA and Au-2-TMBA NCs were plotted in Supplementary Figure 39 in the supplementary file.

4. The main claim of the paper (enhancement increase from 0.2 % to 59.6% and 90.3%) is reported on a graph (Figure 1a) where the PL intensity is in arbitrary units. Authors should better describe the technique used for absolute measurement (are arbitrary normalizations needed?) of PL yield.

Reply: Thank you for this insightful comment. We are sorry for this typo. It should be mentioned that we only normalize the absorption intensity of Au-1, Au-2, and Au-3 NCs at 300 nm to highlight the slight difference in absorption peaks in their absorption spectra. However, we did not attempt to normalize their PL spectra. Moreover, the values of absolute QYs of Au-1, Au-2, and Au-3 NCs were directly measured on a FLS1000 spectrofluorometer (Edinburgh Instruments Ltd.) attached with an integrating sphere and BaSO₄ layer as a reference. The PL spectra illustrated in Figure 1 were

measured on a Hitachi F-4700 spectrometer, and have no connection to the displayed QYs values. We have added an additional description for the measurements of absolute QYs in the supplementary file. Accordingly, the unit of PL intensity in Figure 1 was corrected to be “counts” in this revision. Thank you. We have also re-examined this typo in the main text and supplementary file for a correct and accurate presentation.

Revisions:

Page 5, Figure 1:

The unit of the right y-axis in Figure 1b has been corrected as “counts”

SI, Page 8, 9, 11, 43, 44, Supplementary Figure 2, 3, 5, 37, 38:

The y-axis title in the PL spectra has been corrected as “PL intensity (counts)”

SI, Page 2, Line 15-29:

“Before the absolute PL quantum yields (PLQY) measurement of the dilute solution of three gold NCs, their optical density (OD) at 405 nm was fixed at 0.1 by diluting them with water for Au-1 and Au-2 NCs and toluene for Au-3 NCs. The PLQY of all gold NCs were measured and calculated on a FLS1000 spectrofluorometer (Edinburgh Instruments Ltd.) attached with an integrating sphere coating with a BaSO₄ layer. A Xe lamp with a fixed emission wavelength at 405 nm for three samples was employed as the excitation source. The intensity factor was set as 100, and the excitation and emission slit was set as 2.2 and 0.22 nm, respectively. The scanning range was fixed from 385 nm to 650 nm, and the data were collected 1 nm per step. For the detailed measurements of the PLQY of Au-1, Au-2, and Au-3 NCs, the pure solvents of water (for Au-1 and Au-2 NCs) and toluene (for Au-3 NCs) were measured first in the integrating sphere as the blank reference. Then, the pure solvent was replaced with pre-adjusted gold NCs solutions, and their emission spectra were directly measured without any changes in the experimental conditions. The calculation of absolute QYs values was conducted on the built-in “Fluoracle” software (version 2.13.2).”

5. *Although the paper is written with accuracy, some English sentences still need to be verified (wrong prepositions, missing punctuations, some misspelling, ...).*

Reply: Thanks for your valuable and constructive suggestion. The revised manuscript has been carefully polished by a native English speaker to improve its readability. We hope it will meet the standard of *Nature Communications*.

Comments by Reviewer #2:

The work deals with an important topic, the luminescence of metal clusters. A common strategy to increase the luminescence quantum yield is to increase the rigidity of the

ligand shell. Here the authors claim that for the systems studied it is the suppression of kernel oscillations that leads to high quantum yield. I am not at all convinced for the reasons outlined below and do not recommend publication at this point. In general, several claims are not or not sufficiently supported by the experimental data.

Reply: We're glad that the reviewer agrees with the importance of our research topic. Indeed, optimizing the PL properties of metal NCs is vital not only to understanding the structure-property relationship but also to increasing the acceptance of metal NCs as novel luminescent materials for multiple applications (*Chem. Soc. Rev.* **2021**, 50, 2297; *J. Am. Chem. Soc.* **2015**, 137, 12906; *Angew. Chem. Int. Ed.* **2019**, 58, 8139; *Nature Chem.* **2017**, 9, 689; *Nat. Commun.* **2022**, 13, 3381). We might not have articulated well the new insights revealed in the present work in our initial submission. Through the summary below, we hope to convey the key physicochemical insights into the mechanism of suppressing coherent vibration-induced QYs improvement illustrated in this work.

First, it is true that plenty of works have been reported to increase the QYs of metal NCs by increasing the rigidity of surface shell, such as forming condense oligomeric metal(I)-ligand complexes (*Nano Lett.* **2010**, 10, 2568; *Nanoscale* **2014**, 6, 5777 *Nanoscale* **2014**, 6, 157; *Chem. Commun.* **2015**, 51, 15165), surface shell engineering (*J. Am. Chem. Soc.* **2015**, 137, 8244; *Mater. Chem. Front.* **2018**, 2, 923; *Chem. Mater.* **2017**, 29, 1362), and solvent-/cation-/crystallization-/scaffold confinement-induced AIE enhancement (*J. Am. Chem. Soc.* **2021**, 143, 326; *Nanoscale* **2017**, 9, 15494; *Angew. Chem. Int. Ed.* **2020**, 59, 9934; *J. Am. Chem. Soc.* **2014**, 136, 1246; *Sci. Adv.* **2020**, 6, eaay0107; *Sci. Adv.* **2021**, 7, eabd2091; *Adv. Funct. Mater.* **2015**, 25, 5006). However, it should be pointed out that the PL in most of these reports is phosphorescence, which structurally originates from the surface Au(I)-thiolate staple motifs (surface-state emission) through classical metal-to-ligand charge transfer (MLCT) and/or metal-to-metal-ligand charge transfer (MMLCT) mechanism (*Angew. Chem. Int. Ed.* **2022**, 61, e202205947; *Natl. Sci. Rev.* **2021**, 8, nwaa208). It is well-recognized that increasing the shell rigidity could suppress the non-radiative energy loss and increases the radiative recombination rate, leading to their significantly improved PL properties. Nevertheless, there are fewer reports involving the enhancement of core-state fluorescence which structurally stems from the metal core (*ACS Nano* **2015**, 9, 2328; *Chem. Sci.* **2020**, 11, 8176). In this work, we have remarkably achieved the record QYs (over 90%) of green fluorescence in water-soluble Au NCs through triple-ligand layer-by-layer self-assembly. Most importantly, a new mechanism was proposed for the enhanced core-state fluorescence in serial Au NCs, that is, the reinforced rigidity in the shell can be indirectly transformed into the metal core. The significantly suppressed non-radiative coherent vibration, which was verified by the tremendous amount of TA experiments, enables the promoted fluorescence in serial Au NCs. In this regard, we have proved new insights for manipulating the PL properties of relatively less-involved core-state fluorescence in metal NCs, which is vital to rational customization of functional NCs for applied research.

Second, for core-state fluorescence in metal NCs, it is widely accepted that the excited-state electrons are localized in the metal core (*J. Am. Chem. Soc.* **2019**, 141, 5314; *Chem. Sci.* **2020**, 11, 8176). The QYs of core-state fluorescence is significantly affected by the non-radiative structural relaxation of metal core, such as phonon-assisted structural vibration. The structural vibration can be qualitatively reflected by coherent vibration, which denotes the coherent spatial movement of core atoms. (*Nano Lett.* **2018**, 18, 6842; *ACS Nano* **2021**, 15, 13980; *Proc. Natl. Acad. Sci. U.S.A.* **2017**, 114, E4697; *Angew. Chem. Int. Ed.* **2017**, 56, 16257; *Nanomaterials* **2019**, 9, 933; *J. Phys. Chem. C* **2011**, 115, 6200). In this work, we have demonstrated the coherent vibration in Au-3 NCs has been significantly suppressed by decorating the bulky ligands of ARG and TOA on the surface of Au-1 NCs. The greatly reduced non-radiative structural relaxations in turn promote the boost of QYs of core-state fluorescence in Au-3 NCs. It should be mentioned that similar results were also observed by other researchers. For example, Kang et al. have observed the prominent coherent oscillations in the femtosecond kinetic traces of Pt₁Ag₂₄ NCs. But no such phenomenon was observed for Pt₁Ag₂₈ NCs. They speculated that the stronger phonon behaviors observed in Pt₁Ag₂₄ NCs suggest that more excited-state energy is dissipated into the environment through lattice vibration, which finally leads to weaker luminescence in Pt₁Ag₂₄ NCs (0.1% QYs) than that of Pt₁Ag₂₈ NCs (4.9% QYs, *Chem. Sci.* **2017**, 8, 2581). However, the values of as-reported QYs were not ideal and there is a lack of systematic metal core-directed structural relaxation of electron dynamics. In our work, this is the first report advancing the mechanistic study of the record-high QYs of core-state fluorescence in metal NCs.

Last but not least, we are sorry that we didn't articulate well in some terminology in our first submission. We have refined our writing in this revision to improve the preciseness of our phrasing, neither overstating nor understating our findings. Therefore, we believe that the current revision of our manuscript (with better conveyed scientific contents) will be appealed to the heterogeneous readership of *Nature Communications*.

Specific comments:

1. The fits to the TA traces cannot (to say the least) be correct (most of the ones in the SI, but also the ones in Fig. 3). It seems like the authors have fixed the long times to the mean-lifetime result of the TCSPC (if the TCSPC is biexponential with 2 nanosecond lifetimes, the TA should be so too). E.g. the long lifetime in Fig. S28 is always the same...this can hardly be true.

Reply: Thank you for this insightful comment. We are in complete agreement with the reviewer that the longest decay time in the fits of TA traces cannot be fixed as the mean-lifetime collected by TCSPC. We are sorry for this confusion caused by the less-detailed interpretation of TA data in the original submission. Indeed, due to the limited time window in our TA measurements (~ 8 ns), the fitted-out longest decay time is less accurate. On this basis, we carried out approximate treatment to this parameter and included this decay time as "> 1 ns", which has been extensively employed in other

reports (*Proc. Natl. Acad. Sci. U.S.A.* **2017**, 114, E4697; *J. Am. Chem. Soc.* **2014**, 136, 15559; *J. Am. Chem. Soc.* **2019**, 141, 18715; *J. Phys. Chem. C* **2010**, 114, 19935; *J. Phys. Chem. C* **2011**, 115, 6200; *J. Phys. Chem. A* **2013**, 117, 10294; *J. Phys. Chem. C* **2015**, 119, 20224.)

Revisions:

Page 9, Line 3-6:

“The global fittings generally give four decay components: 320 fs, 2.2 ps, 80.6 ps, and > 1 ns for Au-1 NCs; 240 fs, 2.9 ps, 80.7 ps, and > 1 ns for Au-2 NCs; 118 fs, 4.1 ps, 80.5 ps, and > 1 ns for Au-3 NCs, respectively (Fig. 3e-g and Supplementary Fig. 18).”

Page 9, Line 8-11:

“First, the long-lived (> 1 ns) relaxation component is assigned to the transition from the lowest excited singlet state (S_1) to the ground state (S_0), and this relaxation is not possible to acquire an accurate lifetime due to the limited decay time window (~ 8 ns) in our TA measurements.”

Page 10, Figure 3; SI, Page 40-42, 52, Supplementary Figure 34-36, 46; SI, Page 62, 63, Supplementary Table 6, 7:

The TA traces in these figures have been re-fitted, and the longest time constants in corresponding figures and tables were approximately treated as “> 1 ns”.

2. *“...can be assigned to the internal conversion (IC) process of hot electrons from S_n to S_1 state.” I am unsure about that. The prominent ESA band between 510 and 550 at short times could also be the hot S_1 state and its cooling. If IC would really take that long, one should also see SE from the higher excited states. I rather think it is ultrafast IC and then hot S_1 which is cooling.*

Reply: We thank the reviewer for his/her careful critique, which has spurred us to re-examine our fittings of the TA traces. We agree that the IC time constants of 4.8, 2.8, and 1.5 ps for Au-1, Au-2, and Au-3 NCs in our original manuscript are relatively larger. As suggested by the reviewer, ultrafast decay between 510 and 550 nm should be ascribed to the IC decay and cooling of excited electrons in the hot S_1 state. Accordingly, in our revised manuscript, we have re-performed TA fittings, and four decaying components in total for each Au NCs were fitted out, that is, 320 fs, 2.2 ps, 80.6 ps, and > 1 ns for Au-1 NCs; 240 fs, 2.9 ps, 80.7 ps, and > 1 ns for Au-2 NCs; 118 fs, 4.1 ps, 80.5 ps, and > 1 ns for Au-3 NCs, respectively. On this basis, we rationally assign the time components of several hundreds of femtoseconds (320, 240, and 118 fs) to ultrafast IC decay. Similar time constants for this ultrafast process were also previously reported in other luminescent gold NCs, such as 330 fs in $[Au_{23}(SR)_{16}]^-$ (*J. Am. Chem. Soc.* **2019**, 141, 5314), 150 and 400 fs in the Zn^{2+} -mediated assemblies of $Au_4(SR_{COO}^-)_4$ (*J. Am. Chem. Soc.* **2021**, 143, 326), and 220 fs in our recent

investigated Au₂₂(SG)₁₈ NCs (*Nat. Commun.* **2022**, 13, 3381). In addition, the subsequent few-picosecond (2.2, 2.9, and 4.1 ps) and dozens of picoseconds (80.6, 80.7, and 80.5 ps) decays in all Au NCs should be both contributed to the structural relaxation in the hot S₁ state. Especially, the few-picosecond component is mainly caused by metal core-directed structural vibration, while the dozens of picosecond component originates from metal core-ligand-directed interface vibration (*ACS Nano* **2015**, 9, 2328). Both processes result in the cooling of excited electrons to reach the lowest S₁ state. We have revised the manuscript according to the reviewer's good suggestion.

Revisions:

Page 9, Line 3-8:

“The global fittings generally give four decay components: 320 fs, 2.2 ps, 80.6 ps, and > 1 ns for Au-1 NCs; 240 fs, 2.9 ps, 80.7 ps, and > 1 ns for Au-2 NCs; 118 fs, 4.1 ps, 80.5 ps, and > 1 ns for Au-3 NCs, respectively (Fig. 3e-g and Supplementary Fig. 18). Based on the distinct discrepancy in the time scale, we can rationally assign different time constants to different relaxation processes of excited electrons for decoding the photodynamics in serial Au NCs.”

Page 9, Line 12-18:

“Second, we take the initial several hundreds of femtoseconds (320, 240, and 118 fs) into consideration. The applied excitation energy of 3.10 eV (400 nm) is obviously larger than their HOMO-LUMO gap energies (Supplementary Fig. 19), and thus can give excess excited state energy to pump electrons higher than the S₁ state (e.g., hot S_n state). Therefore, this ultrafast decay component should be better assigned to the internal conversion (IC) of hot electrons from S_n to the S₁ state (S_n → S₁).”

Page 9, Line 18-24:

“Third, the subsequent few-picosecond (2.2, 2.9, and 4.1 ps) and dozens of picoseconds (80.6, 80.7, and 80.5 ps) decays in all Au NCs should be both contributed to the structural relaxation in the sublevels of S₁. Especially, the few-picosecond component is mainly caused by above mentioned core-directed structural vibration, while the dozens of picoseconds component originates from core-ligand-directed interface vibration⁵². Both processes result in the dissipation of the energy of excited electrons through the redistribution of the electron density to cool down them to the lowest S₁ state.”

Page 9, Line 24 to Page 10, Line2:

“The first femtosecond decaying component shortens from 320, 240, to 118 fs from Au-1, Au-2, to Au-3 NCs, indicating the promoted IC relaxation whose occurrence generally related to the energy gap between S_n and S₁, and their electronic state⁵³. Intriguingly, we also find the few-picosecond decaying component associated to core-directed structural vibration experiences a remarkable increase from 2.2, 2.9,

to 4.1 ps. On the contrary, the dozens of picoseconds component related to core-ligand-directed interface vibration is relatively more stable (80.6, 80.7, and 80.5 ps). These results suggest that the core-directed structural vibration is the dominant non-radiative pathway for luminescence quenching, which has been significantly inhibited through the triple-ligands surface engineering strategy. As a result, more population of excited electrons can slower relax to the lowest sublevel of the S₁ state, and subsequently, participate in the radiative process from S₁ to S₀ state.”

3. *The calculations of k_r and k_{nr} in Table S2 seem incorrect. Note also, that not only does k_{nr} decrease (as they postulate), but also does k_r increase when going from Au-1 to Au-3. Why is k_r increasing by so much? Is this reflected in the absorption counterpart (i.e. epsilon)?*

Reply: Thank you for this insightful comment. As described in the section of the calculation of decay rates in the supplementary file, we calculated the radiative decay rate of k_r and non-radiative decay rate of k_{nr} by using the following equations:

$$k_r = \frac{1}{\tau_r} = \frac{QY}{\tau_{ave}} \quad (1)$$

$$k_{nr} = \frac{1}{\tau_{nr}} = \frac{1 - QY}{\tau_{ave}} \quad (2)$$

However, it is true these equations are only suitable for luminescent materials involved in simple double-energy level systems (i.e., S₁ and S₀). While for more complicated multiple-energy level systems (i.e., S_n ... S₁, and S₀), the ignorance of non-radiative relaxation from S_n to S₁ level may bring deviations in the calculation of radiative and non-radiative decay rates. To solve this problem, we have carefully collected the values of absolute QYs of serial Au NCs excited by near-band gap energy (i.e., 490 nm for Au-1 NCs, 500 nm for Au-2, and Au-3 NCs, respectively), which were < 0.3% (~ 0.2%), 51.2 ± 2.6%, and 86.0 ± 3.3% for Au-1, Au-2, and Au-3 NCs, respectively. In addition, their fluorescent lifetimes were also re-measured to be 4.7, 47.8, and 52.7 ns, respectively, by using a 475 nm pulsed LED excitation source. In this regard, the PL process in serial Au NCs can be simplified as a double-energy level system, and therefore the above equations can be employed to calculate their radiative and non-radiative decay rates. As a result, the radiative decay rates are 4.3 × 10⁵, 10.7 × 10⁵, 15.3 × 10⁵ s⁻¹, and the non-radiative decay rates are 212.3 × 10⁶, 10.2 × 10⁶, 3.7 × 10⁶ s⁻¹ for Au-1, Au-2, and Au-3 NCs, respectively. We have provided additional statements and revised these values in the supplementary file accordingly.

We also agree with the reviewer’s notification that not only does k_{nr} decrease but also k_r increased from Au-1 to Au-2 to Au-3 NCs. However, it should be pointed out that k_{nr} experienced a ~57.4-fold reduction (from 212.3 × 10⁶ s⁻¹ to 3.7 × 10⁶ s⁻¹) while k_r only increased for ~3.6 folds (from 4.3 × 10⁵ s⁻¹ to 15.3 × 10⁵ s⁻¹). Therefore, the increment of k_r is slight when compared with the dramatically declined k_{nr} parameter. According to the following calculation equation of QY:

$$QY = \frac{k_r}{k_r + k_{nr}} \quad (3)$$

it is reasonable to conclude that the remarkably enhanced QY values in Au-2 and Au-3 NCs mainly contributed to the suppressed non-radiative decay rate instead of the slightly promoted radiative decay rate.

However, it is interesting why k_r also increased. A vast of theoretical and experimental results in the literature suggest that the value of k_r is tightly related to the inherent local density of photon states (LDOS, *J. Phys. Chem. C* **2007**, 111, 4047), the refractive index of emitters (n_{eff}), and the surrounding media due to the modification of local field (*Nanoscale* **2011**, 3, 3164; *JETP Lett.* **2008**, 88, 12). Especially, for present Au-1, Au-2, and Au-3 NCs, their theoretical values of k_r can be approximatively expressed as follow (*Appl. Phys. Lett.* **2012**, 100, 081104):

$$k_r \propto \frac{f(ED)fn_{\text{eff}}}{\lambda_0^2} \quad (4)$$

where $f(ED)$ is the electric dipole strength, λ_0 is wavelength in vacuum, and f is a factor dependent on n_{eff} . As can be seen, k_r is in a positive proportion to n_{eff} . Additionally, the values of n_{eff} were measured to be 1.3334 for Au-1 NCs, 1.3339 for Au-2 NCs, and 1.4943 for Au-3 NCs, respectively. Therefore, the observed slight increase of k_r from Au-1 to Au-3 NCs may be related to the increased n_{eff} factor.

Revisions:

Page 5, Line 2-9:

“Combining the measurements of the absolute PLQYs and fluorescent lifetimes of serial Au NCs excited by near-band gap energies, we found the radiative decay rate (k_r) is slightly promoted (~ 3.6 -fold) in the sequence of Au-1, Au-2, and Au-3 NCs, which should be related to the increased refractive index (n_{eff}) from 1.3334 in Au-1 NCs, to 1.3339 in Au-2 NCs, and to 1.4943 in Au-3 NCs (Supplementary Table 1)⁴³. Instead, the non-radiative decay rate (k_{nr}) is sharply declined (~ 57.4 -fold) in Au-3 NCs, indicating that the as-adopted triple-ligands surface engineering can greatly suppress the non-radiative relaxation channels.”

Page 22, Line 16-17:

Additional reference has been cited to support our statement:

43. Wang, Y. H. et al. Eu³⁺ doped KYF₄ nanocrystals: synthesis, electronic structure, and optical properties. *Nanoscale* **3**, 3164–3169 (2011).

SI, Page 2, Line 13-14:

“The refractive indexes of serial Au NCs were tested on Abbe WAY-2W refractometer at room temperature.”

SI, Page 5, Line 26 to Page 6, Line 2:

“Note that, equations (5) and (6) are merely suitable for luminescent materials involved in simple double-energy level systems. In this regard, the absolute QY values of Au-1, Au-2, and Au-3 NCs were measured to be $< 0.3\%$ ($\sim 0.2\%$), 51.2

$\pm 2.6\%$, and $86.0 \pm 3.3\%$, respectively, upon near-band gap excitation (i.e., 490 nm for Au-1 NCs, 500 nm for Au-2, and Au-3 NCs, respectively). And their corresponding fluorescent lifetimes were recorded to be 4.7, 47.8, and 52.7 ns, respectively, excited by a 475 nm (near-band gap excitation) pulsed LED excitation source. These parameters were employed to calculate their corresponding radiative decay rate (k_r) and non-radiative decay rate (k_{nr}) according to the following two formulas:"

SI, Page 57, Supplementary Table 1:

The as-calculated radiative decay rates are revised to be 4.3×10^5 , 10.7×10^5 , and $15.3 \times 10^5 \text{ s}^{-1}$, and the non-radiative decay rates are revised to be 212.3×10^6 , 10.2×10^6 , and $3.7 \times 10^6 \text{ s}^{-1}$ for Au-1, Au-2, and Au-3 NCs, respectively.

4. *The fluorescence lifetime is NOT the radiation lifetime (as the authors state multiple times in the MS).*

Reply: Thank you for this insightful comment. We are sorry that we didn't articulate well in some terminology in our first submission. We are clear that the fluorescent lifetime does not stand for a radiative lifetime because it also includes a non-radiative lifetime in it. Accordingly, we have refined our writing in this revised manuscript.

Revisions:

Page 5, Line 1, 3, Page 9, Line 12, Page 14, Line 5, Page 15, Line 13, SI, Page 2, Line 30, Page 5, Line 16, 21, 30, Page 57, Line 1:

The terminology of "radiative lifetime" in the manuscript has been replaced with "fluorescent lifetime".

5. *Where is the stimulated emission in the TA spectra? In principle the SE should be as prominent as the GSB (in the end it is the same transition).*

Reply: This is another very inspiring comment. We agree with the review that SE usually appears as a prominent component of negative GSB signals. This observation has been extensively proved in colloidal quantum dots (*J. Phys. Chem. C* **2020**, 124, 8448), graphene materials (*Nat. Commun.* **2016**, 7, 11010), and organic emitters (*Chem. Phys. Lett.* **2000**, 319, 157), etc. However, as for metal NCs, it is widely reported that no SE signals were acquired in the TA spectra of luminescent metal NCs, even though their QYs were recorded to be extremely high (*Sci. Adv.* **2021**, 7, eabd2091; *J. Phys. Chem. C* **2015**, 119, 18790). We speculate that the undetected SE signals in metal NCs may be caused by their blending with stronger GSB signals (*Nat. Nanotechnol.* **2016**, 11, 872) and/or the competition between SE and ESA processes (*Nature* **2018**, 563, 541).

6. *It also seems that the TA-spectra are not centered around $t=0$, but rather shifted to*

some positive time-delay (see e.g Fig.3).

Reply: We would like to thank the reviewer for his/her careful check on our TA spectra! In this revised manuscript, “Time Zero Correction” was re-applied for all TA maps to make sure TA data are starting at the time zero position. We sincerely appreciate the reviewer’s good suggestion.

Revisions:

**Page 10, Figure 3; Page 12, Figure 4, SI, Page 26, 28, 30, 34, 36, 38, 52, 53
Supplementary Figure 20, 22, 24, 28, 30, 32, 46, 47:**

All TA maps were re-drawn after conducting “Time Zero Correction”.

**Page 10, Figure 3; Page 12, Figure 4, SI, Page 40-42, 52, Supplementary Figure
34-36, 46:**

All the fits of TA traces were re-performed based on the corrected TA maps.

SI, Page 62, 63, Supplementary Table 6, 7:

All the time constants were re-fitted based on the corrected TA maps.

7. What do the authors mean by "the time-constant of electron-phonon coupling"?

Reply: We are sorry that we didn’t articulate well in some terms. What we want to express here should be “the electron-phonon coupling time”, which is a typical process for the photoexcited electrons to release their energy to the lattice in metallic Au NPs (*ACS Nano* **2021**, 15, 13980; *ACS Nano* **2015**, 9, 2328; *Nano Lett.* **2018**, 18, 6842). We have revised the related statement in the manuscript accordingly.

Revisions:

Page 10, Line 12-13:

The sentence of “the time constant of typical electron-phonon coupling” has been revised as “the typical electron-phonon coupling time”.

8. The authors should use a colormap, which allows for knowing where $DA = 0$ is. It seems like the scale is always changing.

Reply: We appreciate the good suggestions of the reviewer. In the revised manuscript, we have labeled the position of $DA = 0$ in the colormap of each TA map to improve their readability.

Revisions:

Page 10, Figure 3; Page 12, Figure 4, SI, Page 26, 28, 30, 34, 36, 38, 52, 53

Supplementary Figure 20, 22, 24, 28, 30, 32, 46, 47:

The intensity position of DA = 0 has been included in the colormap of each TA map.

9. "...the LUMO → HUMO transition that is dominated by the metal kernel." I am not sure the authors can show that convincingly. What specifically is the evidence for this assignment? In particular, changing the ligand layers does change the absorption spectrum (quite significantly), so that would rather imply the HOMO-LUMO has some ligand contribution.

Reply: Thank you for this insightful comment. Given the absorption property of metal NCs, i) it is well-recognized that the UV-vis absorption of metal NCs is dictated closely by the configuration of Au-S frameworks in the thiolate-protected metal NCs (*J. Am. Chem. Soc.* **2008**, 130, 5883; *Nanoscale* **2015**, 7, 1549); ii) different coating ligands can only little shift the peak position but don't change the absorption profile, for example, the characteristic absorption peaks at 365, 482, and 590 nm for Au₂₈(S-C₆H₁₁)₂₀ NCs vs. 355, 460, and 550 nm for Au₂₈(SPh-^tBu)₂₀ NCs (*J. Am. Chem. Soc.* **2016**, 138, 1482).

Considering the change in the absorption spectra of serial NCs in our system, that is "the absorption spectrum of Au-1 is blueshifted with respect to Au-2 and -3", the reason should include the two aspects. On one hand, the absorption peaks of Au-2 and Au-3 NCs are indeed sharper with respect to Au-1 NCs. This can be caused by the stronger vibration of the metal core of Au-1 NCs in water, which results in the weak absorption peak easily overlapping with other absorption peaks, and thus the broader absorption peaks are captured for Au-1 NCs. On the other hand, all the absorption peaks of three Au NCs are laying in the range of 350-550 nm, but the absorption peaks of Au-1 NCs are slightly blue-shifted compared with that of Au-2 and Au-3 NCs. We reasoned that the surface charge re-distribution, which is induced by the diverse supramolecular interactions such as hydrogen bonding between ATT and ARG ligands and electrostatic interactions between ARG and TOA ligands, should account for this shift in absorption spectra. The surface charge re-distribution has been previously demonstrated by Jin's group and our group to induce slight distortion of geometric structure, further leading to the vibration in absorption peaks of gold NCs (*J. Phys. Chem. C* **2008**, 112, 14221; *Chem. Commun.* **2016**, 52, 5234).

10. Taking the GSB/ESA zero crossing line for the wave-packet identification is a bit strange (also, what about the SE, which should be exactly there?). Usually people opt for looking at the oscillations either explicitly in the GSB, SE or ESA, to be able to assign where the wavepacket is located (in the GS or ES).

Reply: Thank you for this insightful comment. In the TA maps of metal NCs, significant overlap between GSB and ESA signals is usually observed (*Science* **2019**, 364, 279; *Proc. Natl. Acad. Sci. U.S.A.* **2017**, 114, E4697; *Angew. Chem. Int. Ed.* **2017**,

56, 16257; *J. Am. Chem. Soc.* **2020**, 142, 18086). This character may be caused by the dense excited energy levels in metal NCs. Therefore, it is challenging to identify the wave-packet at these positions. As for present Au NCs, the reason accounting for our choice at the GSB/ESA zero crossing line is that the electronic decay signals at this position are minimized, and thus it is easier to identify the acoustic oscillation signals. A similar processing method was also reported (*J. Phys. Chem. C* **2011**, 115, 6200). We have included the necessary explanation in the revised supplementary file accordingly.

Revisions:

SI, Page 3, Line 32 to Page 4, Line 1:

“For the fast Fourier Transform (FFT), the decay traces at the crossing line between GSB and ESA bands (zero position) were extracted because of the minimized electronic decay signals at this position, and thus the acoustic oscillation signals can be easier identified.”

11. Like always - the entire photophysics remain mysterious: Why are the emission decays multiexponential? (interestingly this biexponentiality in TCSPC does not show up in the TA...). Also, the absorption spectrum of Au-1 is blueshifted with respect to Au-2 and -3, but the emission spectra are almost congruent. Why is this?

Reply: Thank you for this insightful comment. The biexponential fluorescent decay for serial Au NCs indicates that there are underlying non-radiative relaxation channels in the PL process (otherwise the fluorescent decay should be monoexponential to reflect the sole radiative process). We rationalize these non-radiative relaxations to be defect state (*J. Am. Chem. Soc.* **2017**, 139, 4318) or the dynamic dissociation of outside TOA⁺ ligands or even inner ARG⁻ ligands (*Nat. Commun.* **2020**, 11, 5498; *J. Am. Chem. Soc.* **2018**, 140, 15430; *J. Am. Chem. Soc.* **2012**, 134, 13316). However, due to the limited time window in our TA measurements (~ 8 ns), the biexponential fluorescent lifetimes recorded by TCSPC cannot be completely reflected in TA kinetics.

In addition, as the mentioned explanation above (reply to comment 9) for why “*the absorption spectrum of Au-1 is blueshifted with respect to Au-2 and -3*”, it should be noted that the optical absorption of metal NCs is greatly dictated by the population and packing manner of Au-S frameworks for thiolate-protected metal NCs (*J. Am. Chem. Soc.* **2008**, 130, 18, 5883). For Au-1, Au-2, and Au-3 NCs, their elaborate structural and compositional characterizations, including transmission electron microscopy (TEM), matrix-assisted laser desorption ionization mass spectra (MALDI-MS), X-ray photoelectron spectroscopy (XPS), and X-ray absorption near edge structure (XANES) spectra, have demonstrated that their metal core is nearly identical, leading to the almost congruent emission in three Au NCs. However, we agree with the reviewer that their absorption characteristics are slightly changed. On one hand, the absorption peaks of Au-2 and Au-3 NCs are indeed sharper with respect to Au-1 NCs. This can be caused

by the stronger vibration of the metal core of Au-1 NCs in water, which results in the weak absorption peak easily overlapping with other absorption peaks, and thus the broader absorption peaks are captured for Au-1 NCs. On the other hand, all the absorption peaks of three Au NCs are laying in the range of 350-550 nm, but the absorption peaks of Au-1 NCs are slightly blue-shifted compared with that of Au-2 and Au-3 NCs. We reasoned that the surface charge re-distribution, which is induced by the diverse supramolecular interactions such as hydrogen bonding between ATT and ARG ligands and electrostatic interactions between ARG and TOA ligands, should account for this shift in absorption spectra. The surface charge re-distribution has been previously demonstrated by Jin's group and our group to induce slight distortion of geometric structure, further leading to the vibration in absorption peaks of gold NCs (*J. Phys. Chem. C* **2008**, 112, 14221; *Chem. Commun.* **2016**, 52, 5234). We have added additional discussion in the revised manuscript accordingly.

Revisions:

Page 9, Line 8-11:

“First, the long-lived (> 1 ns) relaxation component is assigned to the transition from the lowest excited singlet state (S_1) to the ground state (S_0), and this relaxation is not possible to acquire an accurate lifetime due to the limited decay time window (~ 8 ns) in our TA measurements.”

12. Figure S12: Why is there a calculated band for Au-2 at almost 600nm? (in contrast to: "...can well reproduce the experimental peaks"). Fitting single Gaussians (for the individual electronic transitions) to the absorption spectrum vs. wavelength makes no sense whatsoever. The authors use this to propose the existence of a band at 457nm. This is not convincing.

Reply: Thank you for this insightful comment. We are sorry that we didn't articulate well in some terms in our original manuscript. As shown in Figure R4, no small peak at 600 nm can be observed in the absorption spectrum of Au-1 NCs using Gaussian software, while it appears in the absorption spectrum of Au-2 NCs. Therefore, we attribute the small absorption peak at 600 nm to the ARG molecules, wherein bind to the ATT molecules through supramolecular hydrogen bonding interactions. Note that the small peak at 600 nm is too weak to be observed in the experimental absorption spectrum of Au-2 NCs. A similar phenomenon can also be observed when performing the Dmol3 calculation, in which a peak of 750 nm is found instead of one at 600 nm.

In addition, we also show our best gratitude to the reviewer for his/her kind reminder about the Gaussians' fitting of absorption spectra. As shown in Figure R5, the physics parameter of wavelength was transformed to be wavenumber. The subsequent Gaussians fitting on the absorption spectra of Au-1 NCs gives three subpeaks located at 21089, 24129, and 25500 cm^{-1} , respectively. We have included Figure R5 in this revision as Supplementary Figure 12 in the supplementary file.

Figure R4. The simulated UV absorption spectra of Au-1 and Au-2 NCs using (a,b) Dmol and (c,d) Gaussian software packages, respectively. The small peaks are highlighted in red.

Figure R5. The experimental UV-vis absorption spectra of Au-1 and Au-2 NCs. The latter peak at 24875 cm^{-1} in experimental UV-vis absorption spectra of Au-1 NCs was fitted to be additional peaks at 24129 and 25500 cm^{-1} , which were colored in green and blue, respectively.

Revisions:

SI, Page 18, Supplementary Figure 12:

Figure R5 has been included as Supplementary Figure 12b in the supplementary file.

13. It is important to know how the QYs with the integrating sphere were measured. Very often such a measurement requires to have the reference (solvent only) being measured using ND-filters, so as to remain in the linear regime of the detector (not doing so could give unreasonably large QYs). Nothing is mentioned here. The authors should explain the exact procedure and show the raw data.

Reply: Thank you for this professional comment. We measured the absolute PLQYs of all gold NCs on a FLS1000 spectrofluorometer (Edinburgh Instruments Ltd.) attached with an integrating sphere coating with a BaSO₄ layer. Before the measurements, the optical density (OD) of all Au NCs at 405 nm was fixed at 0.1 by diluting them with purified water for Au-1 and Au-2 NCs and toluene for Au-3 NCs. For detailed measurements, a Xe lamp with a fixed emission wavelength at 405 nm for three samples was employed as the excitation source. The intensity factor was set as 100, and the excitation and emission slit was set as 2.2 and 0.22 nm, respectively. The scanning range was fixed from 385 nm to 650 nm, and the data were collected 1 nm per step. For the detailed measurements of the PLQY of Au-1, Au-2, and Au-3 NCs, the pure solvents of water (for Au-1 and Au-2 NCs) and toluene (for Au-3 NCs) were measured first in the integrating sphere as their corresponding blank references. Then, the pure solvent was replaced with pre-adjusted gold NCs solutions, and their emission spectra were directly measured without any changes in the experimental conditions. The calculation of absolute QYs values was conducted on the built-in “Fluoracle” software (version 2.13.2). The corresponding raw data of Au-1, Au-2, and Au-3 NCs were shown in Figure R6a, Figure R6b, and Figure R6c, respectively. This measurement method is suitable for Au-2 and Au-3 NCs because their emission peak can be easily distinguished. While for Au-1 NCs, their PLQY is extremely low, leading to their inconspicuous emission profile.

In this regard, it is necessary to increase the excitation intensity, and therefore the excitation and emission slit was set to be 5.5 and 0.55, respectively. In order to control the saturation condition of the visible PMT-900 detector, the intensity factor was fixed as 50 (which is realized by the self-equipped electronic ND-filter in FLS1000, but the detecting intensity is automatically compensated by the measurement software) in the scan processes of reference and Au-1 NCs scattering from 390 nm to 430 nm. This factor was reset to 100 (without ND-filter) in the scan processes of reference and Au-1 NC emission from 430 nm to 650 nm. This measurement method is usually employed for the PLQY measurements of emitters with NIR emission or upconversion emission because of their inherent weak emission intensity. As shown in Figure R6d, the emission of Au-1 NCs thus can be identified, and their PLQY value was calculated to be 0.2%, which is still too low and should have relatively large uncertainty. In this regard, we have revised the PLQY of Au-1 NCs to be < 0.3% throughout the manuscript to improve its rigor.

Figure R6. Measurements of the absolute PLQY of (a) Au-1, (b) Au-2, and (c) Au-3 NCs, respectively. The insets in (a-c) show the amplified spectra in the wavelength range of 480-600 nm. (d) Measurement of the absolute PLQY of Au-1 NCs by recording the scattering and emission of reference and Au-1 NCs separately. The inset in (e) shows the amplified spectra in the wavelength range of 480-600 nm.

Revisions:

SI, Page 2, Line 15-29:

“Before the absolute PL quantum yields (PLQY) measurements of the dilute solution of three gold NCs, their optical density (OD) at 405 nm was fixed at 0.1 by diluting them with water for Au-1 and Au-2 NCs and toluene for Au-3 NCs. The PLQY of all gold NCs were measured and calculated on a FLS1000 spectrofluorometer (Edinburgh Instruments Ltd.) attached with an integrating sphere coating with a BaSO₄ layer. A Xe lamp with a fixed emission wavelength at 405 nm for three samples was employed as the excitation source. The intensity factor was set as 100, and the excitation and emission slit was set as 2.2 and 0.22 nm, respectively. The scanning range was fixed from 385 nm to 650 nm, and the data were collected 1 nm per step. For the detailed measurements of the PLQY of Au-1, Au-2, and Au-3 NCs, the pure solvents of water (for Au-1 and Au-2 NCs) and toluene (for Au-3 NCs) were measured first in the integrating sphere as the blank reference. Then, the pure solvent was replaced with pre-adjusted gold NCs solutions, and their emission spectra were directly measured without any changes

in the experimental conditions. The calculation of absolute QYs values was conducted on the built-in “Fluoracle” software (version 2.13.2).”

Page 4, Line 14, 24, 28, 30, Page 15, Line 11, 12, SI, Page 43, 44, 58, Supplementary Figure 37, 38, Supplementary Table 2:

The error regions for the recorded PLQY values of all gold NCs have been calculated and added accordingly in the revised manuscript and supplementary file.

SI, Page 57, 62, 63, Supplementary Table 1, 6, 7:

The error regions for the fitted lifetimes of all gold NCs have been calculated and given accordingly in the revised supplementary file.

14. Figure S18: Are these not species associated spectra (SAS) instead of DAS?

Reply: Yes, they are DAS but not the SAS. As to the technical processing of TA data, we use “Surface Xplorer 4.2.0”, which is the built-in analytical software of TA data recorded on Ultrafast Systems (USA), to conduct global fitting analysis. As a result, the DAS can be acquired.

15. The chemical formula provided by the authors (page 5) for the three samples is an approximation at best. I guess in solution the situation is quite dynamic. Maybe NMR DOSY experiments could give some more information.

Reply: We share a similar view with the reviewer that the serial Au NCs are quite dynamic in the solution state, and therefore our proposed chemical formulas for three Au NCs are approximate. As suggested by the reviewer, we conducted ¹H-NMR DOSY measurements of water-soluble model NCs of Au-1 and Au-2 to verify this speculation. Au-1 and Au-2 NCs were purified by using an ultrafiltration tube (Millipore, 50 kDa) twice and a dialysis bag against the ultrapure water (100 mL, pH = 10.20) for 1 h, respectively, to remove excess ATT and ARG ligands. As shown in Figure R7, the diffusional coefficients of Au-1 and Au-2 NCs were not a constant value. They both feature a continuous band, ranging from $1.17 \times 10^{-10} \pm 6.137 \times 10^{-11}$ to $1.67 \times 10^{-11} \pm 3.002 \times 10^{-25}$ m²/s for Au-1 NCs, and from $9.31 \times 10^{-11} \pm 3.859 \times 10^{-11}$ to $4.68 \times 10^{-11} \pm 2.848 \times 10^{-11}$ m²/s for Au-2 NCs, respectively. Namely, the metal NCs are quite dynamic as there is a balance between the absorption and desorption of external ligands on the surface. Notably, the broader distribution of diffusional coefficient is detected in the ¹H-NMR DOSY spectra of Au-1 NCs, in comparison to that of Au-2 NCs. It would be attributed to the decreased vibration magnitude of the metal kernel of Au-2 NCs after anchoring the second-layer ligand of ARG.

Figure R7. ^1H -NMR and corresponding DOSY spectra of (a) ATT ligands, (b) ARG ligands, (c) Au-1 NCs, and (d) Au-2 NCs.

16. Also, the writing (language) needs to be improved substantially.

Reply: Thank you for this good suggestion. We have carefully refined our writing in this revised version. We hope it will meet the standard of *Nature Communications*.

17. In conclusion, the work does not meet the high standards normally required for publication in *Nature Communications*.

Reply: We would like to thank the reviewer again for his/her constructive comments and suggestions, which have spurred significant improvements in both the scientific content and the readability of this manuscript. After revising the original manuscript point-by-point, we expect to meet the high standards of *Nature Communications*. In addition, the mechanistic insights revealed in our study as well as the developed synthetic strategy of triple-ligand layer-by-layer self-assembly for brighter gold NCs, will be of interest to heterogeneous readers from communities of noble metal materials, luminescent materials, cluster chemistry, colloid and interface chemistry, and optical physics, which will also rapidly stimulate more basic and applied research in these cutting-edge areas.

Comments by Reviewer #3:

The manuscript by Zhong et. al. entitled “Suppression of Kernel Oscillation Boosts 90.3% Absolute Photoluminescence Quantum Yields of Metal Nanoclusters”, studied the efficient suppression of kernel oscillation achieving remarkable enhancement in emission intensity, by rigidifying the surface of metal NCs and propagating as-developed strains into the metal core. They used a layer-by-layer triple-ligands surface engineering to make the solution-phase Au NCs have strong metal core-dictated fluorescence, and the absolute quantum yield reached a record of 90.3%. They used ultrafast transient absorption (TA) to map out the photophysics of three Au nanoclusters. However, there is no crystal data in this work, and the current data cannot prove the formula of the samples. Moreover, the explanations of the TA data are confused, and the conclusions cannot be backed up by the current data. The following comments should be taken into consideration before the manuscript can be accepted.

Reply: We are glad that the reviewer finds this work interesting and important. We also gratefully appreciate the reviewers’ insightful and encouraging comments/suggestions, which have been taken into careful account in this revision. It is true that we have not gained the crystal data in this work. Therefore, we have removed the crystal structures of serial Au NCs as presented in the original submission. Instead, we further proposed a reasonable structure in the revised Supplementary file, based on the matchable experimental and theoretical absorption spectra of Au₁₀(SR)₆ NCs, whose total structure has already been theoretically predicted. Moreover, we have carefully checked our TA data, and re-performed TA fitting according to the reviewers’ good suggestions. The explanations of the TA data are also re-sorted and consolidated accordingly to support our conclusions. Please see our detailed responses and revisions below for more details.

Specific comments:

1. How did the authors obtain the accurate structures of Au NCs (Figure 1c) without knowing the crystal data? If there is no solid evidence, the crystal structure of nanocluster cannot be drawn as shown in Figure 1.

Reply: Thank you for this insightful comment. Yes, due to the lack of single crystal data of serial Au NCs, we cannot give the accurate structures of Au NCs as shown in Figure 1 in our first submission. We have revised it to emphasize our strategy of triple-ligand layer-by-layer self-assembly in preparing the serial Au NCs, from Au-1 to Au-2 to Au-3 NCs, in leveraging the hydrogen bonding interactions between ATT and ARG ligands and electrostatic interactions between ARG and TOA ligands.

As for water-soluble metal NCs, a widely accepted consensus is that it is extremely difficult to acquire X-ray-quality single crystals. This fact is largely caused by the flexible ligands on the surface of aqueous-phase metal NCs, which are significantly

affected by the electrostatic attraction and repulsion, and dipolar interactions in water. In addition, it is more challenging in our system because of the disturbance of multiple ligands and the diverse supramolecular interactions in our serial NCs including hydrogen bonding and electrostatic interactions. We have tried to grow their single crystals but all our attempts failed to acquire X-ray-quality ones.

Moreover, to push our step in addressing the atomic-/molecular-precision structure, we proposed a reasonable structure in the supplementary information file. In the aspect of structural determination of water-soluble metal NCs, an empirical rule is that the population and packing manner of Au-S frameworks in the thiolate-protected metal NCs are closely related to their absorption characters (*J. Am. Chem. Soc.* **2008**, 130, 5883). In other words, it is feasible to assign the synthesized metal NCs by checking whether their absorption spectra are matching well or not with that of the target metal NCs. The target metal NCs of Au-1 in this work is Au₁₀(SR)₆, whose crystal structure has already been predicted (*J. Am. Chem. Soc.* **2015**, 137, 15809). We have simulated the UV-vis absorption spectrum of as-reported Au₁₀(SR)₆ NCs and found it matched well with our experimentally recorded absorption spectrum, indicating similar Au-S frameworks between Au-1 NCs and the reported Au₁₀(SR)₆ NCs. That is, the Au₇ kernel is fused by two Au₄ tetrahedrons and three monomeric [S-Au-S] staple motifs are anchored on the surface of the Au₇ kernel by covalent Au-S bonding. While the second-layer ligands of ATT and the third-layer ligands of TOA can be further anchored through supramolecular hydrogen bonding and electrostatic interactions to form Au-2 and Au-3 NCs, respectively (Figure R8). We have revised the corresponding section and included more necessary descriptions in the revised manuscript accordingly.

Figure R8. The proposed structural evolution from Au-1 to Au-2 to Au-3 NCs through layer-by-layer triple-ligand self-assembly.

Revisions:

Page 5, Figure 1:

Figure 1c has been redrawn and reorganized to be Figure 1a to highlight the triple-ligand layer-by-layer self-assembly strategy from Au-1, Au-2, to Au-3 NCs.

Page 6, Line 11-31:

“As for the crystal structure of serial Au NCs, we have tried to grow their high-quality single crystals but all our attempts failed to acquire X-ray-quality ones. This should be attributed to i) the supramolecular interactions of metal NCs are more diverse and complex (e.g., the synergistic and competitive effect between hydrogen bonding and electrostatic interactions), and thereby challenging to balance multiple driving forces for crystallization; ii) the triple-ligands layer-by-layer self-assembly in our cases enables relatively flexible and variant surface environments of metal NCs, which further impedes the formation of long-range-ordered packing to generate high-quality single crystals. Moreover, to push our step in addressing their atomic-level crystal structure, we proposed a reasonable structure in the supplementary information file. It should be mentioned that an empirical rule to estimate the crystal structure of water-soluble metal NCs is comparing their absorption spectra with that of known metal NCs⁴⁶. The target metal NCs of Au-1 in this work is Au₁₀(SR)₆, whose crystal structure has already been predicted⁴⁷. We have simulated the UV-vis absorption spectrum of as-reported Au₁₀(SR)₆ NCs and found it matched well with our experimentally recorded absorption spectrum (Supplementary Fig. 12), indicating similar Au-S frameworks between Au-1 NCs and the reported Au₁₀(SR)₆ NCs⁴⁸. That is, the Au₇ kernel is fused by two Au₄ tetrahedrons and three monomeric [S-Au-S] staple motifs are anchored on the surface of the Au₇ kernel by covalent Au-S bonding (Supplementary Fig. 13). While the second-layer ligands of ATT and the third-layer ligands of TOA can be further anchored through supramolecular hydrogen bonding and electrostatic interactions to form Au-2 and Au-3 NCs, respectively.”

Page 22, Line 23-27:

Additional references have been included:

46. Zhu, M. Z., Aikens, C. M., Hollander, F. J., Schatz, G. C. & Jin, R. C. Correlating the crystal structure of a thiol-Protected Au₂₅ cluster and optical properties. *J. Am. Chem. Soc.* **130**, 5883–5885 (2008).

47. Liu, C. Y., Pei, Y., Sun, H. & Ma, J. The nucleation and growth mechanism of thiolate-protected Au nanoclusters. *J. Am. Chem. Soc.* **137**, 15809–15816 (2015).

SI, Page 19, Supplementary Figure 13:

Figure R8 has been included as Supplementary Figure 13 in the supplementary file.

2. The authors claimed that ATT and ARG are bound by hydrogen bond on page 4. How did the authors determine the bonding between ligand layers?

Reply: Thank you for this insightful comment. There are structural and theoretical bases for us to determine the hydrogen bonding between ATT and ARG ligands. Structurally, ARG molecules are in Y-shaped and planar-orientated geometry and have a very high content of hydrogen bond donors. The guanidine groups attaching to the side chain of ARG molecules have been theoretically reported to have extreme advantages for binding other molecules through hydrogen bonding (*Coord. Chem. Rev.* **2003**, 240, 3; *Chem. Soc. Rev.* **2007**, 36, 198; *Chem. - Eur. J.* **2007**, 13, 6644; *Chem. Mater.* **2017**, 29, 1362).

In addition, we also give experimental evidence for the formed hydrogen bonding between ATT and ARG ligands by conducting $^1\text{H-NMR}$ DOSY experiments. As shown in Figure R9, the diffusional coefficient (D) of pure ATT ligands (peak 1 in Figure R9a) was determined to be $4.46 \times 10^{-10} \pm 4.816 \times 10^{-14} \text{ m}^2/\text{s}$. While this value for peak 1-4 of pure ARG ligands in Figure R9b was measured to be $4.62 \times 10^{-10} \pm 1.694 \times 10^{-13}$, $4.75 \times 10^{-10} \pm 2.561 \times 10^{-13}$, $4.71 \times 10^{-10} \pm 1.841 \times 10^{-13}$, and $4.72 \times 10^{-10} \pm 3.748 \times 10^{-13} \text{ m}^2/\text{s}$, respectively. After mixing ATT and ARG ligands in D_2O with the molar ratio of 1 to 1, the recorded D values for peak 1-5 in Figure R9c were $3.14 \times 10^{-10} \pm 8.975 \times 10^{-13}$, $3.14 \times 10^{-10} \pm 3.848 \times 10^{-13}$, $3.10 \times 10^{-10} \pm 2.839 \times 10^{-25}$, $3.16 \times 10^{-10} \pm 2.678 \times 10^{-13}$, and $3.14 \times 10^{-10} \pm 4.470 \times 10^{-13} \text{ m}^2/\text{s}$, respectively. Obviously, both diffusional coefficient of ATT signals (peak 3) and ARG signals (peak 1, 2, 4, 5) in ATT-ARG mixtures were slowed down, and ATT ligands share nearly the same diffusional coefficient with ARG ligands. These results demonstrated ATT and ARG ligands have interacted with each other. Based on the inherent structural features of ARG molecules, the most possible supramolecular interactions between these two species should be hydrogen bonding interactions.

Figure R9. $^1\text{H-NMR}$ and corresponding DOSY spectra of (a) ATT, (b) ARG, and (c) ATT-ARG mixtures.

3. In Figure S11, the MALDI-TOF mass spectra of three nanoclusters cannot prove the purity and the formula of the sample. ESI should be performed to further prove the purity and to determine the formula of three nanoclusters in the manuscript.

Reply: Thank you for this insightful comment. To further check the purity and chemical formulas of serial Au NCs, we have carried out ESI-MS measurements in both positive-

and negative-ion modes. As shown in Figure R10, it can be observed that the mass signals of intact NCs are hard to be distinguished for all three Au NCs regardless of the measuring mode. It can be rationally explained from the following insights. i) The ATT ligand is inherently hard to be ionized. From the viewpoint of the molecular structure of ATT, the attaching hydroxy groups are stable and hard to be charged in an electric field, which is distinctively different from the acidic groups (e.g., -COOH and -SO₃H) commonly used in the synthesis of water-soluble NCs that are easily analyzable by ESI-MS. In order to check this point, we conducted the ESI-MS measurements of pure ATT ligands and Au(I)-ATT complexes. Nevertheless, there are no obvious mass peaks detected for these two species, demonstrating that ATT ligands are not suitable for ESI-MS analysis. ii) The excess ARG⁻ and TOA⁺ ligands in the solution of Au-2 and Au-3 NCs. To maintain the stability and superior PL properties of Au-2 and Au-3 NCs, excess ARG⁻ (~ 3 folds higher in molar concentration) and TOA⁺ (~ 18 folds higher in molar concentration) are typically indispensable. The excess self-charged ARG⁻ and TOA⁺ ligands can significantly interfere with the mass signals of intact Au-2 and Au-3 NCs. Even if we have attempted to purify these NCs several times, there are still only ARG⁻ and TOA⁺ mass signals were detected for Au-2 and Au-3 samples, respectively. However, it should be noted that metal NCs in a solution state are very dynamic due to the underlying absorption and desorption of external ligands on the surface of metal NCs. For our Au NCs systems with multiple ligand layers, quantificationally acquiring their chemical formulas are more difficult. Therefore, our proposed chemical formulas for Au-1, Au-2, and Au-3 NCs based on their corresponding MALDI-TOF and XPS results are approximate.

Figure R10. ESI mass spectra of pure ATT ligands, Au(I)-ATT complexes, Au-1, Au-2, and Au-3 NCs operated in (a) positive- and (b) negative-ion mode.

4. The emission band observed in three NCs are sharp and the stokes shift is very small, which is unusual for gold clusters. Is there an explanation for this?

Reply: Thank you for this insightful comment. The origin of photoluminescence in metal NCs is an open question in this field. Generally speaking, two kinds of PL

mechanisms have been empirically summarized by researchers: the core-state emission and the surface-state emission (*Natl. Sci. Rev.* **2021**, 8, nwaa208; *Chem. Soc. Rev.* **2019**, 48, 2422; *Coord. Chem. Rev.* **2019**, 378, 595). Namely, the core-state emission structurally originates from the inner metal core, and features with small Stokes shift (less than 100 nm), high emissive energy (> 2.20 eV), narrow emission band (FWHM < 100 nm), and short decay lifetimes (*ns*-level). On the other hand, the surface-state emission is stemming from the staple motifs shell, and features with large Stokes shift, low emissive energy, broad emission band, and long decay lifetimes (μ s-level). In our work, three Au NCs exhibit *ns*-level fluorescent lifetimes (3.5-61.0 ns), small Stokes shift (1479-982 cm^{-1}), relatively higher emitting energy (~ 2.33 eV), and narrow-band emission. Based on these characteristics, the as-detected PL in Au-1, Au-2, and Au-3 NCs can be rationally assigned to be the core-state fluorescence. Similar PL properties with sharp emission band and small Stokes shift were also reported in the Zn^{2+} -ion-mediated assemblies of $\text{Au}_4(\text{SRCOO}^-)_4$ clusters, which were reported to possess bright greenish-blue fluorescence at 485 nm, FWHM of 25 nm, and fluorescent lifetimes of 33.0 ns (*J. Am. Chem. Soc.* **2021**, 143, 326). We think these distinct PL features are tightly related to the core size of metal NCs, allowing a stronger quantum confinement effect and thus leading to higher energetic emission with a narrower emission band and small Stokes shift.

5. Suppressing the structural vibration will prevent nonradiative relaxation, but suppression of the coherent oscillations of metal kernel cannot suppress the non-radiative processes. How did the authors understand the differences between “coherent vibrations” and “structural vibrations”?

Reply: Thank you for this insightful comment. We share the reviewer’s view that suppressing the structural vibration will prevent non-radiative relaxation but is unable for coherent vibration. In our opinion, coherent vibration denotes the coherent spatial movement of core atoms induced by the irradiation of a short impulsive laser. Without external pumping sources, coherent vibration of metal NCs cannot show up. Coherent vibration is reflected by the optical coherent oscillation signals, which can be distinguished from the TA decays at an early time at most probed wavelengths. The vibrational frequencies and decay times directly provide the periods and damping of corresponding mechanical vibrational modes (*Nano Lett.* **2018**, 18, 6842; *ACS Nano* **2021**, 15, 13980). On the other hand, structural vibration is the intrinsic feature of metal NCs. No matter whether applied with an external impulsive laser or not, structural vibration is always existing in metal NCs. However, coherent vibration is significantly affected by structural vibration, resulting in the differences in the amplitude of optical coherent oscillation signals and further varied PL properties in metal NCs. For example, Kang et al. have observed the prominent coherent oscillations in the femtosecond kinetic traces of $\text{Pt}_1\text{Ag}_{24}$ NCs. But no such phenomenon was observed for $\text{Pt}_1\text{Ag}_{28}$ NCs. They speculated that the stronger phonon behaviors observed in $\text{Pt}_1\text{Ag}_{24}$ NCs suggest that more excited-state energy is dissipated into the environment through lattice vibration, which finally leads to weaker luminescence in $\text{Pt}_1\text{Ag}_{24}$ NCs (0.1% QYs) than

that of Pt₁Ag₂₈ NCs (4.9% QYs, *Chem. Sci.* **2017**, 8, 2581). Accordingly, we have included more interruptions in the revised manuscript.

Revisions:

Page 13, Line 1-3:

“The coherent oscillation, which is the reflection of the pump pulse-induced coherent spatial movement of core atoms, thus can be distinguished from the TA decays at an early time at most probed wavelengths.”

6. The PLQY of Au-2 and Au-3 NCs are 59.6% and 90.3%, respectively. Anyway, if there is any strong emission observed, strong SE should be observed in the fs-TA spectra. I check the fs-TA data in Figure 3, unfortunately, why is there no any stimulated emission (SE, 500-600 nm) signals observed on the TA map. Also, I check the fs-TA map in Figure 3, it seems that the authors should recheck the chirp correction (it seems that the chirp correction is not fully corrected or completed), that the chirp corrections for fs-TA spectra should be rechecked carefully. By the way, I also suggest that the authors should redraw all the fs-TA map with time-delay related spectra line-by-line, because map cannot judge the spectral quality. All the fitted data should give error bar or error region, and necessary residual and χ^2 should also be given to judge the fitting quality.

Reply: We would like to acknowledge this reviewer for his/her constructive and professional comments/suggestions. We are in complete agreement with the review that strong SE signals usually appear for strong luminescent materials, such as colloidal quantum dots (*J. Phys. Chem. C* **2020**, 124, 8448), graphene materials (*Nat. Commun.* **2016**, 7, 11010), and organic emitters (*Chem. Phys. Lett.* **2000**, 319, 157), etc. However, as for metal NCs, this kind of material is very special. Typically, metal NCs are composed of few- to hundred- metal atoms, and their size is less than 3 nm, approaching to Fermi wavelength of electrons. As a result, they typically exhibit a strong quantum confinement effect and possess many molecular-like properties, including discrete energy levels, multiple absorption peaks, and intense luminescence (*Chem. Rev.* **2016**, 116, 10346; *Chem. Rev.* **2017**, 117, 8208; *Chem. Soc. Rev.* **2019**, 48, 2422). Nevertheless, it is widely reported that no SE signals were acquired in the TA spectra of luminescent metal NCs, even though their QYs were recorded to be extremely high. For example, Song et al. reported the ultrabright Au@Cu₁₄ nanoclusters with 71.3% QYs in a non-degassed solution at room temperature (*Sci. Adv.* **2021**, 7, eabd2091). But no SE signals were recorded at the emission position of these NCs in their corresponding TA spectra. Similar results were also discovered in Zhou’s research (*J. Phys. Chem. C* **2015**, 119, 18790). We speculate that the undetected SE signals in metal NCs may be caused by their blending with stronger GSB signals (*Nat. Nanotechnol.* **2016**, 11, 872) and/or the competition between SE and ESA processes (*Nature* **2018**, 563, 541).

In addition, following the reviewer's kind suggestions, we have reconducted the chirp correction for all TA maps. Time zero correction was also carried out to ensure that the TA data are starting at the time zero position ($t = 0$). The fitting results of TA kinetics were accordingly revised. The TA maps depicted within line-by-line formation were included in the supplementary file for better identification of spectral quality. The error region, fitting residual, and reduced χ^2 factor for lifetime fitting were also added.

Revisions:

**Page 10, Figure 3; Page 12, Figure 4, SI, Page 26, 28, 30, 34, 36, 38, 52, 53
Supplementary Figure 20, 22, 24, 28, 30, 32, 46, 47:**

All TA maps were re-drawn after conducting chirp correction and time zero correction.

**Page 10, Figure 3; Page 12, Figure 4, SI, Page 40-42, 52, Supplementary Figure
34-36, 46:**

All the fits of TA traces were re-performed based on the corrected TA maps, and the fitting residual and reduced χ^2 factor were also provided.

SI, Page 62, 63, Supplementary Table 6, 7:

All the time constants with additional error regions were re-fitted based on the corrected TA maps.

7. *The authors mentioned that “the amplitude of ps-level components gradually decreased in corresponding DAS spectra (Figure S18), but it is strengthened for the last ns-level component. Namely, the non-radiative pathways are significantly inhibited and the saved energy in excited electrons is transferred and contributed to the radiative transition, leading to the pronouncedly lengthened radiation lifetime.” The statement in these sentences makes no sense. First, the DAS spectra cannot stand for the spectra of transient species, thus the opinion of inhibiting nonradiative pathways cannot be obtained. Second, the “amplitude” in the above sentence are in the GSB region, which cannot be used to compare the “radiative transition”.*

Reply: Thank you for this insightful comment. We totally agree with the reviewer that the DAS spectra cannot stand for the spectra of transient species, and the variation of DAS spectral amplitude at the GSB region cannot be directly compared to illustrate the changes of radiative transition in three Au NCs. Based on your nice suggestion and to avoid any possible confusion, we have deleted the related statement in the revised manuscript.

8. *In page 10, the authors state “Under near-HOMO-LUMO gap pump, only the two ps-level time components are fitted out.” However, time constants extracted from the global fitting of corresponding TA maps of Au-1, Au-2, and Au-3 NCs excited at near-HOMO-LUMO gap pump are ps and ns scales, respectively. Please recheck this*

sentence.

Reply: We would like to thank the reviewer for his/her careful eyes! And we are sorry that we didn't articulate well in some terms. In the revised manuscript, we have re-fitted the TA traces according to reviewers' constructive comments/suggestions. And it has been changed to be that two picosecond-level time constants and > 1 ns component are fitted out under the same pumping condition. We have included this key information in the revised manuscript.

Revisions:

Page 11, Line 13-15:

“However, upon near-HOMO–LUMO gap pumping, only two picosecond-level time constants and > 1 ns component are fitted out (Supplementary Figs. 34-36 and Supplementary Table 7).”

9. Why the lack of ultrafast decay component under near-bandgap excitation means the absence of core-shell relaxation? From Figure S27-S29 and Table S7, it seems that the second process disappeared, which means the ultrafast decay component didn't disappear.

Reply: This is another very insightful comment. In our statement, we have rationally assigned the as-detected PL in Au-1, Au-2, and Au-3 NCs to be the core-state fluorescence based on its experiential identifying standard of *ns*-level PL lifetimes, small Stokes shift, relatively higher emitting energy, and narrow-band emission (small FWHM). Here we attempt to rule out the possible surface-state emission in three Au NCs. An obvious difference between core-state and surface-state emission is the existence of ultrafast core-shell relaxation (namely, intersystem crossing, ISC) from S_1 to T_1 state in surface-state emission (e.g., 1 ps component in Au₂₅(SR)₁₈ and Au₃₈(SR)₂₄ NCs, *J. Phys. Chem. C* **2010**, 114, 22417; *J. Phys. Chem. C* **2013**, 117, 23155; *J. Phys. Chem. C* **2010**, 114, 19935; *J. Phys. Chem. C* **2017**, 121, 10686). The verification of ISC can be conducted by employing the TA technique excited by near-band gap energy. Under this condition, the ultrafast IC decay vanished while the ISC component can be clearly distinguished in surface-state emitting NCs. According to the reviewer and other reviewers' concerns, we have re-performed the global fitting of TA maps. It thus gives four decay components: 320 fs, 2.2 ps, 80.6 ps, and > 1 ns for Au-1 NCs; 240 fs, 2.9 ps, 80.7 ps, and > 1 ns for Au-2 NCs; 118 fs, 4.1 ps, 80.5 ps, and > 1 ns for Au-3 NCs, respectively. The TA traces illustrated in Supplementary Figures 34-36 in the revised supplementary file were also re-fitted. The results show that the ultrafast *fs*-level IC decay is completely disappeared and no additional ultrafast component is detected under near-bandgap excitation for all Au NCs. However, the second *ps*-level component change to be much smaller: from 2.2 to 1.5 ps for Au-1 NCs; from 2.9 to 1.9 ps for Au-2 NCs; from 4.1 to 2.5 ps for Au-3 NCs. The shortened second *ps*-level component mainly contributed to the fast decay component. Therefore, there is no ISC

contribution for the PL in three Au NCs, and the excited electrons are mainly localized in the metal core to form core-state emission. We have added this discussion in the revised manuscript accordingly.

Revisions:

Page 11, Line 13-24:

“However, upon near-HOMO–LUMO gap pumping, only two picosecond-level time constants and > 1 ns component are fitted out (Supplementary Figs. 34-36 and Supplementary Table 7). It should be noted that the near-HOMO–LUMO gap pumping can merely excite the metal core state for metal NCs. The remaining two picosecond-level decay components again evidence that they belong to metal core-directed structural relaxation. In addition, the ultrafast IC decay is completely disappeared and no additional ultrafast component is detected under near-bandgap excitation for all Au NCs. Keeping this in mind, we are now safe to rule out the contribution of possible core-shell relaxation to the inherent PL of serial Au NCs because this process is typically observed in an especially short time scale (e.g., 1 ps in Au₂₅(SR)₁₈⁵⁶ and Au₃₈(SR)₂₄⁵⁷). Therefore, the excited electrons in serial Au NCs are limited in the domain of metal kernel, irrelevant of any surface-related non-radiative channels.”

Page 23, Line 10-13:

Additional references have been added to support our statement:

56. Qian, H. F., Sfeir, M. Y. & Jin, R. C. Ultrafast relaxation dynamics of [Au₂₅(SR)₁₈]^q nanoclusters: effects of charge state. *J. Phys. Chem. C* **114**, 19935–19940 (2010).

57. Zhou, M. et al. Ultrafast relaxation dynamics of Au₃₈(SC₂H₄Ph)₂₄ nanoclusters and effects of structural isomerism. *J. Phys. Chem. C* **121**, 10686–10693 (2017).

10. In Figure 4, the FFT should be carried out after subtracting the population dynamics, refer to *J. Phys. Chem. Lett.* 2018, 9, 7085–7089, *ACS Nano*, 2010, 4, 3406–2412. Because the high amplitude may be caused by excited-state population dynamics.

Reply: Thank you for the good suggestion. We have extracted pure acoustic oscillation signals by subtracting the electron dynamics using exponential decay. The corresponding FFT on these acoustic oscillations was following re-conducted. In addition, we have cited and briefly discussed this data processing method reported by Iwamura, Tahara, and Goodson’s group (*J. Phys. Chem. Lett.* **2018**, 9, 7085; *Acc. Chem. Res.* **2015**, 48, 782; *ACS Nano*, **2010**, 4, 3406) in the revised manuscript.

Revisions:

Page 13, Line 9-10:

“The pure acoustic oscillations (the insets in Fig. 4d-f) were extracted by

subtracting the electron dynamics using exponential decay⁵⁹⁻⁶¹.”

Page 13, Line 14-15:

“As shown in Fig. 4g-i, serial Au NCs exhibit similar vibration frequency of 59.0 (1.77 THz), 58.7 (1.76 THz), and 57.3 cm⁻¹ (1.72 THz) but a remarkable reduction in the variation amplitude.”

Page 23, Line 16-23:

New references have been added to support our statements:

59. Iwamura, M. et al. Metal–metal bond formations in [Au(CN)₂]⁻_n (n = 3–5) oligomers in water identified by coherent nuclear wavepacket motions. *J. Phys. Chem. Lett.* **9**, 7085–7089 (2018).

60. Iwamura, M., Takeuchi, S. & Tahara, T. Ultrafast excited-state dynamics of copper(I) complexes. *Acc. Chem. Res.* **48**, 782–791 (2015).

61. Varnavski, O., Ramakrishna, G., Kim, J., Lee, D. & Goodson III, T. Optically excited acoustic vibrations in quantum-sized monolayer-protected gold clusters. *ACS Nano* **4**, 3406–3412 (2010).

Comments by Reviewer #4:

Briefly, the work is interesting but preliminary. The importance of the high QY in a gold nanocluster and the ways in which they can be achieved is of interest for Nature Communications, but the current manuscript draft and structural hypothesis needs to be improved. The authors either need to provide better support for their proposed structures or completely remove the current proposed structures.

Reply: We are grateful to this reviewer for his/her professional efforts in reviewing our manuscript. Indeed, we have achieved a significant improvement of QYs from < 0.3% (~ 0.2%) to 59.6% to 90.3% in gold NCs through a triple-ligand layer-by-layer self-assembly strategy. The underlying mechanism was decoded and ascribed to suppressed coherent vibration of metal core indirectly affected by the surface rigidity. Our findings give new insights for the optimization of PL properties of metal NCs, which would increase their acceptance as novel luminescent materials and promote their practical applications. We also appreciate the useful and comprehensive comments/suggestions by the reviewer, which have spurred improvements in both the readability and scientific content of our manuscript. These comments/suggestions have been taken into careful consideration and a point-to-point response could be found in the coming paragraphs.

Regarding the presented structures of serial Au NCs in the original submission, we have removed them due to the lack of single crystal data. This fact is largely caused by the flexible ligands on the surface of aqueous-phase metal NCs, which are significantly affected by the electrostatic attraction and repulsion, and dipolar interactions in water. In addition, it is more challenging in our system because of the disturbance of multiple ligands and the diverse supramolecular interactions in our serial NCs including

hydrogen bonding and electrostatic interactions. we have tried to grow their single crystals but all our attempts failed to acquire X-ray-quality ones. Instead, we have revised Figure 1 to emphasize our strategy of triple-ligand layer-by-layer self-assembly in preparing the serial Au NCs, from Au-1 to Au-2 to Au-3 NCs, in leveraging the hydrogen bonding interactions between ATT and ARG ligands and electrostatic interactions between ARG and TOA ligands.

Moreover, to push our step in addressing the atomic-/molecular-precision structure, we proposed a reasonable structure in the supplementary information file. In the aspect of structural determination of water-soluble metal NCs, an empirical rule is that the population and packing manner of Au-S frameworks in the thiolate-protected metal NCs are closely related to their absorption characters (*J. Am. Chem. Soc.* **2008**, 130, 5883). In other words, it is feasible to assign the synthesized metal NCs by checking whether their absorption spectra are matching well or not with that of the target metal NCs. The target metal NCs of Au-1 in this work is Au₁₀(SR)₆, whose crystal structure has already been predicted (*J. Am. Chem. Soc.* **2015**, 137, 15809). We have simulated the UV-vis absorption spectrum of as-reported Au₁₀(SR)₆ NCs and found it matched well with our experimentally recorded absorption spectrum, indicating similar Au-S frameworks between Au-1 NCs and the reported Au₁₀(SR)₆ NCs. That is, the Au₇ kernel is fused by two Au₄ tetrahedrons and three monomeric [S-Au-S] staple motifs are anchored on the surface of the Au₇ kernel by covalent Au-S bonding. While the second-layer ligands of ATT and the third-layer ligands of TOA can be further anchored through supramolecular hydrogen bonding and electrostatic interactions to form Au-2 and Au-3 NCs, respectively. We have revised the corresponding section and included more necessary descriptions in the revised manuscript accordingly.

Specific comments:

1. p. 4 “Stocks” shift should be “Stokes”

Reply: The misspelling of the terms “Stocks” has been corrected to “Stokes” throughout the manuscript.

2. Do the authors have a crystal structure for the Au₁₀(SR)₆ structure? The top and bottom faces appear very unprotected. The authors state “we further reconstruct the total structure of serial Au NCs based on grand unified model for describing the growth of the gold cores and ring model for describing interfacial interactions between the protection motifs and gold cores in thiolate-protected gold nanoclusters.[45] As depicted in Figure 1c, the resultant refined total structure ...” Again, it is not clear how they get this unless they are hypothesizing the structure based on other structures for other NC stoichiometries. The authors are doing some theory, or so they say in the SI (but it is not discussed at all in the main text?). Have they considered assessing isomer energies? Furthermore, the scheme presented in Figure 1 seems rather simplistic. The TOA ligands are unlikely to be planar as shown. They should be quite tetrahedral around the N atom, and the ligand is likely to ‘clump’.

Reply: Thank you for this insightful comment. Indeed, we have tried to grow the single crystals of water-soluble Au-1 NCs, but all attempts failed to acquire X-ray-quality ones. Thus, we have removed Figure 1c from our original submission. However, in previous work by Pei et al. (*J. Am. Chem. Soc.* **2015**, 137, 15809), the authors combined a genetic algorithm by using the local minimums primarily searched by the basin-hopping as the initial individuals in the search of local minima of Au₁₀(SR)₆. A genetic algorithm is an evolutionary algorithm based on the population, which generates new individuals by applying the crossover and mutation operation to the parent populations and achieves a better offspring according to the “survival of the fitness” principle. With the combination of the basin-hopping algorithm and genetic algorithm, they picked out the global minimum and some energetically degenerate or higher-lying metastable isomers, as shown in Figure R11, in which three stable isomers of Au₁₀(SR)₆ were identified. On this base, the most stable isomer 3 with the lowest formation energies was employed to reconstruct the structures of Au-1, Au-2, and Au-3 NCs, and further simulate their corresponding absorption spectra.

In addition, we totally agree with the reviewer’s suggestion that the molecules of TOA are more likely in a tetrahedral configuration. However, due to the lack of direct single crystal data of Au-3 NCs, we cannot determine the configuration of TOA ligands in Au-3 NCs. In this context, we referenced the crystal structure of [Au₂₅(SR)₁₈]⁺[TOA⁺]⁻ NCs (*J. Am. Chem. Soc.* **2008**, 130, 5883; *J. Am. Chem. Soc.* **2008**, 130, 3754), in which the TOA ligands play a role as counter ions to stabilize [Au₂₅(SR)₁₈]⁺ NCs through similar electrostatic interactions to Au-3 NCs. We found the configuration of TOA ligands did not change significantly in the optimizing process of Au-3 structures, indicating that the nearly planar configuration of TOA ligands in Au-3 NCs is thermodynamically stable. We speculate this result may be caused by the collective influences of steric hindrance and electrostatic interactions.

Figure R11. The comparison of stable isomers of Au₁₀(SR)₆ NCs with different formation energies (reproduced from *J. Am. Chem. Soc.* **2015**, 137, 15809).

*Reprinted (adapted) with permission from Liu C, Pei Y, Sun H, Ma J. The Nucleation and Growth Mechanism of Thiolate-Protected Au Nanoclusters. *J. Am. Chem. Soc.* 2015, 23;137(50):15809-16. Copyright 2015 American Chemical Society.

Revisions:

Page 6, Line 11-31:

“As for the crystal structure of serial Au NCs, we have tried to grow their high-quality single crystals but all our attempts failed to acquire X-ray-quality ones. This should be attributed to i) the supramolecular interactions of metal NCs are more diverse and complex (e.g., the synergistic and competitive effect between hydrogen bonding and electrostatic interactions), and thereby challenging to balance multiple driving forces for crystallization; ii) the triple-ligands layer-by-layer self-assembly in our cases enables relatively flexible and variant surface environments of metal NCs, which further impedes the formation of long-range-ordered packing to generate high-quality single crystals. Moreover, to push our step in addressing their atomic-level crystal structure, we proposed a reasonable structure in the supplementary information file. It should be mentioned that an empirical rule to estimate the crystal structure of water-soluble metal NCs is comparing their absorption spectra with that of known metal NCs⁴⁶. The target metal NCs of Au-1 in this work is Au₁₀(SR)₆, whose crystal structure has already been predicted⁴⁷. We have simulated the UV-vis absorption spectrum of as-reported Au₁₀(SR)₆ NCs and found it matched well with our experimentally recorded absorption spectrum (Supplementary Fig. 12), indicating similar Au-S frameworks between Au-1 NCs and the reported Au₁₀(SR)₆ NCs⁴⁸. That is, the Au₇ kernel is fused by two Au₄ tetrahedrons and three monomeric [S-Au-S] staple motifs are anchored on the surface of the Au₇ kernel by covalent Au-S bonding (Supplementary Fig. 13). While the second-layer ligands of ATT and the third-layer ligands of TOA can be further anchored through supramolecular hydrogen bonding and electrostatic interactions to form Au-2 and Au-3 NCs, respectively.”

Page 22, Line 23-27:

Additional references have been included:

46. Zhu, M. Z., Aikens, C. M., Hollander, F. J., Schatz, G. C. & Jin, R. C. Correlating the crystal structure of a thiol-Protected Au₂₅ cluster and optical properties. *J. Am. Chem. Soc.* **130**, 5883–5885 (2008).

47. Liu, C. Y., Pei, Y., Sun, H. & Ma, J. The nucleation and growth mechanism of thiolate-protected Au nanoclusters. *J. Am. Chem. Soc.* **137**, 15809–15816 (2015).

3. *Why are two different levels of theory used (for Au-1 and Au-2 vs. for Au-3)? Why not reoptimize Au-1 and Au-2 with the second level of theory if the first is too demanding for the larger system? Actually, it does not specify in the SI if the two structures are optimized or not (it states that Au-3 is optimized). Consistency is desirable for comparing the systems.*

Reply: Thank you for this good suggestion. We share the reviewer’s view that the consistency of theoretical simulations is desirable for comparing different systems. However, the employment of the calculation method of the Gaussian 09 program package for Au-1 and Au-2 NCs, and the Materials Studio Dmol3 program for Au-3 NCs is based on the consideration of the advantage and applicability of different programs. The Gaussian09 quantum chemistry software package has unparalleled

advantages in simulating the structures and properties of nanoclusters compared with other software packages and has become an essential research tool for thousands of chemists working in universities, research institutes, and commercial companies. When compared with the Dmol3 software package, the Gaussian09 software package has higher computational accuracy and more accurate solvation models. For example, the SMD (Solvation Model Based on Density) supported by Gaussian09/16 is almost the best implicit solvent model at present. The reason why it is good is that both the polar and non-polar parts have good functional forms and the parameterization process is well done. While the COSMO (Conductor-like Solvation Model) supported by Dmol3 only clearly defines how to calculate the polar part of the solvent, the calculation method of the non-polar part is not clearly given. We have also simulated the absorption spectra of Au-1 NCs and Au-2 NCs using Dmol3 as shown in Figure R12. It can be found that the results from Dmol3 cannot reproduce the experimental results. Although the Gaussian09 software package has great advantages, there are still shortcomings. For example, it's very time-consuming for it to deal with a big system. While an important feature of the Dmol3 software package is that it can handle big systems. Therefore, we choose Gaussian09 to simulate the small system of Au-1 and Au-2 NCs and the Dmol3 software package to optimize the very big system of Au-3 NCs. Moreover, it should be mentioned that the structures of Au-1 and Au-2 NCs have also been optimized by Gaussian09. We are sorry for the confusion caused by the less-detailed articulation in the last submission. We have included additional statements in this revised version.

Figure R12. The simulated UV absorption spectra of Au-1 and Au-2 NCs using (a,b) Dmol and (c,d) Gaussian software packages, respectively. The small peaks are highlighted in red.

Revisions:

SI, Page 4, Line 32 to Page 5, Line 1:

“With the optimized Au-1 and Au-2 structures reported by Pei et al.¹, density-functional theory (DFT) calculations using the Gaussian 09 program package² are performed to obtain electronic properties of these clusters.”

SI, Page 5, Line 7-10:

“The choice of different software packages to optimize Au-1, Au-2, and Au-3, and further simulating their corresponding electronic properties is based on the consideration of guaranteeing the accuracy of simulated models but reducing the calculation time at the same time.”

4. The authors might consider simulating EXAFS spectra for their proposed structure/DFT structure in order to see if it matches experiment. A larger theory component to this work could help.

Reply: Thanks for this constructive suggestion. We totally agree with the reviewer that the manuscript could be improved by additional evidence that could unambiguously support our proposed and DFT-simulated structure of three Au NCs produced by layer-by-layer triple-ligands self-assembly strategy. As shown in Supplementary Figures 15 and 37 in the original supplementary file (they have been changed to Supplementary Figures 16 and 45 in our revised supplementary file), we have shown the FT of the k^2 -weighted EXAFS spectrum and corresponding fitting in R space, along with the $k^2\chi(k)$ space spectra and their corresponding fitting of Au-1, Au-2, Au-3, Au-2-TMPA, and Au-2-TMBA NCs. The fine-fitting results between our proposed serial Au NCs and experimental EXAFS data demonstrate that these structures are convincing. We have included the necessary illustrations in our revised manuscript.

Revisions:

Page 7, Line 20-24:

“The subsequent fine fitting results of least-squares $\chi(R)$ and $k^2\chi(k)$ space spectra demonstrate that our proposed structures of serial Au NCs are reliable (Supplementary Fig. 16). The quantitative coordination number (CN) was also extracted for three Au NCs (Supplementary Table 3-5), which exhibits a similar value of $CN_{Au-S} = \sim 1.1$ and $CN_{Au-Au} = \sim 3.3$ in average.”

5. For the NMR signals in Figure S10 parts b, d, f, please provide the integrations. Not very much ATT appears in Au-3 NCs according to part f, as far as it appears to me. Also, ARG broadens more than ATT seems to, which may indicate that it is closely associated with gold atoms (not far away as suggested by the Scheme).

Reply: Thank you for your insightful suggestions. As the reviewer suggested, the integration of ¹H-NMR signals in Supplementary Figures 10b, d, and f (in the original supplementary file) has been carried out (Figure R13). We also agree with the reviewer that few ATT signals can be observed in Supplementary Figure 10f (in the original

supplementary file). This result may be caused by significantly reduced flexibility in solution after ATT ligands were anchored with ARG⁻ ligands or external ARG-TOA binary complexes through supramolecular hydrogen bonding (*Chem. Rev.* **2019**, 119, 195). It should be pointed out that the broadening of ATT signals in the ¹H-NMR spectra of Au-1 NCs is attributed to the formation of Au-S bonds between Au atoms and ATT ligands. In addition, ARG molecules are in Y-shaped and planar-orientated geometry and have a very high content of hydrogen bond donors. The guanidine groups attaching to the side chain of ARG molecules have been theoretically reported to have extreme advantages for binding other molecules through hydrogen bonding (*Coord. Chem. Rev.* **2003**, 240, 3; *Chem. Soc. Rev.* **2007**, 36, 198; Arginine. *Chem. - Eur. J.* **2007**, 13, 6644; *Chem. Mater.* 2017, 29, 1362). Therefore, the prominent broadening of ¹H-NMR peaks at 1.59, 3.16, and 3.28 ppm in the ¹H-NMR spectra of Au-2 NCs can be reasonably assigned to the formations of hydrogen bonding between ATT and ARG ligands. Nevertheless, compared with ARG ligands, ATT ligands are more likely to interact with gold atoms owing to the preferred anchoring points in ARG ligands to form stable Au-S bonds.

Figure R13. $^1\text{H-NMR}$ spectra of (a) ATT ligand, (c) ARG ligand, (e) TOA ligand, and the comparison of (b) Au-1 NCs, (d) Au-2 NCs, and (f) Au-3 NCs with corresponding surface ligands. The integrations for $^1\text{H-NMR}$ signals were also provided.

Revisions:

SI, Page 16, Supplementary Figure 10:

Figure R13 has been included as Supplementary Figure 10 in the revised supplementary file.

SI, Page 47, 48, Supplementary Figure 41, 42:

The integrations of $^1\text{H-NMR}$ signals in Supplementary Figures 41b and 42b have been added accordingly.

6. Could the authors provide a reference for “For molecular-like metal NCs, a short impulsive laser can excite the coherent acoustic vibrations of metal kernel for a few to tens of ps.”? Is it the same as the following sentence? Could the authors also provide their coherent oscillation in wavenumbers and compare with other known values for acoustic vibrations? The authors say that other systems are “similar” regardless of size (Refs. 58-60), but do not provide numbers for comparison. It could be good to include a table of these values here?

Reply: We appreciate the insightful comment and nice suggestion from the reviewer. Yes, the sentence “for molecular-like metal NCs, a short impulsive laser can excite the coherent acoustic vibration of the metal kernel for a few to tens of ps” shares the same reference with the following sentence. In addition, we have included a new table in the supplementary file for the comparison of the coherent oscillation in wavenumbers in different metal NCs. It should be noted that the diction of “regardless of size” is not very accurate. For large counterparts of gold NPs, the continuum mechanical model can be used to predict the coherent oscillation, which gives rise to the empirical relationship $\nu = 50/R$ (R is the particle radius). However, for molecular-like NCs, the frequencies of the oscillation deviate from the $1/R$ law of the plasmonic NPs, and they are not strongly correlated with the size (Table R1, *Nano Lett.* **2018**, 18, 6842; *ACS Nano* **2021**, 15, 13980; *Chem. Sci.* **2022**, 13, 8124; *Science*, **2019**, 364, 279). Therefore, the structure rather than size plays a more significant role in the coherent vibration in these ultrasmall molecular-like Au NCs. We have included more discussion about this point in the revised manuscript.

Table R1. The experimentally reported frequency of coherent oscillations in molecular-like metal NCs.

Gold NCs	Au ₂₄₆	Au ₁₄₄	Au ₃₈	Au ₃₀	Rod Au ₂₅	Spherical Au ₂₅	Au ₁₀
Frequency (cm ⁻¹)	16.7	52.0	25.0	16.7	26.0	40.0	59.0/58.7/ 57.3

Reference	7	8	9,10	11	12,13	14	This work
-----------	---	---	------	----	-------	----	-----------

Revisions:

Page 13, Line 14-15:

“As shown in Fig. 4g-i, serial Au NCs exhibit similar vibration frequency of 59.0 (1.77 THz), 58.7 (1.76 THz), and 57.3 cm⁻¹ (1.72 THz) but a remarkable reduction in the variation amplitude.”

Page 13, Line 19-23:

“Plenty of pioneering works have confirmed that the oscillation frequency is an intrinsic parameter of metal NCs, and it shows less connection with their size with respect to the discovered empirical relationship of $\nu = 50/R$ (ν and R are frequency and particle diameter, respectively) in Au NPs (Supplementary Table 8)⁶⁴⁻⁶⁶.”

SI, Page 64, Supplementary Table 8:

Table R1 has been included as Supplementary Table 8 in the revised supplementary file.

SI, Page 67, Line 14-30:

Additional references have been added in the supplementary file to support our statement:

7. Zhou, M. et al. On the non-metallicity of 2.2 nm Au₂₄₆(SR)₈₀ nanoclusters. *Angew. Chem. Int. Ed.* **56**, 16257–16261 (2017).
8. Zhang, W. et al. Coherent vibrational dynamics of Au₁₄₄(SR)₆₀ nanoclusters. *Chem. Sci.* **13**, 8124–8130 (2022).
9. Senanayake, R. D., Guidez, E. B., Neukirch, A. J., Prezhdo, O. V. & Aikens, C. M. Theoretical investigation of relaxation dynamics in Au₃₈(SH)₂₄ thiolate-protected gold nanoclusters. *J. Phys. Chem. C* **122**, 16380–16388 (2018).
10. Zhou, M. et al. Ultrafast relaxation dynamics of Au₃₈(SC₂H₄Ph)₂₄ nanoclusters and effects of structural isomerism. *J. Phys. Chem. C* **121**, 10686–10693 (2017).
11. Zhou, M. et al. Three-orders-of-magnitude variation of carrier lifetimes with crystal phase of gold nanoclusters. *Science* **364**, 279–282 (2019).
12. Zhou, M. et al. Electron localization in rod-shaped triicosahedral gold nanocluster. *Proc. Natl. Acad. Sci. U.S.A.* **114**, E4697–E4705 (2017).
13. Sfeir, M. Y., Qian, H. F., Nobusada, K. & Jin, R. C. Ultrafast relaxation dynamics of rod-shaped 25-Atom gold nanoclusters. *J. Phys. Chem. C* **115**, 6200–6207 (2011).
56. Qian, H. F., Sfeir, M. Y. & Jin, R. C. Ultrafast relaxation dynamics of [Au₂₅(SR)₁₈]^q nanoclusters: effects of charge state. *J. Phys. Chem. C* **114**, 19935–19940 (2010).

7. The structures in Figure 5 do not look similar to the structures in the first figure?

Reply: Thank you for this insightful comment. The structures of three Au NCs in Figure 5 were simplified. What we attempt to schematically illustrate in Figure 5 by using these simplified structures is that the amplitude of coherent vibration of metal core in Au-2 and Au-3 NCs is significantly suppressed through layer-by-layer triple-ligands self-assembly of ARG⁻ and TOA⁺ ligands on the surface of Au-1 NCs. The reduced non-radiative relaxation enables the improved PL properties in Au-2 and Au-3 NCs. In order to minimize any possible confusion, we have re-drawn Figure 5 to improve the readability and scientific content of this figure.

Revisions:

Page 14, Figure 5:

The structures of Au NCs in Figure 5 have been re-drawn for clarity.

8. *I think the authors are overestimating the time required for structural relaxation. 95.4 ps seems very high.*

Reply: Thank you for this insightful comment. We agree that the reviewer that the time constant of 95.4 ps for the structural relaxation component is relatively over high. Based on the collective consideration of your concerns and another reviewer's comment on the photophysics of three Au NCs, we have re-performed lifetime fittings of TA kinetics, and four decaying components in total for each Au NCs were fitted out, that is, 320 fs, 2.2 ps, 80.6 ps, and > 1 ns for Au-1 NCs; 240 fs, 2.9 ps, 80.7 ps, and > 1 ns for Au-2 NCs; 118 fs, 4.1 ps, 80.5 ps, and > 1 ns for Au-3 NCs, respectively. The resultant time constants for structural relaxation are 2.2, 2.9, and 4.1 ps, which are much rationalized and approaches to the reported values in luminescent metal NCs showing core-state emission, such as 1.7 ps in [Au₂₁(SR)₁₂(PCP)₂]⁺ and 2.5 ps in [Au₂₃(SR)₁₆]⁻ (*J. Am. Chem. Soc.* **2019**, 141, 5314), and 11 ps in Au₂₈(CHT)₂₀ and Au₂₈(TBBT)₂₀ NCs.

Revisions:

Page 9, Line 3-6:

“The global fittings generally give four decay components: 320 fs, 2.2 ps, 80.6 ps, and > 1 ns for Au-1 NCs; 240 fs, 2.9 ps, 80.7 ps, and > 1 ns for Au-2 NCs; 118 fs, 4.1 ps, 80.5 ps, and > 1 ns for Au-3 NCs, respectively (Fig. 3e-g and Supplementary Fig. 18).”

9. *The figures shown in Figure 5 a,b,c are essentially the same except for the times listed. Perhaps this figure could be changed to emphasize the differences between the systems? The authors could improve this section to better explain what they are doing, what they learned, etc. for a nonspecialist reader.*

Reply: This is another very inspiring comment. According to the reviewer's nice

suggestion, we have reorganized the content of Figure 5. As shown in Figure R14, the schematical illustrations of Au-1, Au-2, and Au-3 NCs with strong, medium, and weak coherent vibrations of central gold atoms were added. Their corresponding QYs were also provided to connect the inherent structural features and resultant PL properties in serial Au NCs. Moreover, the energy diagrams for Au-1, Au-2, and Au-3 NCs were carefully decorated. The time constants for corresponding decaying components of excited state electrons were revised based on our re-fitted results of TA kinetics. The outer glow was depicted on the green lines to indicate the step-by-step enhancement of QYs from Au-1, to Au-2, and finally to Au-3 NCs. Therefore, we believe that the scientific content in the current revision of Figure 5 will be attractive and can be easier identified by the heterogeneous readers of *Nature Communications*.

Figure R14. Schematic illustration of the excited-state dynamics and coherent vibration of Au-1, Au-2, and Au-3 NCs. The red, orange, and gray solid curves in the energy diagrams stand for internal conversion, core-directed, and core-ligand-related structural vibration, respectively. The green solid lines with outer glow denote the fluorescent relaxation. The symbols of S_n , S_1 , and S_0 are the higher singlet state, the lowest singlet state, and the ground state, respectively. The purple, orange, and red circles at the bottom plane stand for layer-by-layer coating of ATT, ARG, and TOA ligands, and the black circles refer to the central gold atoms in three Au NCs.

Revisions:

Page 14, Figure 5:

Figure R14 has been included as Figure 5 in the revised manuscript.

Page 14, Line 9 to Page 15, Line 3:

“Based on the above results obtained from ultrafast excited-state dynamics and optical coherent oscillations, now we are ready to construct a schematic illustration

(Fig. 5) to correlate the coherent vibration and PL properties in three Au NCs. In a general model, the external excitation can pump electrons into a high-energy singlet state of S_n . Then, the excited electrons experience ultrafast IC relaxation within several hundreds of femtoseconds to cool into the S_1 state. The singlet populations continue to non-radiatively relax to the S_1 state with the lowest energy through the multiple core-directed and core-ligand-directed structural vibration in few-picosecond and dozens of picosecond time scales, respectively. Nevertheless, the core-directed structural vibration gives dominating non-radiative channels for the contribution of hot electrons in the S_1 state with respect to the core-ligand-directed structural vibration. Subsequently, the excited electrons localized in the gold core return to the S_0 ground state by radiatively emitting the ns -level core-state green fluorescence. Note that there is no additional ultrafast core-shell electronic relaxation in the excited-state dynamics, which is understandable because of the lack of μs -level surface-state phosphorescence in three Au NCs. The triple-ligands layer-by-layer decorations of ARG and TOA on the surface of Au-1 NCs enable the indirect delivery of the confinement effect from the surface into the gold core, leading to significantly suppressed coherent oscillation in Au-2 and especially Au-3 NCs. As a result, the suppressed non-radiative relaxation channels in turn boost the core-state green fluorescence⁶⁷. Thus, the absolute QYs are enhanced from < 0.3% to 59.6% from Au-1 to Au-2 NCs, and a record high value of 90.3% is achieved in the terminal Au-3 NCs.”

Page 24, Line 1-2:

Additional reference has been added to support our statement:

67. Kang, X. et al. The tetrahedral structure and luminescence properties of bimetallic $Pt_1Ag_{28}(SR)_{18}(PPh_3)_4$ nanocluster. *Chem. Sci.* **8**, 2581–2587 (2017).

10. *Some grammar/word choices could be improved. For example, the authors use the following in the abstract that need to be restated for grammar/clarity: “restriction of structural dynamic”; “yet remains challenging in consideration of the metal kernel as its low accessibility”; “resultant of marginal change” etc.*

Reply: We really appreciated the reviewer’s good suggestions to improve the quality of our manuscript. We have modified the related grammar/word accordingly and carefully checked out the related contents throughout the manuscript for general and impactful readability.

11. *In general, the authors are trying to be too “fancy” with their word choices in many sentences, but it is making the paper very challenging to read. Simplifying the sentences could improve readability. For example, “This scenario therewith impedes the nonradiative internal conversion and structural relaxation of electron dynamics [essentially, nonradiative internal conversion], rendering the Au NCs with strong emission. The presented study exemplifies the crucial linkage between surface chemistry and core-state emission of metal NCs and deepens the concept of the metallic*

molecule for metal NCs [restate] by regulating the interior metal kernel involved motion [aka core vibrations] toward a bright emission.”“exterior interference immunity to surroundings” [?] “allow metal NCs more dynamic” Note: “dynamic” is not used properly throughout the paper. I’m not sure in these places if “dynamics”, “vibrational motion”, or other word choices would be best. Moreover, I really don’t understand what the authors mean by “accessing” the metal kernel. If they were talking about catalysis, I would understand them referring to the ability of a small molecule to bind to the metal kernel. Here, they say “we report accessing the metal kernel of metal NCs to suppress their low-frequency acoustic vibration”; “negative-liner”. There are quite a few fragment sentences as the paper goes on.

Reply: We sincerely appreciate the reviewer for his/her great efforts in carefully reviewing our manuscript, and feel sorry for giving you less enjoyment of reading brought by improper phrasing. To minimize any possible confusion, we have simplified these sentences to improve the readability and scientific content of our manuscript. The corresponding changes have been highlighted in yellow for ease of identification. We hope the current revision is satisfactory to the reviewer.

Reviewer comments, second round –

Reviewer #1 (Remarks to the Author):

The answers by the author to my questions and criticisms are well motivated and convincing. In my opinion the paper can be accepted for a publication in Nature Communications. I reviewed only briefing authors replies to other three reviewers, which raised many pertinent and detailed remarks.

I do not share one of the considerations added: "Plenty of pioneering works have confirmed that the oscillation frequency is an intrinsic parameter of metal NCs, and it shows less connection with their size with respect to the discovered empirical relationship of $\nu = 50/R$ (ν and R are frequency and particle diameter, respectively) in Au NPs". I would not state that the oscillation frequency is an "intrinsic" parameter of the Au clusters. Authors should better say that vibrational frequency still depends on the overall size of the clusters, but with a dependence more irregular with respect to larger spherical gold nanoparticles due to the discrete structure of Au clusters and their ligands. Also, in the expression $50/R$ (if this expression is maintained) please specify dimensions of ν and R (cm^{-1} ? Hz? R is nm?).

Reviewer #2 (Remarks to the Author):

The authors responded in detail to all reviewers and they improved the manuscript. After reading through the response to all reviewers (47 pages) I am still not completely convinced if this work meets the high standards of Nature Communications. There are some points where this reviewer would wish to have more clear evidence.

It starts with the structure of the clusters. The authors write that the three samples have "nearly identical" cores, but the exact structure is unknown (no crystal structure). Based on the quite different absorption spectra for cluster Au-1 compared to the other samples one would wish to have more evidence on the structure (XANES does not help much in this context).

The absence of SE remains somewhat mysterious. Also, the authors should elaborate on the exact relationship between the amplitude of the oscillation in femtosecond time-resolved signal and the amplitude of the low-frequency coherent oscillation to substantiate their claim. Is there a fundamental theoretical relationship?

Reviewer #3 (Remarks to the Author):

After reading all the revised manuscript and all the responses to the referees' comments, I found that this work is still some misleading though the authors have made a lot of improvements. From my personal opinion, I like to recommend for accepting this work for NC, but there are still some shortages.

1) the work is based on a proposed structure, while most of the interpretation indeed required real structure, for example, the authors should give some evidence to show what the oscillation from as shown in their fs-TA data.

2) The interpretation of the observed oscillation in fs TA data is not correct, where the oscillation in fs-TA should be the excited state electronic-phonon coupling from the lowest frequency of the vibrational structure of clusters, the authors cannot see the high-frequency vibration in their IRF used for fs-TA measurements because the high frequency vibration was quickly relaxed to the lowest frequency vibrational state during excited state relaxation. In fact, generally, if the amplitude of oscillation became smaller, the fluorescence should become weaker rather than stronger.

Replies to reviewers' comments and descriptions of revisions made

Comments by Reviewer #1:

The answers by the author to my questions and criticisms are well motivated and convincing. In my opinion the paper can be accepted for a publication in Nature Communications. I reviewed only briefing authors replies to other three reviewers, which raised many pertinent and detailed remarks.

Reply: We are very glad to learn that our revisions are almost satisfactory to the reviewer, along with the reviewer's recommendation of accepting our paper in *Nature Communications*. The following listed comment is taken careful consideration and revised accordingly. We would like to thank the reviewer again for his/her constructive comments and suggestions, which have spurred significant improvements in both the scientific content and readability of this manuscript.

Specific comments:

1. I do not share one of the considerations added: "Plenty of pioneering works have confirmed that the oscillation frequency is an intrinsic parameter of metal NCs, and it shows less connection with their size with respect to the discovered empirical relationship of $\nu = 50/R$ (ν and R are frequency and particle diameter, respectively) in Au NPs". I would not state that the oscillation frequency is an "intrinsic" parameter of the Au clusters. Authors should better say that vibrational frequency still depends on the overall size of the clusters, but with a dependence more irregular with respect to larger spherical gold nanoparticles due to the discrete structure of Au clusters and their ligands. Also, in the expression $50/R$ (if this expression is maintained) please specify dimensions of ν and R (cm⁻¹? Hz? R is nm?).

Reply: We appreciate the reviewer's careful and rigorous attitude toward the scientific expression of the oscillation frequency of NCs. We are also in complete agreement with the reviewer's view that the oscillation frequency is not an intrinsic parameter of Au NCs, but should be better stated as that the vibration frequency of Au NCs still depends on their overall size, but have a more irregular dependence on respect to larger spherical gold NPs due to the discrete structure of Au NCs and their ligands. We have revised the corresponding statement in the manuscript accordingly. In addition, the expression of $\nu = 50/R$ in this paragraph is deleted.

Revisions:

Page 13, Line 19-22:

“Plenty of pioneering works have confirmed that the oscillation frequency of Au NCs still depends on their overall size, but has a more irregular dependence with respect to larger spherical gold NPs due to the discrete structure of Au NCs and

their ligands (Supplementary Table 8)⁶⁶⁻⁶⁸.”

Comments by Reviewer #2:

The authors responded in detail to all reviewers and they improved the manuscript. After reading through the response to all reviewers (47 pages) I am still not completely convinced if this work meets the high standards of Nature Communications. There are some points where this reviewer would wish to have more clear evidence.

Reply: We are very glad that the reviewer appreciates our improvement on the manuscript, many of which are spurred by the significant comments and suggestions from the reviewers. The remaining concerns about the structure of Au NCs, SE signals, and the relationship between two different kinds of amplitude have been taken into careful consideration and responded to the specific comments point-to-point.

Improving the photoluminescent performance of metal NCs is a long-pursued goal to promote practical applications in lighting, sensing, bioimaging, etc. Nevertheless, the lack of a detailed understanding of the fundamental photophysical mechanisms underlying their emissions has greatly hampered the rational design of metal NCs with improved and tailored optical properties (*Science* **2016**, 353, 571; *Science* **2018**, 361, 645; *Science* **2018**, 361, 686). In our work, a layer-by-layer triple-ligands self-assembly strategy was designed and applied on the surface of Au NCs in leveraging multiple supramolecular interactions. Surprisingly, the absolute PLQY in the third NCs of Au-3 was recorded to be $90.3 \pm 3.5\%$, which is the highest value reported so far for metal NCs in an aqueous solution. The corresponding mechanistic investigations by employing the *fs*-TA technique successfully demonstrated that the improved surface rigidity in Au NCs can deliver the confinement effect into the metal core, leading to a remarkable reduction in the amplitude of low-frequency coherent oscillation of metal core. This effect further works to significantly weaken the non-radiative-related pathway of metal core-directed structural relaxation of electron dynamics and devote more energy to core-state fluorescence boosting. The presented layer-by-layer triple-ligands self-assembly strategy also can be treated as a popularized method to enrich the surface chemistry of metal NCs in diverse colloid environments for broadening their scope of practical applications. Therefore, we believe this work will be of interest to the wide and heterogeneous readers of *Nature Communications*, and it will rapidly stimulate more studies on noble metal materials, luminescent materials, cluster chemistry, colloidal and interface chemistry, and optical physics.

We would like to show our best respect and greatest appreciation to the reviewer again for taking his/her comments/suggestions, which are very professional, valuable, and pertinent. During the responding process to the reviewer's comments/suggestions, we increased our cognition of the *fs*-TA technique, *fs*-TA data processing, and kinetic trace fitting, and corrected our misunderstandings in time.

Revisions:

Page 3, Line 1-3:

“... the lack of a detailed understanding of the underlying photophysical mechanisms has greatly hampered the rational design of metal NCs with improved and tailored optical properties. To date ...”

Specific comments:

1. It starts with the structure of the clusters. The authors write that the three samples have “nearly identical” cores, but the exact structure is unknown (no crystal structure). Based on the quite different absorption spectra for cluster Au-1 compared to the other samples one would wish to have more evidence on the structure (XANES does not help much in this context)

Reply: Thank you for your insightful comment. Considering the open challenge to acquire high-quality single crystals of water-soluble metal NCs and thus their precise structure, we completely agree with the reviewer’s concern about the preciseness of our phrasing of “nearly identical cores” in the main text. By revisiting the variation of core-related absorption peaks around 500 nm in serial Au NCs, we are aware that the strains in the motif interfaces in metal NCs can affect their core structure (*Chem* **2021**, *7*, 2227), leading to the significant distortion of metal core. We, therefore, changed our statement to be that the core structure is slightly distorted from Au-1 to Au-2 NCs, but almost maintained from Au-2 to Au-3 NCs. Our previous claim of “nearly identical cores” is also revised throughout the manuscript.

Notably, the lacking precise structure does not affect our emphasis on the significant contribution of the suppression of kernel vibration to the emission enhancement of metal NCs: **i)** the emission peak position and the full width at half maximum (FWHM) values are almost unchanged in the serial Au NCs (Figure 1b in the revised manuscript and Table S2 in the revised supplementary file); **ii)** combining with the measurements of absolute PLQY upon near-band gap excitation, it is clarified that the radiative decay rate (k_r) is basically unchanged when compared to the ~57.4-fold reduction in the non-radiative decay rate (k_{nr} , Table S1 in the revised supplementary file); **iii)** there is no redox in the metal core during the applying process of triple-ligands layer-by-layer self-assembly, as evidenced by their XPS results (Figure 2b in the revised manuscript); **iv)** the core compositions of serial Au NCs are identical, evidenced by serial MALDI-TOF mass spectra (Figure S11 in the revised supplementary file). To sum up, the electronic structure and corresponding photoluminescence mechanism of the serial Au NCs are consistent. Only the different kernel vibrations induced non-radiative relaxation of electron dynamics are involved, resulting in the significantly boosted absolute PLQY from Au-1, Au-2, to Au-3 NCs.

Revisions:

Page 5, Line 2-4:

“The differences in the absorption characters between Au-1 and Au-2 NCs indicate that the core structure is slightly distorted from Au-1 to Au-2 NCs, but almost maintained from Au-2 to Au-3 NCs.”

2. *The absence of SE remains somewhat mysterious.*

Reply: We sincerely thank the reviewer for his/her very professional and valuable comment, which motivates us to revisit the basic concepts and their relationship to each other on the *fs*-TA technique. We completely agree with the reviewer’s viewpoint that SE should be as prominent as GSB signals, which is also coincident with our TA data. After rechecking our *fs*-TA data, we rationalize that the ambiguous SE signals should overlap with the broad GSB band due to the small Stokes shift in the Au NCs (taking Au-2 NCs sample as an example, Figure R1), adding challenges to distinguish them from the intrinsic GSB signals. In this regard, we have consciously analyzed the kinetic decay by selecting the pure ESA signals around 530, 545, and 545 nm for Au-1, Au-2, and Au-3 NCs, respectively.

Figure R1. Absorption and PL spectra of Au-2 NCs (top plane). The absorption spectrum is normalized at 300 nm. The corresponding TA spectra of Au-2 NCs within different decay times were pumped by a 400 nm laser (bottom plane).

3. *Also, the authors should elaborate on the exact relationship between the amplitude of the oscillation in femtosecond time-resolved signal and the amplitude of the low-frequency coherent oscillation to substantiate their claim. Is there a fundamental*

theoretical relationship?

Reply: Thank you for your suggestion. The as-observed oscillation in *fs*-TA data is caused by the excited state electronic-phonon coupling from the vibrational structure of metal NCs. We first extracted the pure acoustic oscillation signals from *fs*-TA data by subtracting the electron dynamics. After applying Fast Fourier Transform (FFT), only one FFT peak with similar low vibrational frequencies was found for the serial Au NCs. It indicates the observed oscillation in the time-resolved *fs*-TA signal exclusively corresponds to that of metal core-involved low-frequency coherent oscillation. Namely, the amplitude of oscillation in the time-resolved *fs*-TA signal reflects completely the amplitude of low-frequency coherent oscillation

In addition, inspired by the reviewer's suggestion, we have completed a careful literature survey to figure out how to quantitatively evaluate the amplitude contribution of low-frequency coherent oscillation to the overall amplitude of oscillation in the time-resolved *fs*-TA data, in those cases of the recorded oscillation in *fs*-TA data composed of multiple coherent oscillations (e.g., high-, low-frequency, and others). However, there is still no fundamental theoretical relationship established.

Comments by Reviewer #3:

After reading all the revised manuscript and all the responses to the referees' comments, I found that this work is still some misleading though the authors have made a lot of improvements. From my personal opinion, I like to recommend for accepting this work for NC, but there are still some shortages.

Reply: We are glad that the reviewer finds improvements in the revised manuscript, many of which are spurred by the comments and suggestions from the reviewers. We are also excited by the reviewer for his/her recommendation for accepting our work to publish in the esteemed *Nature Communications*. All the remaining concerns have been taken into careful consideration in this revision, and a point-to-point response could be found below.

Specific comments:

*1. The work is based on a proposed structure, while most of the interpretation indeed required real structure, for example, the authors should give some evidence to show what the oscillation from as shown in their *fs*-TA data.*

Reply: Thank you for your insightful comment. We share a similar view with the reviewer that the atomically precious crystal structure of metal NCs is a more direct and solid base to reveal the origin of the observed oscillation in the *fs*-TA data. However, the growth of high-quality single crystals is an open challenge for water-soluble metal NCs due to their labile surface environment.

Although it lacks the crystal structure, we may rationally assign the structural origin of observed oscillation in *fs*-TA data to the metal core of metal NCs, according to the relative long periodic time (0.60~0.65 ps) and short variational frequency (57.3~59.0 cm^{-1}). Such an empirical inference has already been demonstrated for metal NCs both theoretically and experimentally. For example, Fatti *et al.* have experimentally proved the period of acoustic vibrations in serial metal NCs ($\text{Au}_{10}(\text{SPhtBu})_{10}$, $\text{Au}_{15}(\text{SG})_{13}$, $\text{Au}_{25}(\text{SCH}_2\text{CH}_2\text{Ph})_{18}$, $\text{Au}_{25}(\text{SG})_{18}$, and $\text{Au}_{102}(\text{SPhCOOH})_{44}$). Then, they applied the DFT computations to well-assign them as the low-frequency vibrational modes of the metal core (*Nano Lett.* **2018**, 18, 6842). Besides, the metal core-dictated low-frequency vibration was successively determined in other metal NCs' systems, for example, 16.7 cm^{-1} for $\text{Au}_{246}(\text{SR})_{80}$ (*Angew. Chem. Int. Ed.* **2017**, 56, 16257), 52.0 cm^{-1} for $\text{Au}_{144}(\text{SR})_{60}$ (*Chem. Sci.* **2022**, 13, 8124), 25.0 cm^{-1} for $\text{Au}_{38}(\text{SR})_{24}$ (*J. Phys. Chem. C* **2018**, 122, 16380), 16.7 cm^{-1} for $\text{Au}_{30}(\text{SR})_{18}$ (*Science* **2019**, 364, 279), 26.0 cm^{-1} for rod-like $\text{Au}_{25}(\text{SR})_{18}$ (*Proc. Natl. Acad. Sci. U.S.A.* **2017**, 114, E4697), and 40.0 cm^{-1} for spherical $\text{Au}_{25}(\text{SR})_{18}$ (*J. Phys. Chem. C* **2010**, 114, 19935) (Table S8 in the revised supplementary file).

Revisions:

Page 12, Line 14-16:

“For molecular-like metal NCs, a short impulsive laser can excite the coherent acoustic vibration of the metal core for a few to tens of picoseconds.”

2. *The interpretation of the observed oscillation in fs TA data is not correct, where the oscillation in fs-TA should be the excited state electronic-phonon coupling from the lowest frequency of the vibrational structure of clusters, the authors cannot see the high-frequency vibration in their IRF used for fs-TA measurements because the high frequency vibration was quickly relaxed to the lowest frequency vibrational state during excited state relaxation.*

Reply: We are sincerely grateful for the reviewer's venerable correction to our interpretation of the observed oscillation in *fs*-TA data. According to the reviewer's elaborate suggestion and subsequently our detailed literature survey, we completely agree with the interpretation mentioned by the reviewer. We have revised the corresponding statement and supplied additional references (ref. 58-60) in the revised manuscript.

Revisions:

Page 12, Line 17 to Page 13, Line 3:

“The coherent oscillation, which is the reflection of the excited state electronic-phonon coupling from the lowest frequency of the vibrational structure of clusters⁵⁸⁻⁶⁰, thus can be distinguished from the TA decays at an early time at most

probed wavelengths.”

Page 23, Line 14-22:

The following references have been added to support our statement:

58. Iwamura, M., Watanabe, H., Ishii, K., Takeuchi, S. & Tahara, T. Coherent nuclear dynamics in ultrafast photoinduced structural change of bis(diimine)copper(I) complex. *J. Am. Chem. Soc.* **133**, 7728–7736 (2011).
59. Wei, Z. R., Nakamura, T., Takeuchi, S. & Tahara, T. Tracking of the nuclear wavepacket motion in cyanine photoisomerization by ultrafast pump–dump–probe spectroscopy. *J. Am. Chem. Soc.* **133**, 8205–8210 (2011).
60. Iwamura, M., Nozaki, K., Takeuchi, S. & Tahara, T. Real-time observation of tight Au–Au bond formation and relevant coherent motion upon photoexcitation of $[\text{Au}(\text{CN})_2^-]$ oligomers. *J. Am. Chem. Soc.* **135**, 538–541 (2013).

3. *In fact, generally, if the amplitude of oscillation became smaller, the fluorescence should become weaker rather than stronger.*

Reply: Thank you for your insightful comment, which motivated us to take care of disclosing the relationship between the amplitude of oscillation and the intensity of fluorescence.

We notice that the reduction of the amplitude of oscillation can result in weaker fluorescence, especially in those biological fluorescent protein-related systems. For example, the oscillatory-involved excited-state proton transfer (ESPT) reaction (*Nature* **2009**, 462, 200; *J. Am. Chem. Soc.* **2016**, 138, 3942). In parallel, in some systems of AIE molecules (*J. Am. Chem. Soc.* **2021**, 143, 11820), semiconductor quantum dots (*Phys. Rev. B* **1998**, 57, 341), and carbon nanotubes (*Science*, **1997**, 275, 187), the optical coherent oscillation are a result of electronic-phonon coupling in the excited state, which relates to the non-radiative process of excited electrons. It is in general not beneficial to fluorescence enhancement.

In our work, we implemented a layer-by-layer triple-ligands surface engineering to solution-phase Au NCs. We found that the as-rigidified surface imposed by synergistic supramolecular interactions greatly influences the low-frequency acoustic vibration of the metal kernel, resulting in a subtle change in vibration frequency but a remarkable reduction in the amplitude of oscillation. Serial *fs*-TA measurements evidenced that the non-radiative core-directed structural relaxation has been significantly suppressed, leading to the ~57.4-fold reduction in the non-radiative decay rate (Table S1 in the revised supplementary file). As a result, the record-high PLQY of $90.3 \pm 3.5\%$ was successfully achieved in green-emitting Au-3 NCs in an aqueous solution. After experimental and mechanistic investigation, it confirms that the smaller amplitude of oscillation can bring stronger fluorescence in our system. It should be mentioned that similar results were also observed by other researchers in luminescent metal NCs. For

example, Kang *et al.* have observed the prominent coherent oscillations in the *fs* kinetic traces of Pt₁Ag₂₄ NCs but not for Pt₁Ag₂₈ NCs. They speculated that the stronger phonon behaviors observed in Pt₁Ag₂₄ NCs lead to weaker luminescence (0.1% PLQY), in contrast to that of Pt₁Ag₂₈ NCs (4.9% PLQY, *Chem. Sci.* **2017**, 8, 2581).

Reviewer comments, third round –

Reviewer #2 (Remarks to the Author):

I appreciate the effort of the authors and I also think that the work is in principle interesting. However, even after the second revision I feel not comfortable to propose acceptance. If the other reviewers enthusiastically support publication the editor may just ignore my comments.

Concerning my concerns:

1) I think the structure of the cluster is essential for the correct interpretation of the results. The fact that it is difficult to grow crystals for water-soluble clusters should not be used as an excuse. The authors provided other data to provide evidence for the same structure of the cluster but, in view of the very different absorption spectrum of sample 1 compared to the other samples, I am still not convinced.

2) Relationship between the amplitude of the oscillation in femtosecond time-resolved signal and the amplitude of the low-frequency coherent oscillation: The authors write that there is no fundamental relationship established. So how can they then learn something about the amplitude of the low-frequency oscillation from the amplitude of the oscillation in femtosecond time-resolved signal if the relationship is not clear?

Reviewer #3 (Remarks to the Author):

I read all the revision and the responses to the comments from referees. The authors had added most of the necessary interpretation, and some necessary improvement are made in the revised manuscript.

I think this work is ok now for publishing in NC.

Replies to reviewers' comments and descriptions of revisions made

Comments by Reviewer #2:

I appreciate the effort of the authors and I also think that the work is in principle interesting. However, even after the second revision I feel not comfortable to propose acceptance. If the other reviewers enthusiastically support publication the editor may just ignore my comments.

Reply: We greatly appreciate the reviewer's time and efforts in reviewing our manuscript again. We are encouraged by the reviewer for his/her thought that our work is in principle interesting. All remaining concerns have been taken into careful consideration in this revision, and a point-to-point response could be found below.

Concerning my concerns:

1. I think the structure of the cluster is essential for the correct interpretation of the results. The fact that it is difficult to grow crystals for water-soluble clusters should not be used as an excuse. The authors provided other data to provide evidence for the same structure of the cluster but, in view of the very different absorption spectrum of sample 1 compared to the other samples, I am still not convinced.

Reply: Thank you for your insightful comment. We share a similar view with the reviewer that the structure of the cluster is in general essential for the correct interpretation of the results. However, we should state that lacking crystal structure in this work does not affect the conclusion and our emphasis, which have been well-supported by rigorous characterizations and discussions. Namely, the electronic structure and corresponding photoluminescence mechanism of the serial Au NCs are consistent. Only the different kernel vibrations induced non-radiative relaxation of electron dynamics are involved, resulting in the significantly boosted absolute PLQY from Au-1, Au-2, to Au-3 NCs. That is, the suppression of kernel vibration significantly contributes to the emission enhancement of metal NCs in serial Au NCs.

Besides, as the reviewer mentioned, there is a variation of core-related absorption peaks around 500 nm in serial Au NCs. This discrepancy should be caused by the strains in the motif interfaces in metal NCs which lead to the significant distortion of the metal core. This inference is reasonable as the variation of the absorption spectrum only exists between the Au-1 and Au-2 NCs, but is negligible between the Au-2 to Au-3 NCs, because of the remarkable surface rigidity enhancement from the Au-1, Au-2, to Au-3 NCs. To this concern, we have changed our statement, and the previous claim of "nearly identical cores" has already been revised throughout the manuscript.

2. Relationship between the amplitude of the oscillation in femtosecond time-resolved signal and the amplitude of the low-frequency coherent oscillation: The authors write

that there is no fundamental relationship established. So how can they then learn something about the amplitude of the low-frequency oscillation from the amplitude of the oscillation in femtosecond time-resolved signal if the relationship is not clear?

Reply: We regret that we didn't articulate well in some terms. As to the no fundamental relationship established, it is aimed at quantitatively evaluating the amplitude contribution of low-frequency coherent oscillation to the overall amplitude of oscillation in the time-resolved fs-TA data, in those cases of the recorded oscillation in fs-TA data composed of multiple coherent oscillations (e.g., high-, low-frequency, and others). This scenario, however, does not affect the analysis and discussion in this work. This is because that only one Fast Fourier Transform (FFT) peak with similar low vibrational frequencies was found for the serial Au NCs. It indicates the observed oscillation in the time-resolved fs-TA signal exclusively corresponds to that of metal core-involved low-frequency coherent oscillation. Namely, the amplitude of oscillation in the time-resolved fs-TA signal reflects completely the amplitude of low-frequency coherent oscillation.

Comments by Reviewer #3:

I read all the revision and the responses to the comments from referees. The authors had added most of the necessary interpretation, and some necessary improvements are made in the revised manuscript.

I think this work is ok now for publishing in NC.

Reply: We are particularly excited by the reviewer's suggestion of acceptance of our manuscript. The noted improvements in terms of scientific content and readability are largely triggered by the insightful comments from the reviewers. We would like to thank the reviewer for his/her great time and efforts in reviewing our manuscript again. Thank you!